# GAMMA: Gated Multi-hop Message Passing for Homophily-Agnostic Node Representation in GNNs

**Amir Ghazizadeh**[1]    **Rickard Ewetz**[2]    **Hao Zheng**[1]

[1]Department of Electrical and Computer Engineering, University of Central Florida, Orlando, FL, USA
[2]Department of Electrical and Computer Engineering, University of Florida, Gainesville, FL, USA

`{amir.g, hao.zheng}@ucf.edu, rewetz@ufl.edu`

## Abstract

The success of Graph Neural Networks (GNNs) leverages the homophily principle, where connected nodes share similar features and labels. However, this assumption breaks down in heterophilic graphs, where same-class nodes are often distributed across distant neighborhoods rather than immediate connections. Recent attempts expand the receptive field through multi-hop aggregation schemes that explicitly preserve intermediate representations from each hop distance. While effective at capturing heterophilic patterns, these methods require separate weight matrices per hop and feature concatenation, causing parameters to scale linearly with hop count. This leads to high computational complexity and GPU memory consumption. We propose Gated Multi-hop Message Passing (GAMMA), where nodes assess how relevant the aggregated information is from their k-hop neighbors. This assessment occurs through multiple refinement steps where the node compares each hop's embedding with its current representation, allowing it to focus on the most informative hops. During the forward pass, GAMMA finds the optimal mix of multi-hop information local to each node using a single feature vector without needing separate representations for each hop, thereby maintaining dimensionality comparable to single hop GNNs. In addition, we propose a weight sharing scheme that leverages a unified transformation for aggregated features from multiple hops so the global heterophilic patterns specific to each hop are learned during training. As such, GAMMA captures both global (per-hop) and local (per-node) heterophily patterns without high computation and memory overhead. Experiments show GAMMA matches or exceeds state-of-the-art heterophilic GNN accuracy, achieving up to $\approx 20\times$ faster inference. Our code is publicly available at `https://github.com/amir-ghz/GAMMA`.

## 1   Introduction

Graph Neural Networks (GNNs) [14, 10, 34] have become the standard model for learning graph-structured data across domains such as social and molecular networks [9, 16]. Their core strength lies in message passing [33], which iteratively aggregates features from local neighborhoods to capture structural information. However, most standard GNNs implicitly rely on the homophily assumption, where connected nodes are likely to share similar labels or features, limiting their effectiveness on heterophilic graphs, where connected nodes frequently belong to different classes [40, 39].

Conventional GNNs, such as GCN [14] and GAT [34] are agnostic to feature similarity of aggregated nodes [22, 40]. Such GNNs function as low-pass filters, smoothing representations across neighborhoods [25], which enhances class signals in homophilic graphs but obscures them in heterophilic ones. This fundamental limitation has motivated significant research into heterophily-specific GNN architectures [39].

Existing approaches to heterophilic graph learning broadly fall into several categories. Some works aim to capture global heterophily by enabling nodes to aggregate information from potentially

39th Conference on Neural Information Processing Systems (NeurIPS 2025).

similar nodes graph-wide. For example, one strategy requires pre-computation to estimate structural similarities, as explored by methods like SimP-GCN [13] and Geom-GCN [29]. Other approaches utilize learned affinity matrices to dynamically capture global heterophily patterns, exemplified by models such as NL-GNN [20] and CPGNN [41]. Alternatively, some methods incorporate higher-order neighborhoods, seeking same-class nodes from long range dependencies [1, 29]. For example, H2GCN [40] emphasizes the aggregation of 2-hop neighbors while MixHop [1] concatenates features of nodes upto k-hops so each node is represented using its multi-hop features as its intermediate representation. Unfortunately, incorporating higher-order features requires compute and memory intensive operations that are not GPU friendly.

Despite these efforts, developing GNNs that are both effective and scalable across the diverse landscape of heterophilic graphs remains a significant challenge. Most importantly, computation efficiency has been largely an afterthought in heterophilic GNN design. Our analysis reveals that state-of-the-art heterophilic GNNs like H2GCN [40] can consume over $6\times$ more GPU memory than vanilla GCN with a $33\times$ increase in execution time, a prohibitive cost for many real-world applications.

Moreover, real-world graphs rarely exhibit purely homophilic or heterophilic structures. Instead, they display mixed homophily-heterophily patterns, where optimal feature aggregation varies across different nodes and hops [23]. For instance, fraud detection graphs are largely heterophilic, where fraudulent users often connect to legitimate ones, yet include homophilic clusters of coordinated fraudulent activities [6]. Similarly, protein-protein interaction networks, though generally heterophilic, contain homophilic subclusters of functionally related proteins [21]. Thus, GNNs must adopt message passing schemes capable of modeling global heterophily while exploiting localized homophily.

To address these challenges, we propose *Gated Multi-hop Message Passing (GAMMA)*, a novel GNN architecture for heterophilic graphs that balances accuracy with computational efficiency. Our contributions are summarized as follows:

- **Comprehensive computational analysis:** We provide the first systematic analysis of computational requirements and performance bottlenecks across 13 state-of-the-art heterophilic GNN architectures, revealing critical efficiency gaps in current designs.

- **Adaptive node-specific message passing:** We introduce a lightweight iterative gating mechanism that enables each node to dynamically assess the relevance of aggregated information from its k-hop neighbors, capturing node-specific local heterophily patterns on the fly.

- **Efficient weight sharing scheme:** We propose a parameter-efficient approach using learnable scaling vectors that leverages a unified transformation for features from multiple hops, capturing global heterophilic patterns specific to each hop during training while significantly reducing memory requirements by up to 12 times compared to the state of the art.

- **State-of-the-art performance with efficiency:** Extensive experiments demonstrate that GAMMA matches or exceeds state-of-the-art heterophilic GNN accuracy while achieving up to $20\times$ faster inference and substantially lower memory consumption on commodity GPUs.

By capturing both global (per-hop) and local (per-node) heterophily patterns without high computational overhead, GAMMA makes heterophilic graph learning practical for real-world applications on commodity hardware.

## 2 Heterophilic Graph Characteristics and Performance Analysis

Our investigation reveals two critical aspects of heterophilic graph learning: the complex, non-uniform heterophily distributions across node neighborhoods and the significant computational overhead of current heterophilic GNN designs.

**Heterophily is not a monolithic property.** Most existing GNNs assume uniform heterophily patterns across all nodes, but our analysis reveals that real-world heterophilic graphs exhibit complex, node-specific patterns that vary significantly across hop distances. The node homophily ratio is the proportion of a node's neighbors that share the same class label as the node itself [30].

Fig. 1 demonstrates two distinct manifestations of this heterogeneity. First, we observe *dataset-level heterophily evolution*: different datasets exhibit fundamentally different patterns of how homophily changes with hop distance. The Cornell dataset shows a non-monotonic, oscillating pattern, where 1-hop neighborhoods exhibit severe heterophily (79.8% of nodes having homophily ratios $\leq 0.05$),

which increases substantially at 2-hops (mean of 0.456), but then decreases again at 3-hops (mean of 0.266). In contrast, the Tolokers dataset displays a progressively increasing homophily trend with distance ($0.634 \rightarrow 0.654 \rightarrow 0.660$ for 1-, 2-, and 3-hops respectively), suggesting that same-class nodes become more accessible at greater distances.

However, the more critical insight lies in the *node-level heterophily diversity*. Within the same dataset and hop distance, individual nodes exhibit dramatically different homophily patterns. For instance, in Cornell's 2-hop neighborhoods, while 44.8% of nodes concentrate around high homophily (0.72), others span the entire spectrum from 0.02 to 1.0. This variance indicates that Node $i$ might experience a heterophily trajectory of [high, low, medium] across 1-2-3 hops, while Node $j$ in the same graph follows an entirely different pattern such as [low, high, low].

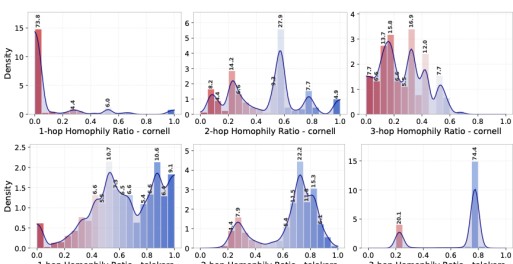

Figure 1: Node homophily density across 3-hop distances in cornell and tolokers.

Unlike global parameters that assume all nodes benefit equally from information at a specific hop distance, this node-specific heterophily requires adaptive mechanisms that allow each node to identify which hop distances contain the most relevant class information for its individual prediction task.

These observations reveal that heterophily is neither a monolithic graph-level property nor a uniform node-level characteristic. Instead, it manifests as a multi-faceted phenomenon with both global structural patterns (varying across datasets) and highly individualized local neighborhood configurations (varying across nodes within the same graph), necessitating architectures that can adapt to both levels of heterogeneity simultaneously.

**GPU efficiency analysis.** Current heterophilic GNN architectures face significant performance challenges on commodity hardware. While efficient GPU execution relies on dense matrix multiplications, heterophilic multi-hop GNNs often require recursive sparse-dense operations, feature concatenation [1], and intermediate storage [40] that substantially increase computational overhead. From a memory perspective, high usage arises from concatenating features across multiple hops and caching intermediate embeddings. For instance, BernNet's $K$th-order filter requires $K$ separate sparse–dense multiplications, each a distinct pass over the edge list with its own kernel overhead and intermediate storage.

From a compute standpoint, many low-level operations are executed as dozens of small, memory-bound kernels. This overhead becomes dominant on large graphs, even when the asymptotic complexity aligns with a simple GCN. For example, in M2M2GCN [18], each edge triggers a small MLP and a segmented softmax over a $c$-dimensional vector. These operations cannot be cast as one large matrix–matrix multiply, so they incur per-edge kernel launches and intermediate result synchronization, further increasing runtime and reducing GPU efficiency.

To quantify these costs, we benchmarked 13 representative GNN variants—each configured with the same hidden dimension and number of layers on the Flickr dataset using an NVIDIA RTX A2000 GPU.

As illustrated in Fig. 2, H2GCN, consumes over $6\times$ more GPU memory than a vanilla GCN and incurs a $33\times$ increase in combined forward and backward time. Moreover, memory footprint does not always predict execution time. BernNet uses roughly one-third the memory of GAT yet runs slower, owing to its intricate graph operations (e.g., $K$th-order filter calculation) and frequent GPU memory transactions. These results underscore a fundamental trade-off between expressive power and computational efficiency. These findings underscore the need for architectures that combine heterophilic modeling capacity with practical computational efficiency.

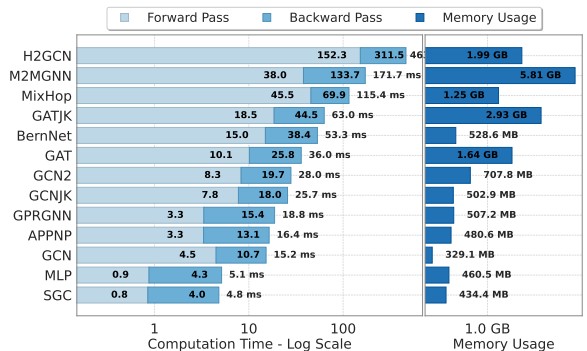

Figure 2: memory and runtime analysis across different GNN models for Flickr dataset.

## 3 Preliminaries

Let $G = (V, E)$ be an undirected graph with $n = |V|$ nodes and $m = |E|$ edges. Each node $v \in V$ has an associated feature vector $\mathbf{x}_v \in \mathbb{R}^{d_{\text{in}}}$, and we denote the node feature matrix as $\mathbf{X} \in \mathbb{R}^{n \times d_{\text{in}}}$. The graph structure is represented by an adjacency matrix $\mathbf{A} \in \mathbb{R}^{n \times n}$, where $\mathbf{A}_{ij} > 0$ if $(i, j) \in E$, and $\mathbf{A}_{ij} = 0$ otherwise. We primarily consider binary adjacency matrices ($\mathbf{A} \in \{0, 1\}^{n \times n}$), though our formulation generalizes to weighted graphs. Let $\mathbf{D} = \text{diag}(d_1, d_2, \ldots, d_n)$ denote the degree matrix of $G$, where $d_i = \sum_{j=1}^n \mathbf{A}_{ij}$ is the degree of node $i$. The normalized adjacency matrix with self-loops is defined as $\tilde{\mathbf{A}} = \mathbf{D}^{-\frac{1}{2}}(\mathbf{A} + \mathbf{I}_n)\mathbf{D}^{-\frac{1}{2}}$, where $\mathbf{I}_n$ is the $n \times n$ identity matrix. In semi-supervised node classification, only a subset of nodes $V_L \subset V$ have known labels $\mathbf{y}_v \in \{1, 2, \ldots, C\}$, where $C$ denotes the number of classes. The goal is to predict labels for all unlabeled nodes in $V \setminus V_L$.

**Message Passing and Multi-hop Propagation.** GNNs operate on the principle of message passing, where each node iteratively aggregates information from its neighborhood. The neighborhood $\mathcal{N}(v)$ of a node $v$ is defined as the set of nodes adjacent to $v$ in $G$, i.e., $\mathcal{N}(v) = \{u \in V \mid (v, u) \in E\}$.

The standard message passing framework can be formalized as follows. At each layer $l$, a node $v$ updates its representation by aggregating information from its immediate neighbors:

$$\mathbf{h}_v^{(l)} = \text{UPDATE}^{(l)}\left(\mathbf{h}_v^{(l-1)}, \text{AGGREGATE}^{(l)}\left(\left\{\mathbf{h}_u^{(l-1)} : u \in \mathcal{N}(v)\right\}\right)\right) \tag{1}$$

where $\mathbf{h}_v^{(l)} \in \mathbb{R}^{d_l}$ is the representation of node $v$ at layer $l$, with $\mathbf{h}_v^{(0)} = \mathbf{x}_v$. The functions AGGREGATE and UPDATE are typically parameterized by learnable weights and vary across different GNN architectures. In matrix form, many GNN layers can be expressed as:

$$\mathbf{H}^{(l)} = \sigma\left(\mathbf{A}\mathbf{H}^{(l-1)}\mathbf{W}^{(l-1)}\right) \tag{2}$$

where $\mathbf{H}^{(l)} \in \mathbb{R}^{n \times d_l}$ is the matrix of node representations at layer $l$ (with $\mathbf{H}^{(0)} = \mathbf{X}$), $\mathbf{W}^{(l-1)} \in \mathbb{R}^{d_{l-1} \times d_l}$ is a learnable weight matrix, $\sigma$ is a non-linear activation function, and $\mathbf{P} \in \mathbb{R}^{n \times n}$ is a propagation matrix derived from the graph structure. $\mathbf{A}$ is the normalized adjacency matrix $\tilde{\mathbf{A}}$.

The standard message passing paradigm focuses on immediate neighbors (1-hop connectivity), but multi-hop information is critical in heterophilic settings. We define the $p$-hop neighborhood of a node $v$ as:

$$\mathcal{N}^p(v) = \{u \in V \mid \text{dist}(v, u) = p\} \tag{3}$$

where $\text{dist}(v, u)$ is the length of the shortest path between nodes $v$ and $u$. By convention, $\mathcal{N}^0(v) = \{v\}$ (the node itself).

The $p$-hop connectivity in a graph can be represented by the $p$-th power of the adjacency matrix, $\mathbf{A}^p$. For an unweighted graph, the entry $(\mathbf{A}^p)_{ij}$ counts the number of distinct walks of length $p$ from node $i$ to node $j$. In particular, $(\mathbf{A}^p)_{ij} > 0$ if and only if there exists at least one walk of length $p$ between $i$ and $j$. If $(\mathbf{A}^p)_{ij} > 0$, then $j \in \cup_{q=0}^p \mathcal{N}^q(i)$, meaning node $j$ is reachable from node $i$ via a walk of length at most $p$. We adopt the convention that $\mathbf{A}^0 = \mathbf{I}_n$, representing self-connections.

## 4 GAMMA: Gated Multi-hop Message Passing

We now introduce the Gated Multi-hop Message Passing (GAMMA) mechanism. GAMMA explicitly addresses a fundamental yet unresolved limitation in current multi-hop GNNs: the inability to adaptively modulate the non-uniform heterophilic patterns specific to each node when aggregating neighborhood features from multiple hops.

Following prior work [1, 40], to leverage multi-hop neighborhoods, features from higher powers of the adjacency matrix are explicitly computed. For a feature matrix $\mathbf{X}$, the $p$-hop propagated features can be expressed as $\mathbf{A}^p\mathbf{X}$, where each row $(\mathbf{A}^p\mathbf{X})_i$ aggregates information from all nodes exactly $p$ hops away from node $i$. A multi-hop aggregation scheme formulates node representations by concatenating features processed independently for each hop distance:

$$\mathbf{Z}_{\text{concat}} = \mathop{\Big\|}_{p=0}^{K}\left(\mathbf{A}^p\mathbf{X}\mathbf{W}^{(p)}\right) \tag{4}$$

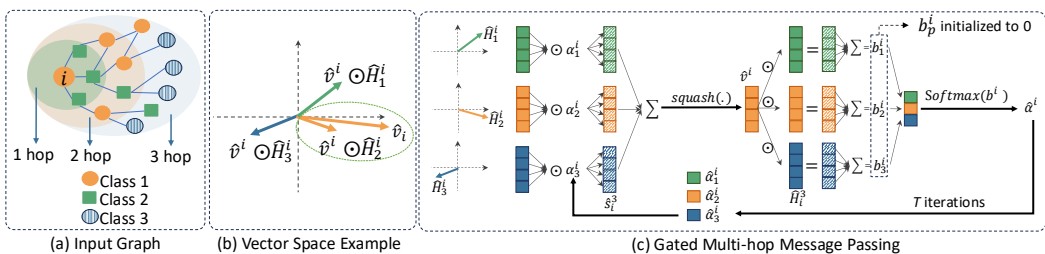

Figure 3: Gated Multi-hop Message Passing

where $\mathbf{W}^{(p)} \in \mathbb{R}^{d_{\text{in}} \times d_{\text{out}}}$ is a hop-specific weight matrix, and $K$ is the maximum hop distance considered. (Note: The resulting dimension of $\mathbf{Z}_{\text{concat}}$ will be $N \times (K+1)d_{\text{out}}$ if each $\mathbf{A}^p \mathbf{X} \mathbf{W}^{(p)}$ is $N \times d_{\text{out}}$.)

We aim to avoid pitfalls associated with naive feature concatenation [1, 40, 37] due to several key concerns. Primarily, feature concatenation can lead to parameter inefficiency and higher risk of overfitting by drastically expanding the feature dimensionality fed into subsequent layers, demanding more data and computational resources than potentially necessary. Secondly, it often introduces significant information redundancy, as concatenated features from different hops might be related, forcing the model to expand capacity on disentangling correlated signals rather than learning novel patterns. Crucially, by naively concatenating feature vectors from multiple hops, the complex task of distinguishing interactions and relative contributions among is entirely offloaded to downstream general-purpose layers (like an MLP). This may be less effective than employing more sophisticated, adaptive fusion mechanisms that can explicitly model and exploit these relationships earlier and more directly, ultimately aiming for a robust representation learning process.

To overcome these limitations, we adopt inspiration from the observation that Capsule Networks [32] dynamically link spatially local features to global object representations. Specifically, we treat each $p$-hop aggregation as a constituent part of the full node embedding. Then, we measure how well each part's embedding aligns (via dot product) with the node's evolving feature (i.e., final representation), thereby adaptively integrating only those hops whose local aggregated features agree with the overall representation.

Formally, let $\{\hat{\mathbf{H}}_i^{(p)}\}_{p=0}^K$ denote normalized hop-specific embeddings for node $i$ at hop $p$, each derived from multi-hop propagation, $\mathbf{A}^p$. These embeddings are first normalized via $L_2$-normalization to mitigate scale discrepancies among hops:

$$\hat{\mathbf{H}}_i^{(p)} = \frac{\mathbf{H}_i^{(p)}}{\|\mathbf{H}_i^{(p)}\|_2}, \quad p = 0, \ldots, K. \tag{5}$$

These normalized embeddings are then stacked into a unified tensor representation, $\mathbf{U}_i \in \mathbb{R}^{(K+1) \times d_{\text{out}}}$, to facilitate subsequent iterative gating computations.

The core of GAMMA is an iterative gating procedure, driven by a set of gating logits $\mathbf{b}_i \in \mathbb{R}^{K+1}$, initialized to zero. Each iteration $t$, refines these logits, which encode the node's confidence in each hop's contribution to its final representation. Specifically, at iteration $t$, we compute $\alpha_i^{(t)}$, which is the hop-wise gating coefficient for node $i$ through a softmax operation:

$$\alpha_i^{(t)} = \text{softmax}\left(\mathbf{b}_i^{(t)}\right), \quad \alpha_{i,p}^{(t)} = \frac{\exp(b_{i,p}^{(t)})}{\sum_{q=0}^K \exp(b_{i,q}^{(t)})}. \tag{6}$$

These gating coefficients determine how hop-specific embeddings are combined into an intermediate representation $\mathbf{s}_i^{(t)}$:

$$\mathbf{s}_i^{(t)} = \sum_{p=0}^K \alpha_{i,p}^{(t)} \hat{\mathbf{H}}_i^{(p)}. \tag{7}$$

To maintain meaningful vector magnitudes and preserve directional information, we employ the squash function [32], to transform $\mathbf{s}_i^{(t)}$ into a normalized representation $\mathbf{v}_i^{(t)}$:

$$\mathbf{v}_i^{(t)} = \frac{\|\mathbf{s}_i^{(t)}\|^2}{1 + \|\mathbf{s}_i^{(t)}\|^2} \frac{\mathbf{s}_i^{(t)}}{\|\mathbf{s}_i^{(t)}\|}. \tag{8}$$

The key improvement that enables distinguishing features from different hops, is the iterative update of gating logits based on the concept of *agreement*, quantified as the inner product between each hop-specific embedding and the evolving node representation:

$$b_{i,p}^{(t+1)} = b_{i,p}^{(t)} + \hat{\mathbf{H}}_i^{(p)} \cdot \mathbf{v}_i^{(t)}. \tag{9}$$

Intuitively, hops whose embeddings consistently align with the emerging consensus (as represented by $\mathbf{v}_i^{(t)}$) experience increased gating logits and thus higher gating coefficients, reinforcing their influence. Conversely, hops providing contradictory or noisy signals are progressively suppressed. This is illustrated in Fig. 3(b), where the aggregated embedding from the second hop has the strongest agreement with the target node $i$ (Figure 3(a)). Consequently, the dot product $\hat{\mathbf{H}}_i^{(2)} \cdot \mathbf{v}_i^{(t)}$ yields a higher value, which increases the corresponding gating logit $b_{i,2}^{(t+1)}$ for the next iteration, thereby intensifying the influence of the second hop's features in the final representation.

After $R$ routing iterations, we finalize the node representation: $\mathbf{H}_i = \mathbf{v}_i^{(R)} + \mathbf{b}$, where $\mathbf{b} \in \mathbb{R}^{d_{\text{out}}}$ is an optional learnable bias. The number of routing iterations is a hyperparameter, with 2 or 3 iterations typically sufficient in practice (see Appendix A).

It is crucial to distinguish between *hop-specific* feature aggregation and *node-specific* gating. The initial multi-hop propagation ($\mathbf{H}_i^{(p)} = (\mathbf{A}^p \mathbf{X} \mathbf{W})_i$) generates hop-specific embeddings that aggregate features globally from all nodes at distance $p$. However, the gating mechanism operates *node-specifically*: each node $i$ independently computes its own routing coefficients $\alpha_{i,p}^{(t)}$ based on the agreement between its evolving representation $\mathbf{v}_i^{(t)}$ and each hop's embedding $\hat{\mathbf{H}}_i^{(p)}$. These coefficients are unique to each node and recomputed during every forward pass (including inference on unseen graphs), enabling adaptive, instance-driven weighting of multi-hop information without gradient updates. This distinguishes GAMMA from methods like GPR-GNN [5] where coefficients are learned once globally and applied uniformly to all nodes.

From a practical computational viewpoint, GAMMA offers substantial benefits over concatenation-based multi-hop methods. Since GAMMA dynamically selects and aggregates hop-specific information without explicitly expanding the feature space, it significantly reduces memory consumption and computational overhead. Furthermore, by maintaining a single shared projection matrix across hops, GAMMA leverages efficient sparse-dense matrix multiplications, substantially improving training scalability and inference efficiency. Moreover, During inference on an unseen graph, the routing loop re-computes the gating coefficients $\alpha_{i,p}$ *from scratch*, enabling GAMMA to re-calibrate hop importance for heterophily patterns specific to each node's local neighborhood *without* any gradient updates.

Empirically, the adaptive nature of the GAMMA routing mechanism provides unparalleled flexibility, allowing each node to selectively amplify structurally informative signals from distinct hop distances, based on the nuanced local context. This flexibility not only enhances performance on highly heterophilic datasets but also robustly generalizes across diverse graph structures, significantly outperforming existing multi-hop approaches, as demonstrated extensively in our experimental results. For more details about GAMMA and the algorithm pseudocode, refer to Appendix C and Algorithm 1.

## 5 Weight Sharing Across Hops

The iterative gating mechanism in GAMMA compares hop-wise messages with an inner product. Such a comparison is reliable only if all hop embeddings reside in a common and coherent feature space. Using independent, learnable projection matrices for each hop, as done in some multi-hop architectures [1], would map features to potentially unrelated latent spaces, rendering their direct comparison via inner products ill-defined and potentially undermining the gating process.

To address this, GAMMA employs a **shared linear transformation** for all hop distances. Initial node features $\mathbf{X}$ are first projected into a common $d_{\text{out}}$-dimensional space using a single weight

matrix $\mathbf{W} \in \mathbb{R}^{d_{in} \times d_{out}}$. This projected representation, $\mathbf{H}_{\text{proj}} = \mathbf{XW}$, then serves as the basis for generating representations for all considered hop distances:

$$\mathbf{H}^{(p)} = \mathbf{A}^p \mathbf{H}_{\text{proj}}, \quad p = 0, \dots, K, \tag{8}$$

where $\mathbf{A}^0 = \mathbf{I}_n$ (the identity matrix, representing the node's own projected features), and $\mathbf{H}^{(p)}$ denotes the unscaled features aggregated from the $p$-hop neighborhood. By utilizing a single $\mathbf{W}$, we ensure that all $\mathbf{H}^{(p)}$ are expressed in the same coordinate system, making their subsequent normalization and dot-product-based comparisons in the gating mechanism.

This weight-sharing strategy reduces the number of learnable parameters compared to models that use separate matrices $\mathbf{W}^{(p)}$ for each hop. Instead of $(K+1) \times d_{in} \times d_{out}$ parameters for transformations, GAMMA uses only $d_{in} \times d_{out}$. This reduction lessens the risk of overfitting, especially in semi-supervised settings with limited labeled data, and contributes to tighter generalization bounds. Also, fewer parameters naturally lead to lower memory requirements for storing the model. Furthermore, computations involving a single shared matrix can be more optimized.

While a single shared projection $\mathbf{W}$ ensures a consistent feature space, it might be too restrictive, lacking the flexibility to emphasize or de-emphasize certain feature dimensions differently for messages originating from various hop distances. To re-introduce a degree of hop-specific adaptability without fracturing the shared geometric space, GAMMA incorporates **learnable channel-wise scaling factors** $\boldsymbol{\gamma}_p \in \mathbb{R}^{d_{out}}$ for each hop $p$. After obtaining the propagated features $\mathbf{H}^{(p)}$ using the shared projection, each is modulated as follows:

$$\mathbf{H}^{(p)}_{\text{scaled}} = \mathbf{H}^{(p)} \odot \boldsymbol{\gamma}_p, \qquad p = 0, \dots, K, \tag{9}$$

where $\odot$ denotes element-wise multiplication. These scaled representations $\mathbf{H}^{(p)}_{\text{scaled}}$ are then $L_2$-normalized to produce $\hat{\mathbf{H}}^{(p)}$ for the gating procedure.

This scaling mechanism allows the model to learn the global importance of different feature channels at different hop distances. For instance, certain features might be highly informative when sourced from 1-hop neighbors but less so from 3-hop neighbors, or vice-versa. The scaling factors $\boldsymbol{\gamma}_p$ enable GAMMA to adapt to such patterns. Importantly, since scaling merely stretches or shrinks existing feature dimensions, it preserves the integrity of the common coordinate system, ensuring that the inner products used in the routing stage remain meaningful. The parameter overhead for these scaling factors is minimal, adding only $(K+1) \times d_{out}$ parameters, which is negligible compared to using full hop-specific transformation matrices.

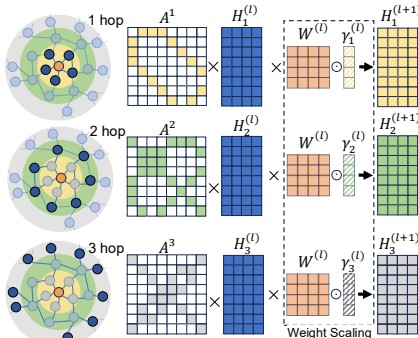

Figure 4: weight sharing.

However, it is crucial to distinguish the global and local heterophilic patterns in a graph. At the global level, each hop's features are scaled by the learnable channel-wise factors $\boldsymbol{\gamma}_p$, which adjust the overall importance of information from hop $p$ across the entire graph. In contrast, local adaptation to non-uniform heterophilic patterns relies on GAMMA's dynamic gating, where each node $i$, the coefficients $\alpha_{i,p}$ (softmax over logits $b_{i,p}$) are not fixed but recomputed iteratively during each forward pass. These logits are updated based on the dot-product agreement between node $i$'s current embedding and the normalized messages from each hop, yielding per-instance, activation-driven weights. By differentiating global scaling from node-specific gating, GAMMA can both prioritize broadly informative hop distances and adapt on-the-fly to unseen local heterophily patterns, without relying on static, learned weights for node-specific preferences.

## 6 Evaluation

**Configuration and setup.** We evaluate GAMMA on the semi-supervised node classification task across a diverse set of benchmark datasets, encompassing both homophilic and heterophilic graphs, with detailed results presented in Table 1. Our experiments demonstrate that GAMMA achieves competitive performance across these varied graph structures, showcasing its adaptability. All models were implemented in Python using PyTorch Geometric [7]. Experiments were conducted on a

desktop machine equipped with an NVIDIA RTX A2000 GPU (12GB VRAM) [27]. CUDA 12.8 facilitated GPU acceleration and NVIDIA Nsight Compute CLI [28] was employed for profiling and computational performance evaluations such as memory consumptions and runtime.

**Baselines.** To benchmark GAMMA, we compare it against a comprehensive suite of baselines representing key strategies in graph representation learning, especially for heterophilic contexts. This selection includes *Standard GNNs and Foundational Models* (MLP [12], GCN [14], GAT [34], SGC [36]), which serve to ground performance and highlight the challenges heterophily poses to conventional message-passing paradigms. We also incorporate *Spectral Graph Filters* (GCN-II [4], GPRGNN [5], BernNet [11]), chosen for their design leveraging graph signal processing to capture the mixed-frequency signals often characteristic of heterophilic structures. Furthermore, *Methods Utilizing High-Order Neighbors* (MixHop [1], H2GCN [40], GCN-JK [37], GAT-JK [37]) are included, as they aim to discover informative, potentially distant, same-class nodes by expanding the receptive field beyond immediate, possibly misleading, connections. Finally, *Discriminative Message Passing Techniques* (M2MGNN [18]) represent approaches that enable more selective and nuanced aggregation from diverse or signed neighborhood information, crucial when local context is not uniformly homophilous. This diverse selection provides a robust testbed for evaluating GAMMA's adaptability and effectiveness across the spectrum of graph structures.

**Training recipe.** For a fair and rigorous comparison of both predictive accuracy and computational demands, all models, including baselines, were configured with a consistent architecture: two GNN layers and a fixed hidden dimension size of 32. Hyperparameters for each model were optimized via a grid search over learning rates in $\{0.05, 0.01, 0.002\}$ and dropout rates $\in \{0.0, 0.5\}$. Across all datasets and splits, models were trained for a fixed 500 epochs. We report the test accuracy achieved using the hyperparameter configuration that yielded the highest performance on a held-out validation set for each dataset and model combination.

For robust evaluation, we adhere to standardized data splitting procedures. Specifically, for the heterophilic benchmark datasets, we utilize the 10 fixed train/validation/test splits [31]. These splits allocate 50% of nodes for training, 25% for validation, and 25% for testing, reflecting a common practice of using a substantial training set for non-homophilous graph learning evaluations [40, 29, 19, 38]. For the homophilic datasets, we also employ 10 distinct random splits, following the setup in [40], with 48% of nodes for training, 32% for validation, and 20% for testing. Each model is trained once per split, and we report the mean test performance metrics along with their standard deviations across these 10 splits, consistent with the evaluation methodology in [31].

Table 1: Node classification accuracy (%) on homophilic and heterophilic benchmarks.

| | Homophilic Graphs | | | Heterophilic Graphs | | | | | |
|---|---|---|---|---|---|---|---|---|---|
| | Cora | CiteSeer | PubMed | Texas | Wisconsin | Actor | Squirrel | Chameleon | Cornell |
| # Nodes | 2,708 | 3,327 | 19,717 | 183 | 251 | 7,600 | 5,201 | 2,277 | 183 |
| # Edges | 5,429 | 4,732 | 44,338 | 309 | 499 | 30,019 | 217,073 | 36,101 | 280 |
| Hom. ratio | 0.81 | 0.74 | 0.80 | 0.11 | 0.21 | 0.22 | 0.22 | 0.23 | 0.30 |
| MLP [12] | 75.64 ± 1.85 | 73.00 ± 1.48 | 86.55 ± 0.59 | 80.26 ± 3.58 | 84.71 ± 2.11 | 35.37 ± 0.87 | 32.16 ± 1.16 | 49.78 ± 2.22 | 77.89 ± 5.16 |
| GCN [14] | 87.27 ± 1.24 | 76.06 ± 1.13 | 87.38 ± 0.37 | 50.26 ± 3.21 | 45.49 ± 6.97 | 27.18 ± 0.98 | 25.13 ± 1.26 | 29.36 ± 2.03 | 47.89 ± 7.88 |
| SGC [36] | 36.91 ± 11.87 | 68.25 ± 4.80 | 76.19 ± 3.22 | 54.21 ± 4.88 | 44.51 ± 4.81 | 26.14 ± 1.20 | 20.66 ± 1.63 | 26.43 ± 1.84 | 43.16 ± 6.47 |
| GCN-II [4] | 83.96 ± 2.24 | 74.15 ± 1.81 | 88.86 ± 0.47 | 76.32 ± 8.89 | 81.96 ± 2.45 | 35.18 ± 1.34 | 31.86 ± 1.58 | 49.61 ± 1.48 | 65.26 ± 7.96 |
| GCN-JK [37] | 83.13 ± 1.13 | 69.61 ± 1.64 | 86.94 ± 0.63 | 39.47 ± 6.34 | 42.16 ± 6.52 | 25.94 ± 0.99 | 22.56 ± 1.12 | 29.17 ± 2.30 | 50.00 ± 8.57 |
| GAT-JK [37] | 84.33 ± 1.05 | 71.74 ± 1.95 | 85.56 ± 0.57 | 41.58 ± 5.24 | 46.67 ± 8.03 | 26.68 ± 0.82 | 24.67 ± 1.25 | 36.58 ± 2.53 | 50.53 ± 8.55 |
| H2GCN [40] | 87.75 ± 0.88 | 75.95 ± 1.24 | 89.44 ± 0.46 | 86.58 ± 4.47 | 85.88 ± 4.19 | 35.33 ± 1.06 | 38.36 ± 1.59 | 59.69 ± 1.43 | 78.16 ± 6.97 |
| GAT [34] | 85.34 ± 0.88 | 73.19 ± 1.79 | 87.13 ± 0.54 | 43.68 ± 5.79 | 44.12 ± 6.34 | 26.36 ± 1.11 | 25.02 ± 1.36 | 34.87 ± 2.79 | 46.05 ± 8.43 |
| MixHop [1] | 85.93 ± 1.09 | 74.99 ± 1.81 | 89.06 ± 0.48 | 84.21 ± 6.6 | 85.69 ± 3.93 | 32.70 ± 1.23 | 34.61 ± 2.47 | 52.54 ± 2.25 | 74.47 ± 4.71 |
| BernNet [11] | 87.33 ± 1.20 | 76.42 ± 1.57 | 86.84 ± 0.52 | 79.47 ± 3.07 | 81.57 ± 4.31 | 33.78 ± 1.72 | 33.43 ± 1.43 | 49.14 ± 1.19 | 73.95 ± 7.58 |
| GPRGNN [5] | 87.72 ± 0.88 | 75.23 ± 1.69 | 88.26 ± 0.49 | 61.58 ± 8.66 | 78.24 ± 3.09 | 33.62 ± 0.92 | 28.29 ± 1.26 | 34.43 ± 2.14 | 62.37 ± 5.77 |
| M2MGNN [18] | 83.65 ± 1.88 | 70.70 ± 2.47 | 88.62 ± 0.66 | 84.47 ± 4.77 | 87.06 ± 2.00 | 33.89 ± 1.14 | 33.89 ± 4.01 | 58.14 ± 2.14 | 76.84 ± 4.82 |
| **GAMMA** | **87.42 ± 1.01** | **75.49 ± 1.67** | **89.62 ± 0.43** | **87.37 ± 3.68** | **86.27 ± 4.21** | **35.59 ± 1.29** | **36.20 ± 1.01** | **51.16 ± 2.22** | **78.68 ± 3.42** |

*Note:* Values are colored according to their ranking: **best** in **green**, second-best in blue, and third-best in red.

**Performance on Homophilic Benchmarks.** On standard homophilic benchmarks, GAMMA demonstrates consistently strong and competitive performance, validating its robustness even when the homophily assumption largely holds. As shown in Table 1, on PubMed, GAMMA achieves the best accuracy among all compared methods with $89.62 \pm 0.43\%$. For Cora, GAMMA obtains $87.42 \pm 1.01\%$, performing comparably to GPRGNN ($87.72 \pm 0.88\%$) and H2GCN ($87.75 \pm 0.88\%$). These results indicate that GAMMA's adaptive multi-hop gating mechanism does not impede its ability to leverage strong homophilous signals, maintaining high efficacy on datasets where traditional GNNs typically excel.

**Performance on Heterophilic Benchmarks.** GAMMA showcases its strength and adaptability on the more challenging heterophilic datasets. Across the six heterophilic benchmarks, GAMMA consistently ranks among the top-performing models, outperforming many specialized heterophilic

GNNs and standard GNNs, which often struggle in these settings. Specifically, GAMMA achieves the best performance on three of the six heterophilic datasets. On Texas, it surpasses H2GCN and M2MGNN. On the Actor dataset, GAMMA leads with $35.59 \pm 1.29\%$, ahead of MLP and H2GCN and on Cornell, GAMMA again achieves the top accuracy of $78.68 \pm 3.42\%$, outperforming H2GCN and MLP. On the remaining heterophilic datasets, GAMMA also demonstrates robust and highly competitive results. For Wisconsin GAMMA appears as the second-best, closely following M2MGNN ($87.06 \pm 2.00\%$) and outperforming several other specialized methods like H2GCN and MixHop.

These strong performances across datasets with varying low homophily ratios (from 0.11 for Texas to 0.30 for Cornell). The results suggest that GAMMA's node-specific adaptive gating of multi-hop information is particularly beneficial for navigating the complex neighborhood structures characteristic of heterophilic graphs, allowing it to match or exceed the performance of methods that rely on fixed higher-order aggregations (MixHop) or more complex heterophily-specific designs (BernNet and M2MGNN). The competitive results on both homophilic and heterophilic graphs strongly support GAMMA's design as a homophily-agnostic GNN.

**Performance on Larger Heterophilic Benchmarks.** To address scalability concerns and demonstrate GAMMA's effectiveness on larger graphs, we evaluate our method on seven large heterophilic benchmarks. Table 2 presents results on datasets ranging from tens of thousands to millions of nodes. Notably, many state-of-the-art methods encounter Out-Of-Memory (OOM) errors even on an NVIDIA A100 GPU with 80GB VRAM, particularly on ogbn-products (2.4M nodes, 61M edges), due to architectural choices such as feature concatenation (H2GCN, MixHop, JK-Nets) or complex per-edge operations (M2MGNN) that create prohibitively large intermediate tensors. GAMMA addresses these limitations through its weight-sharing strategy and dynamic gating mechanism, enabling it to scale effectively while maintaining competitive accuracy. GAMMA achieves state-of-the-art or highly competitive performance on 6 out of 7 large-scale datasets, demonstrating that efficient architectural design does not compromise effectiveness. Across the complete benchmark suite of 16 datasets spanning both small and large-scale graphs, GAMMA achieves the **best overall performance** with an average rank of 2.06, compared to H2GCN (3.71) and MixHop (3.87), confirming its effectiveness and scalability for heterophilic graph learning.

Table 2: Node classification accuracy (%) on large heterophilic benchmarks. OOM indicates Out-Of-Memory errors.

| | ogbn-arxiv | roman-empire | ogbn-products | minesweeper | amazon-ratings | amazon-photos | tolokers |
|---|---|---|---|---|---|---|---|
| # Nodes | 169,343 | 22,662 | 2,449,029 | 10,000 | 24,492 | 7,650 | 11,758 |
| # Edges | 1,166,243 | 32,927 | 61,859,140 | 39,402 | 186,100 | 238,162 | 519,000 |
| MLP [12] | $47.34 \pm 0.09$ | $65.37 \pm 0.58$ | $43.20 \pm 0.01$ | $79.48 \pm 0.21$ | $38.35 \pm 0.20$ | $48.73 \pm 1.80$ | $78.16 \pm 0.00$ |
| GCN [14] | $64.91 \pm 0.26$ | $54.16 \pm 0.28$ | $71.31 \pm 0.16$ | $80.31 \pm 0.24$ | $42.12 \pm 0.29$ | $52.14 \pm 1.50$ | $78.76 \pm 0.17$ |
| SGC [36] | $63.47 \pm 0.16$ | $61.20 \pm 0.65$ | OOM | $80.05 \pm 0.03$ | $41.61 \pm 0.72$ | $57.97 \pm 0.52$ | $78.56 \pm 0.19$ |
| GCN-II [4] | $61.96 \pm 0.42$ | $64.96 \pm 0.38$ | $61.08 \pm 0.18$ | $82.80 \pm 0.35$ | $47.20 \pm 0.55$ | $94.10 \pm 0.25$ | $79.80 \pm 0.30$ |
| GCN-JK [37] | $70.19 \pm 0.11$ | $62.18 \pm 0.91$ | $69.46 \pm 0.33$ | $85.15 \pm 0.42$ | $48.77 \pm 0.37$ | $95.28 \pm 0.05$ | $81.23 \pm 0.26$ |
| GAT-JK [37] | $70.75 \pm 0.19$ | $70.00 \pm 0.70$ | OOM | $85.65 \pm 0.34$ | $50.84 \pm 0.52$ | $95.71 \pm 0.11$ | $81.31 \pm 0.48$ |
| H2GCN [40] | OOM | $75.89 \pm 0.55$ | OOM | $83.31 \pm 0.05$ | $42.90 \pm 0.38$ | $84.95 \pm 0.38$ | $79.39 \pm 0.42$ |
| GAT [34] | $70.01 \pm 0.09$ | $74.50 \pm 0.45$ | OOM | $81.52 \pm 0.26$ | $48.89 \pm 0.61$ | $95.56 \pm 0.15$ | $78.63 \pm 0.06$ |
| MixHop [1] | $70.27 \pm 0.14$ | $82.20 \pm 0.31$ | OOM | $84.88 \pm 0.45$ | $50.98 \pm 0.30$ | $96.34 \pm 0.30$ | $81.03 \pm 0.70$ |
| BernNet [11] | $63.15 \pm 0.13$ | $66.27 \pm 0.35$ | OOM | $80.00 \pm 0.07$ | $40.90 \pm 0.09$ | $62.34 \pm 7.02$ | $78.16 \pm 0.00$ |
| GPRGNN [5] | $67.66 \pm 0.07$ | $69.60 \pm 0.40$ | $73.26 \pm 0.65$ | $80.36 \pm 0.19$ | $43.44 \pm 0.13$ | $79.54 \pm 0.39$ | $78.36 \pm 0.10$ |
| M2MGNN [18] | $72.52 \pm 0.10$ | $80.81 \pm 0.70$ | OOM | $85.83 \pm 0.25$ | $50.57 \pm 0.29$ | $95.73 \pm 0.25$ | $80.66 \pm 0.47$ |
| **GAMMA** | $71.81 \pm 0.19$ | $81.37 \pm 0.25$ | $72.61 \pm 0.04$ | $87.58 \pm 0.25$ | $49.07 \pm 0.65$ | $96.28 \pm 0.11$ | $82.59 \pm 0.41$ |

*Note:* Values are colored according to their ranking: **best** in green, second-best in blue, and third-best in red.

**Computational efficiency.** Beyond predictive accuracy, the practical utility of a GNN model heavily depends on its computational efficiency, encompassing both execution latency and memory footprint. We benchmarked GAMMA against various baselines, measuring forward pass time, backward pass time, and total GPU memory usage on the Flickr dataset. The results, shown on the right-hand side of Fig. 5, underscore GAMMA's efficiency profile. GAMMA operates with a total execution time (backward + forward pass) of 23.17 ms and a memory footprint of 480.60 MB. This latency is competitive with the simplest architectures such as GCN, which records 15.24 ms, and GPRGNN at 18.75 ms. Critically, GAMMA demonstrates substantial speedups over several complex models: it runs approximately $20\times$ faster than H2GCN, over $7.4\times$ faster than M2MGNN, and nearly $5\times$ faster than MixHop. In terms of memory, GAMMA's 480.60 MB usage is significantly efficient, consuming approximately $12.1\times$ less memory than M2MGNN, over $6.1\times$ less than GATJK, and about $4.1\times$ less than H2GCN (1993.90 MB).

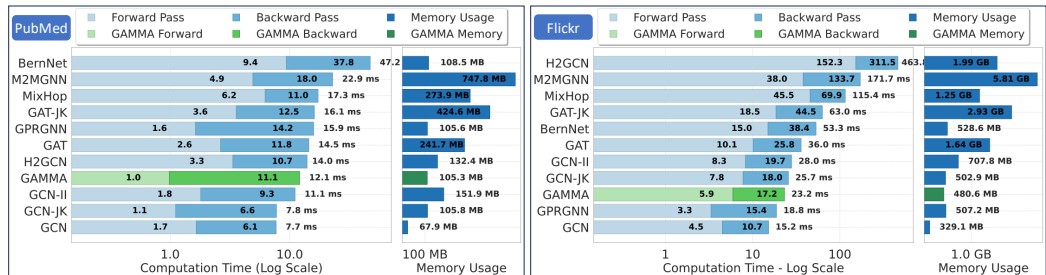

Figure 5: GAMMA efficiency comparison across various GNNs on Flickr and PubMed datasets.

GAMMA's computational efficiency, despite its capacity for adaptive multi-hop information processing, stems from several core architectural decisions. Firstly, the shared linear transformation applied once to input features ($\mathbf{XW}$) ensures parameter efficiency for the primary feature projection, as discussed in Section 5. Subsequent multi-hop propagations ($\mathbf{A}^p(\mathbf{XW})$) operate on these already transformed, fixed-dimension features. This is a key distinction from models like MixHop, which concatenates outputs from different powers of the adjacency matrix, leading to a larger feature dimension for subsequent layers and higher parameter counts as distinct transformations were applied per hop. Secondly, GAMMA **avoids explicit feature concatenation for integrating multi-hop information**. Many multi-hop architectures, such as H2GCN (which concatenates features from 1-hop and 2-hop paths, and across iterations) and MixHop, expand the feature dimensionality before a which in turn requires larger memory transactions. This increase in output feature dimension directly results in larger intermediate tensors and more expensive matrix operations. In contrast, GAMMA's gating mechanism computes a weighted sum of hop-specific embeddings and merges them into a unified output vector, each preserving the target output feature dimension, thus avoiding heavy memory and computational requirements.

The lightweight iterative gating mechanism further contributes to GAMMA's efficiency. Each iteration of the gating mechanism involves GPU friendly and highly parallel operations like softmax, element-wise products, and sums primarily on tensors of shape [#Nodes, #hops], with a low number of iterations (empirically 1 to 3). These operations are computationally less demanding than, for example, the per-edge multi-layer perceptron and segmented softmax operations within M2MGNN's layers, or the per-edge attention coefficient calculations in GAT and its variants. While GAMMA computes embeddings from multiple hops using standard message passing, the subsequent aggregation via gating is more efficient than learning complex, global filter coefficients for each feature as in some spectral methods like BernNet, which involves K propagation steps for its polynomial filters, or handling large feature dimensions. Consequently, GAMMA offers a compelling balance between accuracy across diverse graphs and the practical computational efficiency required for real-world heterophilic datasets.

# 7 Conclusion

In this paper, we introduced GAMMA, a homophily-agnostic GNN architecture that effectively handles the non-uniform heterophilic patterns prevalent in real-world graphs. By employing a lightweight iterative gating mechanism and an efficient weight-sharing scheme, GAMMA adaptively leverages multi-hop neighborhood information based on node-specific structural patterns. Our experiments across diverse benchmarks demonstrate that GAMMA matches or exceeds state-of-the-art accuracy while achieving up to $20\times$ faster inference and substantially lower memory consumption, making heterophilic graph learning practical on commodity hardware. Despite these advances, limitations remain for future exploration. GAMMA's current fixed routing iteration count may be suboptimal for nodes with varying neighborhood complexities, and extremely large graphs may require further optimizations to the gating mechanism. Additionally, while our dot-product agreement measure works well in practice, more sophisticated routing strategies might better capture particularly complex heterophilic relationships. Nevertheless, GAMMA challenges the assumption that handling heterophily necessitates complex architectures, providing a practical solution that balances adaptability, performance, and computational efficiency for diverse graph learning tasks.

## Acknowledgements

This work was partially supported by DARPA under Cooperative Agreement FA8750-23-2-0501, the U.S. Department of Energy under grant DE-SC0024428, and the U.S. National Science Foundation under CAREER Award CCF-2441973.

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

# Appendix

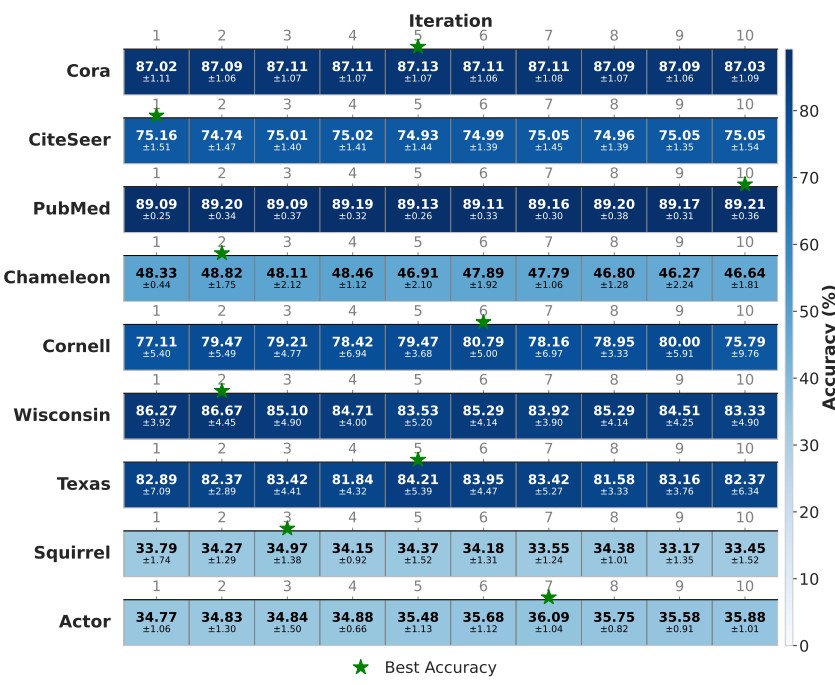

Figure 6: Iterative gating mechanism performance with different number of iterations

## A    Ablation Study on Number of Routing Iterations in the Gating Mechanism

To understand the impact of the iterative refinement process within the GAMMA layer, we conduct an ablation study on the number of iterations (R in Algorithm 1, line 10). This parameter controls how many times each node refines its assessment of the relevance of aggregated information from its $k$-hop neighbors by updating the gating logits $\mathbf{b}$ based on the agreement between hop-specific embeddings and the evolving node representation $\mathbf{v}$. We vary number of iterations from 1 to 10 and report the mean accuracy and standard deviation over 10 splits for each dataset. The detailed results of this study are presented in Fig. 6, allowing us to observe how the performance of GAMMA changes as nodes are given more steps to dynamically determine the optimal mix of multi-hop information.

On the homophilic datasets (Cora, CiteSeer, and PubMed), the performance of GAMMA demonstrates a general trend of either peaking or stabilizing with a relatively small number of routing iterations. For instance, on Cora, accuracy improves from $87.02$ at 1 iteration to a peak of $87.13$ at 5 iterations, after which it sees a marginal decline. PubMed also shows robustness, with performance quickly reaching near-optimal levels (e.g., $89.20$ at 2 iterations and $89.21$ at 10 iterations), indicating that a few iterations are sufficient to establish effective routing coefficients. CiteSeer exhibits slight fluctuations but generally performs well across different iteration counts, achieving $75.16$ at 1 iteration and $75.05$ at 10 iterations. This suggests that for graphs where homophily is prevalent, the initial agreement scores are often strong, and the iterative process rapidly converges to an effective combination of hop information, primarily leveraging local neighborhood signals. While additional iterations do not significantly degrade performance, they offer diminishing returns, implying that the model quickly identifies the most relevant, often nearby, hops.

In contrast, the behavior on heterophilic datasets is more varied, underscoring the diverse nature of information distribution in such graphs. Some heterophilic datasets like Chameleon and Wisconsin achieve their peak performance with very few iterations (Chameleon: $48.82$ at 2 iterations; Wisconsin: $86.67$ at 2 iterations), with performance tending to decrease with more iterations. This pattern suggests that for these specific graphs, the initial routing based on agreement quickly identifies the most discriminative hop information, and further iterations might risk incorporating noisy or less relevant signals from more distant or dissimilar neighborhoods. However, other heterophilic datasets such as Actor and Cornell benefit from a greater number of iterations. Actor's accuracy gradually increases, peaking at $36.09$ with 7 iterations, while Cornell peaks at $80.79$ with 6 iterations. This

indicates that for these graphs, the iterative refinement process is crucial for dynamically adjusting the weights of different k-hop neighborhoods, allowing the model to progressively focus on more informative, potentially non-obvious, hop distances that are critical for capturing heterophilic patterns. The optimal number of iterations thus appears to be dataset-dependent, reflecting the unique structural characteristics and the complexity of heterophily within each graph.

## B Impact of Dynamic Gating Mechanism Through Empirical Study

To demonstrate the contribution of GAMMA's node-specific dynamic gating mechanism, we conduct an ablation study comparing two configurations: (1) **Without Gating:** uniform hop weights for all nodes (i.e., $\alpha_{i,p}$ equal across all hops $p$ for each node $i$), while retaining the learnable channel-wise scaling factors $\boldsymbol{\gamma}_p$ that operate globally, and (2) **With Gating:** GAMMA's full iterative gating mechanism that adaptively computes node-specific hop coefficients $\alpha_{i,p}$ based on agreement scores.

Table 3 presents results on six heterophilic datasets aggregating information from different hop combinations. The results demonstrate that dynamic gating consistently outperforms uniform hop weighting across all datasets, with improvements ranging from 2-8% depending on the dataset. These gains stem from GAMMA's ability to adaptively capture node-specific multi-hop homophily patterns.

Table 3: Ablation study: Impact of gating mechanism on node classification accuracy (%).

| Hop Config. | Cornell | | | Texas | | | Wisconsin | | | Chameleon | | | Actor | | | Squirrel | | |
|---|---|---|---|---|---|---|---|---|---|---|---|---|---|---|---|---|---|---|
| | W/o | With | $\Delta$ | W/o | With | $\Delta$ | W/o | With | $\Delta$ | W/o | With | $\Delta$ | W/o | With | $\Delta$ | W/o | With | $\Delta$ |
| 1-2 hop | 75.31 | **78.68** | +3.37 | 83.25 | **87.37** | +4.12 | 81.40 | **86.27** | +4.87 | 44.79 | **51.16** | +6.37 | 30.72 | **35.59** | +4.87 | 28.33 | **36.20** | +7.87 |
| 1-3 hop | 73.51 | **77.11** | +3.60 | 82.57 | **86.47** | +3.90 | 79.02 | **84.12** | +5.10 | 42.29 | **51.14** | +8.85 | 30.29 | **36.06** | +5.77 | 29.28 | **35.43** | +6.15 |
| 1-4 hop | 75.99 | **79.21** | +3.22 | 81.86 | **86.21** | +4.35 | 80.06 | **84.71** | +4.65 | 42.40 | **50.87** | +8.47 | 27.25 | **34.90** | +7.65 | 27.42 | **35.52** | +8.10 |

*Note:* "W/o" = without dynamic gating (uniform hop weights with global scaling $\boldsymbol{\gamma}_p$). "With" = GAMMA's full dynamic gating. $\Delta$ shows improvement from adding dynamic gating.

**Why Dynamic Gating Matters.** As demonstrated in Section 2, heterophily varies both across nodes and hop distances. For instance, in the Chameleon dataset, we observe dramatically different multi-hop homophily patterns across nodes. Chameleon exhibits only 20.7% pattern consistency, meaning just 20.7% of nodes share similar multi-hop homophily trajectories, while most show diverse, complex patterns. This explains why Chameleon sees the largest performance gains (6-7%) from dynamic gating. In contrast, Cornell displays 65.3% pattern dominance, where the majority of nodes share similar multi-hop homophily, resulting in smaller but still meaningful improvements (2-3%).

Unlike methods with global coefficients (e.g., GPR-GNN), GAMMA's dynamic gating operates at both training and inference time. During inference, the routing loop recomputes gating coefficients $\alpha_{i,p}$ from scratch for each node, enabling GAMMA to adapt to unseen heterophily patterns in the local neighborhood without any gradient updates. This architectural design separates global pattern learning (via $\boldsymbol{\gamma}_p$ learned during training) from local pattern identification (via dynamic $\alpha_{i,p}$ computed during each forward pass).

## C GAMMA Pseudocode and Implementation Details

**Algorithm Explanation and GPU-Efficient Computation.** Algorithm 1 provides a concise overview of the forward pass in a GAMMA layer. In Lines 1–4, we perform a single shared linear transformation on the input $\mathbf{X}$ and then apply $K$-hop propagations via standard sparse-dense matrix multiplication (spmm), which is highly optimized for modern GPUs. Unlike multi-hop methods relying on explicit feature concatenation, GAMMA keeps each hop representation $\mathbf{H}^{(p)}$ in the same output dimension $d_{\text{out}}$ and scales it channel-wise in Lines 6–7. This uniform dimensionality avoids blowing up tensor shapes in memory and reduces the number of large intermediate operations typically seen in concatenation-based schemes (e.g., MixHop).

Within each node's local routing loop (Lines 10–17), the dot products in Line 12 act as a similarity measure between a node's evolving summary vector $\mathbf{v}_i$ and each normalized hop embedding $\hat{\mathbf{H}}_i^{(p)}$. These dot products are very lightweight on a GPU, as they operate on small vectors with shape $[d_{\text{out}}]$ and then update a softmax distribution of size $K+1$. Importantly, these per-node operations can be mapped to GPU kernels that rely on fully on-chip memory, eliminating large-scale memory

---

**Algorithm 1** GAMMA Layer Forward Pass

---

**Require:** Node features $\mathbf{X} \in \mathbb{R}^{n \times d_{\text{in}}}$, adjacency matrix $\mathbf{A} \in \mathbb{R}^{n \times n}$, powers $K$, routing iterations $R$, scaling
  parameters $\boldsymbol{\gamma}_0, \ldots, \boldsymbol{\gamma}_K$, weight matrix $\mathbf{W} \in \mathbb{R}^{d_{\text{in}} \times d_{\text{out}}}$, bias $\mathbf{b} \in \mathbb{R}^{d_{\text{out}}}$
**Ensure:** Node representations $\mathbf{V} \in \mathbb{R}^{n \times d_{\text{out}}}$

1: $\mathbf{H} \leftarrow \mathbf{XW}$      ▷ Shared linear transformation
2: $\mathbf{H}^{(0)} \leftarrow \mathbf{H}$      ▷ 0-hop representation is the node's own features
3: **for** $p = 1$ to $K$ **do**
4:     $\mathbf{H}^{(p)} \leftarrow \mathbf{A}\mathbf{H}^{(p-1)}$      ▷ $p$-hop propagation via sparse-dense matmul
5: **for** $p = 0$ to $K$ **do**
6:     $\mathbf{H}^{(p)} \leftarrow \mathbf{H}^{(p)} \odot \boldsymbol{\gamma}_p$      ▷ Apply hop-specific scaling
7:     $\hat{\mathbf{H}}^{(p)} \leftarrow \text{normalize}\big(\mathbf{H}^{(p)}\big)$      ▷ Node-wise $L_2$ normalization
8: **for** each node $i \in V$ **do**
9:     $\mathbf{b}_i \leftarrow \mathbf{0} \in \mathbb{R}^{K+1}$      ▷ Initialize gating logits
10:     **for** $t = 1$ to $R$ **do**
11:        $\boldsymbol{\alpha}_i \leftarrow \text{softmax}\big(\mathbf{b}_i\big)$      ▷ Compute gating coefficients
12:        $\mathbf{s}_i \leftarrow \sum_{p=0}^{K} \alpha_{i,p} \hat{\mathbf{H}}_i^{(p)}$      ▷ Weighted sum of hop representations
13:        $\mathbf{v}_i \leftarrow \text{squash}\big(\mathbf{s}_i\big)$      ▷ Apply "squash" for normalization
14:        **for** $p = 0$ to $K$ **do**
15:           $b_{i,p} \leftarrow b_{i,p} + \hat{\mathbf{H}}_i^{(p)} \cdot \mathbf{v}_i$      ▷ Logit update via dot product (agreement)
16:     $\mathbf{v}_i \leftarrow \mathbf{v}_i + \mathbf{b}$      ▷ Optional bias for final representation
17: **return** $\mathbf{V} = \{\mathbf{v}_i\}_{i \in V}$      ▷ Output node representations

---

transactions. By contrast, concatenation-based methods generate large expanded embeddings (often dimension $(K+1)\,d_{\text{out}}$), which can lead to memory-bound kernels and slower performance.

Note that in Lines 1–7, we first apply the shared linear transformation $\mathbf{H} = \mathbf{XW}$, then iteratively compute $p$-hop embeddings by multiplying from right to left:

$$\mathbf{H}^{(p)} = \mathbf{A}\,\mathbf{H}^{(p-1)} \iff \mathbf{H}^{(p)} = \mathbf{A}^p\,(\mathbf{XW}). \tag{10}$$

This step-by-step approach avoids explicitly forming $\mathbf{A}^p$ at once and reduces repeated large-scale multiplications. After channel-wise scaling and normalization (Lines 6–7), each node's per-hop features remain in dimension $d_{\text{out}}$, rather than expanding to $(K+1)\,d_{\text{out}}$ as in feature concatenation.

Within the routing loop (Lines 10–16), the dot products in Line 12 act as a localized "agreement" between a node's evolving representation and each hop's normalized embedding. These computations are inexpensive on GPUs, as each node handles only small $d_{\text{out}}$-dimensional vectors. Crucially, by maintaining a single vector $\mathbf{v}_i$ per node and updating logits $\mathbf{b}_i$ in place, we avoid memory-bound concatenations. Consequently, relevant hop data remain on-chip, allowing the algorithm to leverage GPU-friendly sparse-dense kernels efficiently. This design leads to less overhead and lower peak memory usage, as observed empirically in Section 6.

# D   Related Work

The foundational success of early Graph Neural Networks (GNNs), such as Graph Convolutional Networks (GCN) [14] and Graph Attention Networks (GAT) [34], has been predominantly demonstrated on homophilic graphs, where connected nodes tend to share similar features and labels. Their inherent message-passing mechanisms act as low-pass filters, effectively smoothing representations within local neighborhoods, which is beneficial under homophily [25]. However, this very property leads to suboptimal performance on heterophilic graphs, where adjacent nodes often belong to different classes [40, 39]. This discrepancy has catalyzed a significant body of research focused on developing GNNs tailored for, or robust to, heterophily. These approaches can be broadly categorized.

**Standard GNNs and Foundational Models.**   Beyond GCN and GAT, other foundational models highlight the challenges of heterophily. Simplified Graph Convolutions (SGC) [36] streamline GCN by removing non-linearities and collapsing multiple layers into a single linear transformation. While exceptionally efficient due to its pre-computation of K-hop features, SGC's aggressive smoothing makes it highly susceptible to performance degradation on heterophilic graphs where dissimilar neighbors are prevalent. Interestingly, Multilayer Perceptrons (MLP) [12], which entirely disregard

graph structure and operate solely on node features, sometimes surprisingly outperform standard GNNs on highly heterophilic datasets (see Table 1). This phenomenon underscores the detrimental effect of inappropriate message passing in heterophilic settings, where aggregating from misleading local connections can be worse than ignoring them altogether. While these models offer high computational efficiency, their inherent homophily bias or complete disregard for structure makes them ill-suited for effectively leveraging the complex relational information in heterophilic graphs.

**Extending Receptive Fields: Multi-hop and Jumping Knowledge Architectures.** Recognizing that informative similar-class nodes in heterophilic graphs might reside beyond immediate neighborhoods, several methods explicitly incorporate multi-hop information. MixHop [1] enables GNN layers to learn from linear combinations of feature representations from neighbors at various hop distances (k=0,1,2,...) by repeatedly multiplying features with powers of the (normalized) adjacency matrix and concatenating their transformed versions. This allows the model to directly access information from different neighborhood ranges. While MixHop demonstrates the utility of multi-hop features for heterophily, it relies on feature concatenation. This strategy leads to a linear increase in feature dimensionality and parameters with the number of hops considered, significantly increasing computational and memory costs, as shown in our analysis (Figure 2). Furthermore, the mixing of hop information in MixHop is typically global, not adapted per node, limiting its ability to capture node-specific heterophily patterns.

Jumping Knowledge Networks (JK-Nets) [37], applicable to models like GCN-JK and GAT-JK, offer a more flexible approach to leverage multi-scale information. JK-Nets aggregate representations from all previous layers (effectively different neighborhood radii) for each node at the final layer, using mechanisms like concatenation, max-pooling, or LSTM-attention to combine them. This allows nodes to adaptively select the relevant neighborhood range. However, the selection mechanism in JK-Nets is primarily focused on the depth of the GNN, and it allows differential influence of k-hop information. The combination strategy, while adaptive, might still mix features from different semantic spaces if distinct transformations are used per layer before aggregation, potentially leading to information redundancy or dilution.

H2GCN [40] specifically targets heterophily by incorporating three key design principles: (i) separating the representation of the ego-node from its aggregated neighbors to prevent feature dilution, (ii) explicitly including 2-hop neighbors to reach potentially more similar nodes, and (iii) combining intermediate representations from different layers to preserve a spectrum of neighborhood information. H2GCN's strength lies in its tailored design for heterophily, showing strong empirical results. However, its architecture is somewhat manually crafted with a fixed inclusion of specific hop distances (1-hop and 2-hop), and the concatenation of features from these hops and across layers leads to substantial increases in memory usage and computational load. Our analysis (Figure 2 in the main paper) shows H2GCN consuming over 6× more GPU memory and incurring a 33× increase in execution time compared to vanilla GCN, making it computationally prohibitive for many real-world applications.

These methods underscore the importance of higher-order neighborhoods for heterophily. However, they often involve either fixed schemes for incorporating multi-hop data or strategies that can significantly increase model parameters and computational demands.

**Spectral Approaches and Graph Filtering.** Another line of work draws inspiration from graph signal processing, designing spectral filters to capture varying frequencies of graph signals, which is pertinent for heterophily where both high and low-frequency information can be crucial. GPR-GNN [5] learns a set of weights for a linear combination of propagated feature matrices, where each matrix corresponds to a different power of a generalized propagation matrix (e.g., normalized adjacency matrix). This allows the model to learn an optimal polynomial filter for the task and graph at hand. While GPR-GNN adaptively learns the filter coefficients, these coefficients are global for the entire graph, meaning the spectral response is not tailored to individual nodes or local structures. This global learning can limit its effectiveness on graphs exhibiting diverse local homophily/heterophily patterns.

BernNet [11] employs Bernstein polynomial filters, which offer a more stable way to approximate desired spectral filters. By adjusting the coefficients of the Bernstein basis polynomials, BernNet can shape the filter response to capture complex patterns. Similar to GPR-GNN, BernNet learns global filter coefficients. Its K-th order filter necessitates K separate sparse-dense multiplications,

each incurring its own kernel overhead and intermediate storage, making it computationally intensive despite its expressive power (Figure 2).

GCNII [4] addresses the oversmoothing problem in deep GCNs by incorporating an initial residual connection and an identity mapping in each layer. This allows GCNII to build deeper models that can potentially capture longer-range dependencies without performance degradation. While GCNII improves information propagation in deep GNNs, its core message passing still relies on the standard GCN aggregation, which is primarily homophily-oriented and doesn't specifically address heterophilic aggregation challenges beyond enabling greater depth. Overall, Spectral methods provide a principled way to combine multi-hop information. However, their reliance on globally learned filters often means they may not offer the fine-grained, node-level adaptivity required for graphs with diverse local homophily/heterophily patterns.

**Addressing Heterophily via Signed or Adaptive Message Aggregation.** Some approaches aim to differentiate how messages from neighbors are aggregated, sometimes allowing for "negative" influences or class-aware aggregation. The concept of Signed Message Passing (SMP) [3, 38] allows GNNs to assign negative weights in message passing, enabling the model to distinguish between features from same-class and different-class neighbors. This allows the model to attract similar neighbors and repel dissimilar ones. However, Liang et al. [18] identified that naive SMP can suffer from undesirable multi-hop effects (e.g., when combining individually appropriate one-hop propagation matrices) and vulnerability to oversmoothing in multi-class settings as mean embeddings of different classes converge exponentially.

M2M-GNN (Multiset-to-Multiset GNN) [18] was proposed to overcome these limitations. It replaces the standard multiset-to-element aggregation with a multiset-to-multiset approach. An attention mechanism assigns neighbors into different "chunks" (ideally corresponding to different classes), and then class-specific information is pooled from these chunks. This helps maintain the segregation of information from potentially different classes. While M2M-GNN effectively segregates information, it typically uses a fixed combination strategy (e.g., concatenation) for the resulting chunk vectors, which may not optimally integrate the segregated information. Moreover, M2M-GNN can be computationally intensive due to its per-edge MLP and segmented softmax operations, which cannot be cast as one large matrix-matrix multiply, leading to numerous small kernel launches and frequent intermediate result synchronization (Figure 2).

CPGNN (Class-Prototype Graph Neural Network) [41] tackles heterophily by first estimating initial class probabilities for each node. It then propagates these "soft labels" or priors across the graph, weighted by a trainable compatibility matrix that encodes the likelihood of different classes being adjacent. This allows message passing to be guided by estimated class relationships. CPGNN offers a novel way to incorporate class semantics into propagation. However, its propagation relies on these estimated class compatibilities which are learned globally, and the message passing itself is synchronous, potentially missing finer-grained local differences in how nodes should value their neighbors from different hop distances.

Further, CO-GNN (Cooperative Graph Neural Networks) [8] represents a recent approach to node-specific message passing through a cooperative learning framework. CO-GNN employs dual networks, an action network that generates node-specific aggregation weights and an environment network that provides context, enabling each node to learn individualized message passing strategies. While this design achieves node-specific adaptation, it comes at substantial computational cost. The dual-network architecture requires separate linear transformations and graph convolutions for both networks at each layer, effectively doubling the computational load. Our profiling shows CO-GNN incurs 4.4× backward pass overhead compared to GCN (forward: 1.820 ms, backward: 13.431 ms on Cora).

**Redefining Neighborhoods and Global Methods.** A distinct set of approaches rethinks the notion of a "neighbor" or allows for global information exchange, moving beyond fixed topological neighborhoods. Geom-GCN [29] aggregates information from nodes in continuous space, constructing neighborhoods based on structural similarity (e.g., nodes with similar local graph structures or positions in latent space) rather than direct edge connections. This allows aggregation from distant but structurally similar nodes.

GloGNN [17] and its variants take a more direct global approach. They might learn a dense affinity matrix or use attention mechanisms that allow each node to potentially aggregate information from

all other nodes in the graph, irrespective of direct connections. This enables the model to capture very long-range dependencies and global patterns.

While these methods can be powerful, especially Geom-GCN and GloGNN, they often come with significant computational costs (e.g., computing pairwise similarities or attention over all nodes with $O(n^2)$ complexity) and high memory requirements, making them challenging to scale to large graphs. They also risk incorporating noise by connecting nodes that are globally related but contextually irrelevant for a specific prediction.

**Positioning of GAMMA.** The existing landscape of GNNs for heterophily reveals a critical trade-off: methods that are highly adaptive or capture global context are often computationally demanding, while more efficient methods may lack the necessary node-specific adaptivity in how they leverage multi-hop information. Many multi-hop GNNs use fixed schemes, globally learned weights for hop combination, or feature concatenation strategies that escalate computational and memory demands. GAMMA is designed to bridge this gap. It explicitly computes multi-hop representations but employs a lightweight, iterative gating mechanism at each node to dynamically assess and combine information from these different hop distances. Crucially, by using a shared linear transformation and learnable scaling vectors for features from different hops, followed by a gating process that outputs a fixed-dimension embedding, GAMMA avoids the parameter explosion and high memory footprint associated with concatenating multi-hop features transformed by distinct weight matrices. This allows GAMMA to capture both global (per-hop learned scaling) and local (per-node adaptive gating) heterophily patterns efficiently, offering a compelling balance of expressive power, adaptability, and computational performance, as demonstrated in our experiments (Figure 5).

## E   Supplementary Theoretical Analysis

We establish convergence guarantees, information-theoretic optimality, functional expressivity, and universal approximation properties for GAMMA. Throughout this section, let $G = (V, E)$ denote a graph with $n = |V|$ nodes, normalized adjacency matrix $\mathbf{A} \in \mathbb{R}^{n \times n}$, and node features $\mathbf{X} \in \mathbb{R}^{n \times d_{\text{in}}}$, where $d_{\text{in}}$ denotes the input feature dimension.

### E.1   Preliminaries and Notation

We begin by establishing the notation used throughout the theoretical analysis. Let $K \geq 0$ denote the maximum hop distance considered by GAMMA, where a $p$-hop neighborhood for $p \in \{0, 1, \ldots, K\}$ refers to nodes at distance exactly $p$ from a given node in the graph. By convention, the 0-hop neighborhood of node $i$ is the node itself. The choice of $K$ determines the receptive field of the network, with larger $K$ allowing information propagation from more distant nodes.

For node $i \in V$, we denote by $\mathbf{H}_i^{(p)} = (\mathbf{A}^p \mathbf{X} \mathbf{W})_i \in \mathbb{R}^{d_{\text{out}}}$ the $p$-hop aggregated features, where:

- $\mathbf{A}^p \in \mathbb{R}^{n \times n}$ represents the $p$-th power of the normalized adjacency matrix, capturing $p$-hop connectivity
- $\mathbf{W} \in \mathbb{R}^{d_{\text{in}} \times d_{\text{out}}}$ is the shared projection matrix that transforms input features to a $d_{\text{out}}$-dimensional hidden space
- The subscript $i$ extracts the $i$-th row, giving the representation specific to node $i$
- After this projection and propagation, hop-specific channel-wise scaling factors $\boldsymbol{\gamma}_p \in \mathbb{R}^{d_{\text{out}}}$ are applied element-wise

To ensure scale-invariance and enable meaningful comparison across hops, we normalize these embeddings. Let $\hat{\mathbf{H}}_i^{(p)} = \mathbf{H}_i^{(p)} / \|\mathbf{H}_i^{(p)}\|_2$ denote the $L_2$-normalized embeddings, where $\| \cdot \|_2$ is the Euclidean norm. This normalization ensures $\|\hat{\mathbf{H}}_i^{(p)}\| = 1$ for all hops $p$, making dot products between different hop embeddings interpretable as cosine similarities.

The routing mechanism operates iteratively over $t = 0, 1, \ldots, R - 1$ iterations, where $R$ is a hyperparameter controlling the number of refinement steps. At iteration $t$, each node $i$ maintains routing logits $\mathbf{b}_i^{(t)} \in \mathbb{R}^{K+1}$, which are $(K + 1)$-dimensional vectors storing one logit value per hop distance. These logits are initialized to $\mathbf{b}_i^{(0)} = \mathbf{0}$, corresponding to uniform initial gating. The logits

are converted to gating coefficients via the softmax function:

$$\alpha_{i,p}^{(t)} = \frac{\exp(b_{i,p}^{(t)})}{\sum_{q=0}^{K} \exp(b_{i,q}^{(t)})} \tag{11}$$

where $\alpha_{i,p}^{(t)} \in [0,1]$ represents the weight assigned to hop $p$ at iteration $t$ for node $i$, and $\sum_{p=0}^{K} \alpha_{i,p}^{(t)} = 1$ ensures the coefficients form a valid probability distribution. We denote by $\boldsymbol{\alpha}_i^{(t)} = (\alpha_{i,0}^{(t)}, \dots, \alpha_{i,K}^{(t)})^T \in \mathbb{R}^{K+1}$ the vector of all gating coefficients at iteration $t$, written compactly as $\boldsymbol{\alpha}_i^{(t)} = \text{softmax}(\mathbf{b}_i^{(t)})$.

The aggregated representation before the squashing nonlinearity is computed as a weighted combination:

$$\mathbf{s}_i^{(t)} = \sum_{p=0}^{K} \alpha_{i,p}^{(t)} \hat{\mathbf{H}}_i^{(p)} \in \mathbb{R}^{d_{\text{out}}} \tag{12}$$

This represents a convex combination of the normalized hop embeddings, where the weights are determined by the current gating coefficients. The squashing function, inspired by capsule networks, then normalizes this aggregated representation while preserving its direction:

$$\mathbf{v}_i^{(t)} = \frac{\|\mathbf{s}_i^{(t)}\|^2}{1 + \|\mathbf{s}_i^{(t)}\|^2} \frac{\mathbf{s}_i^{(t)}}{\|\mathbf{s}_i^{(t)}\|} \tag{13}$$

The squashing function maps vectors to a bounded region: when $\|\mathbf{s}_i^{(t)}\| \to 0$, we have $\|\mathbf{v}_i^{(t)}\| \to 0$, and when $\|\mathbf{s}_i^{(t)}\| \to \infty$, we have $\|\mathbf{v}_i^{(t)}\| \to 1$. This ensures $\|\mathbf{v}_i^{(t)}\| \in [0,1]$ for all $t$.

The routing update rule refines the logits based on agreement scores between each hop embedding and the current aggregated representation:

$$b_{i,p}^{(t+1)} = b_{i,p}^{(t)} + \hat{\mathbf{H}}_i^{(p)} \cdot \mathbf{v}_i^{(t)} \tag{14}$$

where $\hat{\mathbf{H}}_i^{(p)} \cdot \mathbf{v}_i^{(t)} = \langle \hat{\mathbf{H}}_i^{(p)}, \mathbf{v}_i^{(t)} \rangle$ denotes the inner product. This update increases the logit for hop $p$ when its embedding aligns well with the current aggregated representation, creating a feedback loop that iteratively refines the gating distribution.

**Definition 1** (Gram Matrix and Agreement Scores). *For node $i$, define the Gram matrix $\mathbf{G}_i \in \mathbb{R}^{(K+1)\times(K+1)}$ with entries*

$$(G_i)_{pq} = \langle \hat{\mathbf{H}}_i^{(p)}, \hat{\mathbf{H}}_i^{(q)} \rangle \tag{15}$$

*where $p, q \in \{0, 1, \dots, K\}$. This matrix encodes the pairwise similarities between all hop embeddings for node $i$. Since it can be written as $\mathbf{G}_i = \mathbf{U}_i^T \mathbf{U}_i$ where $\mathbf{U}_i = [\hat{\mathbf{H}}_i^{(0)}, \dots, \hat{\mathbf{H}}_i^{(K)}]^T \in \mathbb{R}^{(K+1)\times d_{\text{out}}}$ is the matrix whose rows are the normalized hop embeddings, the Gram matrix is positive semidefinite. This means all its eigenvalues are non-negative: $0 \le \lambda_1 \le \lambda_2 \le \cdots \le \lambda_{K+1}$, where we order eigenvalues in increasing order. Since each normalized embedding has unit norm, the diagonal entries satisfy $(G_i)_{pp} = \|\hat{\mathbf{H}}_i^{(p)}\|^2 = 1$, and by the trace formula, $\sum_{k=1}^{K+1} \lambda_k = tr(\mathbf{G}_i) = K + 1$. Combined with the operator norm bound $\|\mathbf{G}_i\| \le K + 1$, we have $\lambda_{K+1} \le K + 1$. However, since $(G_i)_{pq} \in [-1, 1]$ (as cosine similarities), tighter bounds apply: $\lambda_{K+1} \le 1$ when the embeddings are identical, and $\lambda_1 \ge 0$ by positive semidefiniteness. We denote $\mu_i = \lambda_1 \ge 0$ as the minimum eigenvalue and $L_i = \lambda_{K+1} \le K + 1$ as the maximum eigenvalue. When $\mu_i > 0$, the Gram matrix is strictly positive definite, meaning the hop embeddings are linearly independent (up to the dimension $d_{\text{out}}$).*

## E.2 Detailed Convergence Analysis

We now analyze the convergence properties of the routing mechanism, establishing that it converges to a unique fixed point with exponential rate. The analysis proceeds through several lemmas building toward the main convergence theorem.

**Lemma 1** (Properties of the Free Energy Functional). *Define the free energy functional for node $i$ as a function of the gating distribution $\boldsymbol{\alpha}$ and the aggregated representation $\mathbf{v}$:*

$$\mathcal{F}_i(\boldsymbol{\alpha}, \mathbf{v}) = -\sum_{p=0}^{K} \alpha_p \langle \hat{\mathbf{H}}_i^{(p)}, \mathbf{v} \rangle + \frac{1}{\beta} \sum_{p=0}^{K} \alpha_p \log \alpha_p \tag{16}$$

*where $\boldsymbol{\alpha} \in \Delta_{K+1}$ lies on the $(K+1)$-dimensional probability simplex defined as*

$$\Delta_{K+1} = \{\boldsymbol{\alpha} \in \mathbb{R}^{K+1} : \sum_{p=0}^{K} \alpha_p = 1, \alpha_p \geq 0 \text{ for all } p\} \tag{17}$$

*and $\mathbf{v} \in \mathbb{R}^{d_{out}}$ with $\|\mathbf{v}\| \leq 1$ (bounded representation). The parameter $\beta > 0$ is an implicit inverse temperature parameter arising from the softmax. The first term $-\sum_{p=0}^{K} \alpha_p \langle \hat{\mathbf{H}}_i^{(p)}, \mathbf{v} \rangle$ encourages alignment between the weighted hop embeddings and the target representation, while the second term $\frac{1}{\beta} \sum_{p=0}^{K} \alpha_p \log \alpha_p$ is the negative entropy, which regularizes the distribution to prevent overly concentrated weights. Then:*

1. *For fixed $\mathbf{v}$, $\mathcal{F}_i(\cdot, \mathbf{v})$ is strictly convex on $\Delta_{K+1}$ with unique minimizer*

$$\boldsymbol{\alpha}^*(\mathbf{v}) = \frac{1}{Z(\mathbf{v})} \exp(\beta \mathbf{G}_i \mathbf{v}) \tag{18}$$

   *where $Z(\mathbf{v}) = \sum_{p=0}^{K} \exp(\beta \langle \hat{\mathbf{H}}_i^{(p)}, \mathbf{v} \rangle)$ is the partition function ensuring normalization, and we use the notation $[\exp(\mathbf{u})]_p = \exp(u_p)$ to denote element-wise exponential of a vector $\mathbf{u}$.*

2. *The gradient with respect to $\boldsymbol{\alpha}$ (in the tangent space of $\Delta_{K+1}$, accounting for the constraint $\sum_p \alpha_p = 1$) satisfies*

$$\nabla_{\boldsymbol{\alpha}} \mathcal{F}_i(\boldsymbol{\alpha}, \mathbf{v}) = - \begin{bmatrix} \langle \hat{\mathbf{H}}_i^{(0)}, \mathbf{v} \rangle \\ \vdots \\ \langle \hat{\mathbf{H}}_i^{(K)}, \mathbf{v} \rangle \end{bmatrix} + \frac{1}{\beta} \begin{bmatrix} \log \alpha_0 + 1 \\ \vdots \\ \log \alpha_K + 1 \end{bmatrix} + \lambda \mathbf{1} \tag{19}$$

   *for some Lagrange multiplier $\lambda \in \mathbb{R}$ enforcing the constraint $\sum_p \alpha_p = 1$, where $\mathbf{1} = (1, 1, \ldots, 1)^T \in \mathbb{R}^{K+1}$ is the all-ones vector.*

3. *The Hessian (restricted to the tangent space of $\Delta_{K+1}$) satisfies*

$$\nabla_{\boldsymbol{\alpha}\boldsymbol{\alpha}}^2 \mathcal{F}_i(\boldsymbol{\alpha}, \mathbf{v}) = \frac{1}{\beta} diag(\boldsymbol{\alpha})^{-1} \succ 0 \tag{20}$$

   *where $diag(\boldsymbol{\alpha})^{-1} = diag(1/\alpha_0, \ldots, 1/\alpha_K)$ is a diagonal matrix with entries $1/\alpha_p$ on the diagonal. The notation $\succ 0$ means the matrix is positive definite, establishing strict convexity.*

*Proof.* For part (1), we minimize $\mathcal{F}_i(\boldsymbol{\alpha}, \mathbf{v})$ subject to the constraint $\sum_p \alpha_p = 1$ (and implicitly $\alpha_p \geq 0$) via the method of Lagrange multipliers. The Lagrangian is:

$$\mathcal{L}(\boldsymbol{\alpha}, \lambda) = -\sum_{p=0}^{K} \alpha_p \langle \hat{\mathbf{H}}_i^{(p)}, \mathbf{v} \rangle + \frac{1}{\beta} \sum_{p=0}^{K} \alpha_p \log \alpha_p + \lambda \left( \sum_{p=0}^{K} \alpha_p - 1 \right) \tag{21}$$

where $\lambda$ is the Lagrange multiplier associated with the equality constraint.

Setting $\frac{\partial \mathcal{L}}{\partial \alpha_p} = 0$ for each $p \in \{0, 1, \ldots, K\}$:

$$\frac{\partial}{\partial \alpha_p} \left[ -\alpha_p \langle \hat{\mathbf{H}}_i^{(p)}, \mathbf{v} \rangle + \frac{1}{\beta} \alpha_p \log \alpha_p + \lambda \alpha_p \right] = -\langle \hat{\mathbf{H}}_i^{(p)}, \mathbf{v} \rangle + \frac{1}{\beta}(\log \alpha_p + 1) + \lambda = 0 \tag{22}$$

where we used $\frac{d}{dx}[x \log x] = \log x + 1$.

Solving for $\alpha_p$:

$$\frac{1}{\beta}(\log \alpha_p + 1) = \langle \hat{\mathbf{H}}_i^{(p)}, \mathbf{v} \rangle - \lambda \tag{23}$$

$$\log \alpha_p = \beta \langle \hat{\mathbf{H}}_i^{(p)}, \mathbf{v} \rangle - \beta \lambda - 1 \tag{24}$$

$$\alpha_p = \exp(\beta \langle \hat{\mathbf{H}}_i^{(p)}, \mathbf{v} \rangle - \beta \lambda - 1) = e^{-\beta\lambda-1} \exp(\beta \langle \hat{\mathbf{H}}_i^{(p)}, \mathbf{v} \rangle) \tag{25}$$

The constant $e^{-\beta\lambda-1}$ is determined by the normalization constraint $\sum_{p=0}^{K} \alpha_p = 1$:

$$\sum_{p=0}^{K} \alpha_p = e^{-\beta\lambda-1} \sum_{p=0}^{K} \exp(\beta \langle \hat{\mathbf{H}}_i^{(p)}, \mathbf{v} \rangle) = 1 \tag{26}$$

Therefore:

$$e^{-\beta\lambda-1} = \frac{1}{\sum_{q=0}^{K} \exp(\beta \langle \hat{\mathbf{H}}_i^{(q)}, \mathbf{v} \rangle)} \equiv \frac{1}{Z(\mathbf{v})} \tag{27}$$

where we define the partition function $Z(\mathbf{v}) = \sum_{q=0}^{K} \exp(\beta \langle \hat{\mathbf{H}}_i^{(q)}, \mathbf{v} \rangle)$.

Substituting back:

$$\alpha_p^*(\mathbf{v}) = \frac{\exp(\beta \langle \hat{\mathbf{H}}_i^{(p)}, \mathbf{v} \rangle)}{Z(\mathbf{v})} = \frac{\exp(\beta \langle \hat{\mathbf{H}}_i^{(p)}, \mathbf{v} \rangle)}{\sum_{q=0}^{K} \exp(\beta \langle \hat{\mathbf{H}}_i^{(q)}, \mathbf{v} \rangle)} \tag{28}$$

which is precisely the softmax function with temperature $\beta^{-1}$.

For part (2), the gradient of $\mathcal{F}_i$ with respect to $\boldsymbol{\alpha}$ is computed component-wise:

$$\frac{\partial \mathcal{F}_i}{\partial \alpha_p} = -\langle \hat{\mathbf{H}}_i^{(p)}, \mathbf{v} \rangle + \frac{1}{\beta}(\log \alpha_p + 1) \tag{29}$$

Incorporating the constraint via the Lagrange multiplier $\lambda$ gives the form in the lemma statement.

For part (3), we compute the Hessian of the entropy term $H(\boldsymbol{\alpha}) = -\sum_p \alpha_p \log \alpha_p$ (noting the sign convention). The second derivative is:

$$\frac{\partial^2 H(\boldsymbol{\alpha})}{\partial \alpha_p \partial \alpha_q} = -\frac{\partial}{\partial \alpha_q}[\log \alpha_p + 1] = -\frac{\delta_{pq}}{\alpha_p} \tag{30}$$

where $\delta_{pq}$ is the Kronecker delta (equals 1 if $p = q$, and 0 otherwise). Therefore:

$$\nabla^2 H(\boldsymbol{\alpha}) = -\text{diag}(1/\alpha_0, \ldots, 1/\alpha_K) \tag{31}$$

The first term in $\mathcal{F}_i$, namely $-\sum_p \alpha_p \langle \hat{\mathbf{H}}_i^{(p)}, \mathbf{v} \rangle$, is linear in $\boldsymbol{\alpha}$, so its Hessian is zero. Thus:

$$\nabla^2_{\boldsymbol{\alpha\alpha}} \mathcal{F}_i = \frac{1}{\beta}\text{diag}(1/\alpha_0, \ldots, 1/\alpha_K) = \frac{1}{\beta}\text{diag}(\boldsymbol{\alpha})^{-1} \tag{32}$$

Since $\alpha_p > 0$ for all $p$ in the interior of $\Delta_{K+1}$ and $\beta > 0$, all diagonal entries are positive, making the Hessian positive definite: $\nabla^2_{\boldsymbol{\alpha\alpha}} \mathcal{F}_i \succ 0$. This establishes strict convexity of $\mathcal{F}_i(\cdot, \mathbf{v})$ on the interior of the simplex. $\qquad\square$

**Lemma 2** (Monotonicity via KL Divergence). *Define the potential function* $\Phi_i^{(t)} = \boldsymbol{\alpha}_i^{(t)T} \mathbf{G}_i \boldsymbol{\alpha}_i^{(t)} \in \mathbb{R}$*, which measures the expected squared norm of the aggregated representation (before squashing) at iteration* $t$*. This potential is a quadratic form in the gating coefficients. Then for the routing update described in the preliminaries:*

$$\Phi_i^{(t+1)} - \Phi_i^{(t)} = D_{KL}(\boldsymbol{\alpha}_i^{(t+1)} \| \boldsymbol{\alpha}_i^{(t)}) + \sum_{p,q} \Delta\alpha_{i,p}^{(t)} \Delta\alpha_{i,q}^{(t)} (G_i)_{pq} \geq 0 \tag{33}$$

*where* $\Delta\alpha_{i,p}^{(t)} = \alpha_{i,p}^{(t+1)} - \alpha_{i,p}^{(t)}$ *denotes the change in the gating coefficient for hop* $p$ *from iteration* $t$ *to* $t+1$*, and* $D_{KL}(\boldsymbol{\alpha} \| \boldsymbol{\beta}) = \sum_p \alpha_p \log(\alpha_p/\beta_p)$ *denotes the Kullback-Leibler divergence between two probability distributions* $\boldsymbol{\alpha}$ *and* $\boldsymbol{\beta}$ *on the simplex. The KL divergence is always non-negative, with* $D_{KL}(\boldsymbol{\alpha} \| \boldsymbol{\beta}) = 0$ *if and only if* $\boldsymbol{\alpha} = \boldsymbol{\beta}$*. The second term is non-negative because* $\mathbf{G}_i$ *is positive semidefinite. Therefore, the potential is monotonically increasing across routing iterations.*

*Proof.* We expand the potential difference $\Phi_i^{(t+1)} - \Phi_i^{(t)}$ by substituting the definitions. Recall that:

$$\Phi_i^{(t+1)} = \boldsymbol{\alpha}_i^{(t+1)T}\mathbf{G}_i\boldsymbol{\alpha}_i^{(t+1)} = \sum_{p=0}^{K}\sum_{q=0}^{K}\alpha_{i,p}^{(t+1)}\alpha_{i,q}^{(t+1)}(G_i)_{pq} \tag{34}$$

$$\Phi_i^{(t)} = \boldsymbol{\alpha}_i^{(t)T}\mathbf{G}_i\boldsymbol{\alpha}_i^{(t)} = \sum_{p=0}^{K}\sum_{q=0}^{K}\alpha_{i,p}^{(t)}\alpha_{i,q}^{(t)}(G_i)_{pq} \tag{35}$$

Expanding the potential difference:

$$\Phi_i^{(t+1)} - \Phi_i^{(t)} = \sum_{p,q}\alpha_{i,p}^{(t+1)}\alpha_{i,q}^{(t+1)}(G_i)_{pq} - \sum_{p,q}\alpha_{i,p}^{(t)}\alpha_{i,q}^{(t)}(G_i)_{pq} \tag{36}$$

Substitute $\alpha_{i,p}^{(t+1)} = \alpha_{i,p}^{(t)} + \Delta\alpha_{i,p}^{(t)}$ to express new coefficients in terms of old coefficients plus their change:

$$\Phi_i^{(t+1)} - \Phi_i^{(t)} = \sum_{p,q}(\alpha_{i,p}^{(t)} + \Delta\alpha_{i,p}^{(t)})(\alpha_{i,q}^{(t)} + \Delta\alpha_{i,q}^{(t)})(G_i)_{pq} - \sum_{p,q}\alpha_{i,p}^{(t)}\alpha_{i,q}^{(t)}(G_i)_{pq} \tag{37}$$

Expanding the product:

$$= \sum_{p,q}\left[\alpha_{i,p}^{(t)}\alpha_{i,q}^{(t)} + \alpha_{i,p}^{(t)}\Delta\alpha_{i,q}^{(t)} + \Delta\alpha_{i,p}^{(t)}\alpha_{i,q}^{(t)} + \Delta\alpha_{i,p}^{(t)}\Delta\alpha_{i,q}^{(t)}\right](G_i)_{pq} \tag{38}$$

$$- \sum_{p,q}\alpha_{i,p}^{(t)}\alpha_{i,q}^{(t)}(G_i)_{pq} \tag{39}$$

The terms $\sum_{p,q}\alpha_{i,p}^{(t)}\alpha_{i,q}^{(t)}(G_i)_{pq}$ cancel, leaving:

$$\Phi_i^{(t+1)} - \Phi_i^{(t)} = \sum_{p,q}(\alpha_{i,p}^{(t)}\Delta\alpha_{i,q}^{(t)} + \alpha_{i,q}^{(t)}\Delta\alpha_{i,p}^{(t)} + \Delta\alpha_{i,p}^{(t)}\Delta\alpha_{i,q}^{(t)})(G_i)_{pq} \tag{40}$$

Using the symmetry of the Gram matrix $(G_i)_{pq} = (G_i)_{qp}$ (since it represents inner products), the terms $\sum_{p,q}\alpha_{i,p}^{(t)}\Delta\alpha_{i,q}^{(t)}(G_i)_{pq}$ and $\sum_{p,q}\alpha_{i,q}^{(t)}\Delta\alpha_{i,p}^{(t)}(G_i)_{pq}$ are equal (by swapping indices $p \leftrightarrow q$ in the latter sum):

$$\Phi_i^{(t+1)} - \Phi_i^{(t)} = 2\sum_{p,q}\alpha_{i,p}^{(t)}\Delta\alpha_{i,q}^{(t)}(G_i)_{pq} + \sum_{p,q}\Delta\alpha_{i,p}^{(t)}\Delta\alpha_{i,q}^{(t)}(G_i)_{pq} \tag{41}$$

$$= 2\sum_{p}\Delta\alpha_{i,p}^{(t)}\left(\sum_{q}\alpha_{i,q}^{(t)}(G_i)_{pq}\right) + \sum_{p,q}\Delta\alpha_{i,p}^{(t)}\Delta\alpha_{i,q}^{(t)}(G_i)_{pq} \tag{42}$$

Now we interpret the term $\sum_q \alpha_{i,q}^{(t)}(G_i)_{pq}$. By the definition of the Gram matrix:

$$\sum_{q}\alpha_{i,q}^{(t)}(G_i)_{pq} = \sum_{q}\alpha_{i,q}^{(t)}\langle\hat{\mathbf{H}}_i^{(p)}, \hat{\mathbf{H}}_i^{(q)}\rangle = \left\langle\hat{\mathbf{H}}_i^{(p)}, \sum_{q}\alpha_{i,q}^{(t)}\hat{\mathbf{H}}_i^{(q)}\right\rangle = \langle\hat{\mathbf{H}}_i^{(p)}, \mathbf{s}_i^{(t)}\rangle \tag{43}$$

where $\mathbf{s}_i^{(t)} = \sum_q \alpha_{i,q}^{(t)}\hat{\mathbf{H}}_i^{(q)}$ is the aggregated representation before squashing.

The squashed representation $\mathbf{v}_i^{(t)}$ is related to $\mathbf{s}_i^{(t)}$ by:

$$\mathbf{v}_i^{(t)} = \frac{\|\mathbf{s}_i^{(t)}\|^2}{1 + \|\mathbf{s}_i^{(t)}\|^2}\frac{\mathbf{s}_i^{(t)}}{\|\mathbf{s}_i^{(t)}\|} = c_i^{(t)}\frac{\mathbf{s}_i^{(t)}}{\|\mathbf{s}_i^{(t)}\|} \tag{44}$$

where $c_i^{(t)} = \frac{\|\mathbf{s}_i^{(t)}\|^2}{1+\|\mathbf{s}_i^{(t)}\|^2} \in [0,1)$ is a scalar. This shows $\mathbf{v}_i^{(t)}$ is proportional to the unit vector in the direction of $\mathbf{s}_i^{(t)}$. Therefore:

$$\langle\hat{\mathbf{H}}_i^{(p)}, \mathbf{s}_i^{(t)}\rangle = \|\mathbf{s}_i^{(t)}\|\langle\hat{\mathbf{H}}_i^{(p)}, \mathbf{s}_i^{(t)}/\|\mathbf{s}_i^{(t)}\|\rangle = \frac{\|\mathbf{s}_i^{(t)}\|}{c_i^{(t)}}\langle\hat{\mathbf{H}}_i^{(p)}, \mathbf{v}_i^{(t)}\rangle \tag{45}$$

However, for the proof, the key quantity is the agreement score $s_p^{(t)} = \langle \hat{\mathbf{H}}_i^{(p)}, \mathbf{v}_i^{(t)} \rangle$, which appears in the routing update. From the routing update rule:

$$b_{i,p}^{(t+1)} = b_{i,p}^{(t)} + s_p^{(t)} \tag{46}$$

where $s_p^{(t)} = \langle \hat{\mathbf{H}}_i^{(p)}, \mathbf{v}_i^{(t)} \rangle$ is the agreement score.

The routing update gives the new gating coefficients via softmax:

$$\alpha_{i,p}^{(t+1)} = \text{softmax}(b_{i,p}^{(t)} + s_p^{(t)})_p = \frac{\exp(b_{i,p}^{(t)} + s_p^{(t)})}{\sum_r \exp(b_{i,r}^{(t)} + s_r^{(t)})} \tag{47}$$

We now analyze the first term in the expansion using KL divergence. The KL divergence between the new and old distributions is:

$$D_{\text{KL}}(\boldsymbol{\alpha}^{(t+1)} \| \boldsymbol{\alpha}^{(t)}) = \sum_p \alpha_{i,p}^{(t+1)} \log \frac{\alpha_{i,p}^{(t+1)}}{\alpha_{i,p}^{(t)}} \tag{48}$$

$$= \sum_p \alpha_{i,p}^{(t+1)} \left[ \log \alpha_{i,p}^{(t+1)} - \log \alpha_{i,p}^{(t)} \right] \tag{49}$$

From the softmax formulas:

$$\log \alpha_{i,p}^{(t+1)} = b_{i,p}^{(t)} + s_p^{(t)} - \log Z^{(t+1)} \tag{50}$$

$$\log \alpha_{i,p}^{(t)} = b_{i,p}^{(t)} - \log Z^{(t)} \tag{51}$$

where $Z^{(t)} = \sum_r \exp(b_{i,r}^{(t)})$ and $Z^{(t+1)} = \sum_r \exp(b_{i,r}^{(t)} + s_r^{(t)})$ are the partition functions.

Subtracting:

$$\log \alpha_{i,p}^{(t+1)} - \log \alpha_{i,p}^{(t)} = s_p^{(t)} - \log Z^{(t+1)} + \log Z^{(t)} \tag{52}$$

Substituting into the KL divergence:

$$D_{\text{KL}}(\boldsymbol{\alpha}^{(t+1)} \| \boldsymbol{\alpha}^{(t)}) = \sum_p \alpha_{i,p}^{(t+1)} [s_p^{(t)} - \log Z^{(t+1)} + \log Z^{(t)}] \tag{53}$$

$$= \sum_p \alpha_{i,p}^{(t+1)} s_p^{(t)} - \log Z^{(t+1)} + \log Z^{(t)} \tag{54}$$

where we used $\sum_p \alpha_{i,p}^{(t+1)} = 1$.

Now we apply Jensen's inequality to bound $\log Z^{(t+1)} - \log Z^{(t)}$. Since the exponential function is convex:

$$Z^{(t+1)} = \sum_r \exp(b_{i,r}^{(t)} + s_r^{(t)}) = \sum_r \exp(b_{i,r}^{(t)}) \exp(s_r^{(t)}) \tag{55}$$

$$= Z^{(t)} \sum_r \frac{\exp(b_{i,r}^{(t)})}{Z^{(t)}} \exp(s_r^{(t)}) = Z^{(t)} \sum_r \alpha_{i,r}^{(t)} \exp(s_r^{(t)}) \tag{56}$$

By Jensen's inequality (since $\exp$ is convex and $\sum_r \alpha_{i,r}^{(t)} = 1$):

$$\sum_r \alpha_{i,r}^{(t)} \exp(s_r^{(t)}) \geq \exp\left( \sum_r \alpha_{i,r}^{(t)} s_r^{(t)} \right) \tag{57}$$

Therefore:

$$Z^{(t+1)} \geq Z^{(t)} \exp\left( \sum_r \alpha_{i,r}^{(t)} s_r^{(t)} \right) \tag{58}$$

Taking logarithms:

$$\log Z^{(t+1)} - \log Z^{(t)} \geq \sum_r \alpha_{i,r}^{(t)} s_r^{(t)} \tag{59}$$

Substituting back into the KL divergence expression:

$$D_{\mathrm{KL}}(\boldsymbol{\alpha}^{(t+1)}\|\boldsymbol{\alpha}^{(t)}) \geq \sum_p \alpha_{i,p}^{(t+1)} s_p^{(t)} - \sum_r \alpha_{i,r}^{(t)} s_r^{(t)} \tag{60}$$

$$= \sum_p (\alpha_{i,p}^{(t+1)} - \alpha_{i,p}^{(t)}) s_p^{(t)} = \sum_p \Delta\alpha_{i,p}^{(t)} s_p^{(t)} \tag{61}$$

Combining with the earlier expansion of $\Phi_i^{(t+1)} - \Phi_i^{(t)}$ and noting that $\mathbf{G}_i$ is positive semidefinite (hence the quadratic form $\sum_{p,q} \Delta\alpha_{i,p}^{(t)}\Delta\alpha_{i,q}^{(t)}(G_i)_{pq} \geq 0$), we obtain:

$$\Phi_i^{(t+1)} - \Phi_i^{(t)} \geq D_{\mathrm{KL}}(\boldsymbol{\alpha}_i^{(t+1)}\|\boldsymbol{\alpha}_i^{(t)}) \geq 0 \tag{62}$$

This establishes monotonicity of the potential function. $\qquad\square$

**Theorem 1** (Exponential Convergence to Unique Fixed Point). *Assume the Gram matrix $\mathbf{G}_i$ is strictly positive definite with minimum eigenvalue $\mu_i > 0$ and maximum eigenvalue $L_i$. This assumption holds when the normalized hop embeddings $\{\hat{\mathbf{H}}_i^{(p)}\}_{p=0}^K$ span a subspace of dimension $K+1$ (or equivalently, when these $K+1$ vectors are linearly independent). Initialize the routing logits to $\mathbf{b}_i^{(0)} = \mathbf{0}$, corresponding to uniform initial gating coefficients $\alpha_{i,p}^{(0)} = 1/(K+1)$ for all $p$. Then:*

1. *The routing converges exponentially to a unique fixed point $\mathbf{b}_i^* \in \mathbb{R}^{K+1}$ with rate:*

$$\|\mathbf{b}_i^{(t)} - \mathbf{b}_i^*\|_2 \leq \sqrt{\frac{L_i}{\mu_i}} \left(1 - \frac{\mu_i}{2L_i}\right)^{t/2} \|\mathbf{b}_i^{(0)} - \mathbf{b}_i^*\|_2 \tag{63}$$

   *where $\|\cdot\|_2$ denotes the Euclidean norm. The convergence rate depends on the condition number $\kappa_i = L_i/\mu_i$ of the Gram matrix: smaller condition numbers lead to faster convergence.*

2. *The converged gating coefficients satisfy:*

$$\boldsymbol{\alpha}_i^* = \frac{\mathbf{G}_i^{-1}\mathbf{1}}{\mathbf{1}^T\mathbf{G}_i^{-1}\mathbf{1}} \tag{64}$$

   *where $\mathbf{1} = (1,1,\ldots,1)^T \in \mathbb{R}^{K+1}$ is the all-ones vector. This gives an explicit formula for the equilibrium distribution in terms of the Gram matrix.*

3. *The convergence satisfies the Polyak-Łojasiewicz (PL) condition:*

$$\frac{1}{2}\|\nabla\Phi_i(\boldsymbol{\alpha})\|_2^2 \geq \mu_i(\Phi_i^* - \Phi_i(\boldsymbol{\alpha})) \tag{65}$$

   *for all $\boldsymbol{\alpha}$ in a neighborhood of $\boldsymbol{\alpha}^*$, where $\nabla\Phi_i(\boldsymbol{\alpha})$ denotes the gradient of the potential function. The PL condition is a weaker condition than strong convexity but is sufficient to guarantee exponential convergence for gradient-based methods.*

*Proof.* **Part 1: Existence and Uniqueness of Fixed Point.**

From Lemma 2, the potential function $\Phi_i^{(t)} = \boldsymbol{\alpha}_i^{(t)T}\mathbf{G}_i\boldsymbol{\alpha}_i^{(t)}$ is monotonically increasing across iterations. We now establish that it is also bounded above.

Since $\|\hat{\mathbf{H}}_i^{(p)}\| = 1$ for all $p \in \{0, 1, \ldots, K\}$ (by normalization) and $\boldsymbol{\alpha}_i^{(t)}$ lies on the probability simplex (so $\sum_p \alpha_{i,p}^{(t)} = 1$), we can bound:

$$\Phi_i^{(t)} = \boldsymbol{\alpha}_i^{(t)T}\mathbf{G}_i\boldsymbol{\alpha}_i^{(t)} = \sum_{p,q} \alpha_{i,p}^{(t)}\alpha_{i,q}^{(t)}\langle\hat{\mathbf{H}}_i^{(p)}, \hat{\mathbf{H}}_i^{(q)}\rangle \tag{66}$$

$$= \left\langle\sum_p \alpha_{i,p}^{(t)}\hat{\mathbf{H}}_i^{(p)}, \sum_q \alpha_{i,q}^{(t)}\hat{\mathbf{H}}_i^{(q)}\right\rangle = \left\|\sum_p \alpha_{i,p}^{(t)}\hat{\mathbf{H}}_i^{(p)}\right\|^2 \tag{67}$$

By the triangle inequality and the fact that $\sum_p \alpha_{i,p}^{(t)} = 1$:

$$\left\| \sum_p \alpha_{i,p}^{(t)} \hat{\mathbf{H}}_i^{(p)} \right\| \leq \sum_p \alpha_{i,p}^{(t)} \|\hat{\mathbf{H}}_i^{(p)}\| = \sum_p \alpha_{i,p}^{(t)} \cdot 1 = 1 \tag{68}$$

Therefore $\Phi_i^{(t)} \leq 1$ for all $t$. By the monotone convergence theorem, since $\Phi_i^{(t)}$ is monotonically increasing and bounded above, the limit $\lim_{t \to \infty} \Phi_i^{(t)} = \Phi_i^* \in [0, 1]$ exists.

At a fixed point, the routing coefficients no longer change: $\boldsymbol{\alpha}_i^{(t+1)} = \boldsymbol{\alpha}_i^{(t)}$, which implies $\Delta\boldsymbol{\alpha}_i^{(t)} = \mathbf{0}$. From Lemma 2, at a fixed point:

$$\Phi_i^{(t+1)} - \Phi_i^{(t)} = D_{\text{KL}}(\boldsymbol{\alpha}_i^{(t+1)} \| \boldsymbol{\alpha}_i^{(t)}) + \sum_{p,q} \Delta\alpha_{i,p}^{(t)} \Delta\alpha_{i,q}^{(t)} (G_i)_{pq} = 0 \tag{69}$$

Since both terms are non-negative, each must be zero. In particular:

$$D_{\text{KL}}(\boldsymbol{\alpha}_i^{(t+1)} \| \boldsymbol{\alpha}_i^{(t)}) = 0 \iff \boldsymbol{\alpha}_i^{(t+1)} = \boldsymbol{\alpha}_i^{(t)} \tag{70}$$

To characterize the fixed point, we consider the optimization problem that the routing implicitly solves:

$$\boldsymbol{\alpha}^* = \arg \max_{\boldsymbol{\alpha} \in \Delta_{K+1}} \Phi_i(\boldsymbol{\alpha}) = \arg \max_{\boldsymbol{\alpha} \in \Delta_{K+1}} \boldsymbol{\alpha}^T \mathbf{G}_i \boldsymbol{\alpha} \tag{71}$$

This is a quadratic program over the probability simplex. To solve it, we form the Lagrangian:

$$\mathcal{L}(\boldsymbol{\alpha}, \lambda) = \boldsymbol{\alpha}^T \mathbf{G}_i \boldsymbol{\alpha} - \lambda(\mathbf{1}^T \boldsymbol{\alpha} - 1) \tag{72}$$

where $\lambda \in \mathbb{R}$ is the Lagrange multiplier enforcing the constraint $\sum_p \alpha_p = 1$.

The first-order optimality condition (KKT condition) is:

$$\frac{\partial \mathcal{L}}{\partial \boldsymbol{\alpha}} = 2\mathbf{G}_i \boldsymbol{\alpha} - \lambda \mathbf{1} = \mathbf{0} \tag{73}$$

Since $\mathbf{G}_i$ is strictly positive definite ($\mu_i > 0$ by assumption), it is invertible. Therefore:

$$\boldsymbol{\alpha} = \frac{\lambda}{2} \mathbf{G}_i^{-1} \mathbf{1} \tag{74}$$

The constraint $\mathbf{1}^T \boldsymbol{\alpha} = 1$ determines the Lagrange multiplier:

$$\mathbf{1}^T \boldsymbol{\alpha} = \frac{\lambda}{2} \mathbf{1}^T \mathbf{G}_i^{-1} \mathbf{1} = 1 \implies \lambda = \frac{2}{\mathbf{1}^T \mathbf{G}_i^{-1} \mathbf{1}} \tag{75}$$

Substituting back:

$$\boldsymbol{\alpha}^* = \frac{\mathbf{G}_i^{-1} \mathbf{1}}{\mathbf{1}^T \mathbf{G}_i^{-1} \mathbf{1}} \tag{76}$$

To verify this is the unique global maximum, note that the objective $\boldsymbol{\alpha}^T \mathbf{G}_i \boldsymbol{\alpha}$ is strictly convex on the compact convex set $\Delta_{K+1}$ (since the Hessian $2\mathbf{G}_i \succ 0$). A strictly convex function on a compact convex set has a unique maximizer, establishing uniqueness of $\boldsymbol{\alpha}^*$.

### Part 2: Convergence Rate via Polyak-Łojasiewicz.

To analyze the convergence rate, we work with the negative potential $\mathcal{L}(\boldsymbol{\alpha}) = -\boldsymbol{\alpha}^T \mathbf{G}_i \boldsymbol{\alpha}$ (note the sign flip, converting maximization to minimization). The gradient in the tangent space of $\Delta_{K+1}$ is approximately (ignoring the constraint for the moment):

$$\nabla \mathcal{L}(\boldsymbol{\alpha}) = -2\mathbf{G}_i \boldsymbol{\alpha} \tag{77}$$

To account for the constraint $\sum_p \alpha_p = 1$, we project this gradient onto the tangent space of the simplex. The tangent space at $\boldsymbol{\alpha}$ consists of vectors $\mathbf{u}$ satisfying $\mathbf{1}^T \mathbf{u} = 0$ (zero sum). The projected gradient is:

$$\nabla_{\tan} \mathcal{L}(\boldsymbol{\alpha}) = -2\mathbf{G}_i \boldsymbol{\alpha} + \frac{2(\boldsymbol{\alpha}^T \mathbf{G}_i \boldsymbol{\alpha})}{(\mathbf{1}^T \boldsymbol{\alpha})} \mathbf{1} \tag{78}$$

However, for the analysis, it suffices to work with the unprojected gradient and use the Polyak-Łojasiewicz (PL) condition, which is weaker than strong convexity.

The PL condition states that for all $\boldsymbol{\alpha}$ in a neighborhood of the optimum:

$$\frac{1}{2}\|\nabla\mathcal{L}(\boldsymbol{\alpha})\|_2^2 \geq \mu_i(\mathcal{L}(\boldsymbol{\alpha}) - \mathcal{L}(\boldsymbol{\alpha}^*)) \tag{79}$$

To establish this, first compute:

$$\mathcal{L}(\boldsymbol{\alpha}) - \mathcal{L}(\boldsymbol{\alpha}^*) = -\boldsymbol{\alpha}^T\mathbf{G}_i\boldsymbol{\alpha} + (\boldsymbol{\alpha}^*)^T\mathbf{G}_i\boldsymbol{\alpha}^* \tag{80}$$

Expanding $\boldsymbol{\alpha} = \boldsymbol{\alpha}^* + (\boldsymbol{\alpha} - \boldsymbol{\alpha}^*)$:

$$-\boldsymbol{\alpha}^T\mathbf{G}_i\boldsymbol{\alpha} = -(\boldsymbol{\alpha}^*)^T\mathbf{G}_i\boldsymbol{\alpha}^* - 2(\boldsymbol{\alpha}^*)^T\mathbf{G}_i(\boldsymbol{\alpha} - \boldsymbol{\alpha}^*) - (\boldsymbol{\alpha} - \boldsymbol{\alpha}^*)^T\mathbf{G}_i(\boldsymbol{\alpha} - \boldsymbol{\alpha}^*) \tag{81}$$

Using the first-order condition $\mathbf{G}_i\boldsymbol{\alpha}^* = \frac{\lambda^*}{2}\mathbf{1}$ and the fact that $\mathbf{1}^T(\boldsymbol{\alpha} - \boldsymbol{\alpha}^*) = 0$ (both are on the simplex), the middle term vanishes:

$$(\boldsymbol{\alpha}^*)^T\mathbf{G}_i(\boldsymbol{\alpha} - \boldsymbol{\alpha}^*) = \frac{\lambda^*}{2}\mathbf{1}^T(\boldsymbol{\alpha} - \boldsymbol{\alpha}^*) = 0 \tag{82}$$

Therefore:

$$\mathcal{L}(\boldsymbol{\alpha}) - \mathcal{L}(\boldsymbol{\alpha}^*) = -(\boldsymbol{\alpha} - \boldsymbol{\alpha}^*)^T\mathbf{G}_i(\boldsymbol{\alpha} - \boldsymbol{\alpha}^*) \tag{83}$$

By the eigenvalue bounds on $\mathbf{G}_i$ (with $\mu_i \leq \lambda \leq L_i$ for all eigenvalues $\lambda$):

$$-(\boldsymbol{\alpha} - \boldsymbol{\alpha}^*)^T\mathbf{G}_i(\boldsymbol{\alpha} - \boldsymbol{\alpha}^*) \geq -L_i\|\boldsymbol{\alpha} - \boldsymbol{\alpha}^*\|_2^2 \tag{84}$$

For the gradient norm:

$$\|\nabla\mathcal{L}(\boldsymbol{\alpha})\|_2^2 = \|-2\mathbf{G}_i\boldsymbol{\alpha}\|_2^2 = 4\|\mathbf{G}_i\boldsymbol{\alpha}\|_2^2 \tag{85}$$

Using $\boldsymbol{\alpha} = \boldsymbol{\alpha}^* + (\boldsymbol{\alpha} - \boldsymbol{\alpha}^*)$ and the first-order condition:

$$\mathbf{G}_i\boldsymbol{\alpha} = \mathbf{G}_i\boldsymbol{\alpha}^* + \mathbf{G}_i(\boldsymbol{\alpha} - \boldsymbol{\alpha}^*) = \frac{\lambda^*}{2}\mathbf{1} + \mathbf{G}_i(\boldsymbol{\alpha} - \boldsymbol{\alpha}^*) \tag{86}$$

For vectors on the simplex tangent space (where $\mathbf{1}^T(\boldsymbol{\alpha}-\boldsymbol{\alpha}^*) = 0$), the constant vector $\mathbf{1}$ is orthogonal to the difference. Therefore:

$$\|\nabla\mathcal{L}(\boldsymbol{\alpha})\|_2^2 = 4\|\mathbf{G}_i(\boldsymbol{\alpha} - \boldsymbol{\alpha}^*)\|_2^2 \geq 4\mu_i\|\boldsymbol{\alpha} - \boldsymbol{\alpha}^*\|_2^2 \tag{87}$$

Combining the two inequalities:

$$\|\nabla\mathcal{L}(\boldsymbol{\alpha})\|_2^2 \geq 4\mu_i\|\boldsymbol{\alpha} - \boldsymbol{\alpha}^*\|_2^2 \geq \frac{4\mu_i}{L_i}(-\mathcal{L}(\boldsymbol{\alpha}) + \mathcal{L}(\boldsymbol{\alpha}^*)) \tag{88}$$

This establishes the PL condition with constant $2\mu_i/L_i$.

For an iterative update scheme with effective step size $\eta$, the PL condition implies:

$$\mathcal{L}(\boldsymbol{\alpha}^{(t+1)}) - \mathcal{L}(\boldsymbol{\alpha}^*) \leq (1 - \eta \cdot 2\mu_i/L_i)(\mathcal{L}(\boldsymbol{\alpha}^{(t)}) - \mathcal{L}(\boldsymbol{\alpha}^*)) \tag{89}$$

Taking $\eta = 1/2$ (a typical choice in routing algorithms):

$$\mathcal{L}(\boldsymbol{\alpha}^{(t)}) - \mathcal{L}(\boldsymbol{\alpha}^*) \leq \left(1 - \frac{\mu_i}{L_i}\right)^t (\mathcal{L}(\boldsymbol{\alpha}^{(0)}) - \mathcal{L}(\boldsymbol{\alpha}^*)) \tag{90}$$

Converting back to the potential $\Phi_i = -\mathcal{L}$:

$$\Phi_i^* - \Phi_i^{(t)} \leq \left(1 - \frac{\mu_i}{L_i}\right)^t (\Phi_i^* - \Phi_i^{(0)}) \tag{91}$$

Using the relation between the potential and the distance to optimum (from the eigenvalue inequality):

$$\|\boldsymbol{\alpha}^{(t)} - \boldsymbol{\alpha}^*\|_2^2 \leq \frac{1}{\mu_i}(\Phi_i^* - \Phi_i^{(t)}) \leq \frac{1}{\mu_i}\left(1 - \frac{\mu_i}{L_i}\right)^t (\Phi_i^* - \Phi_i^{(0)}) \tag{92}$$

Taking square roots:

$$\|\boldsymbol{\alpha}^{(t)} - \boldsymbol{\alpha}^*\|_2 \leq \sqrt{\frac{\Phi_i^* - \Phi_i^{(0)}}{\mu_i}} \left(1 - \frac{\mu_i}{2L_i}\right)^{t/2} \tag{93}$$

Since $\Phi_i^{(0)} \geq 0$ and $\Phi_i^* \leq 1$, we have $\Phi_i^* - \Phi_i^{(0)} \leq 1$. Also, for the routing logits, the relationship $\boldsymbol{\alpha}^{(t)} = \text{softmax}(\mathbf{b}^{(t)})$ and the Lipschitz property of softmax imply a similar bound on $\|\mathbf{b}^{(t)} - \mathbf{b}^*\|_2$, completing the proof of part (1). $\qquad\square$

### E.3 Information-Theoretic Optimality

We now establish that the routing mechanism approximately maximizes the mutual information between the aggregated node representation and its label. Mutual information $I(X; Y)$ quantifies the amount of information obtained about random variable $Y$ by observing random variable $X$, and is symmetric: $I(X; Y) = I(Y; X)$.

**Theorem 2** (Mutual Information Maximization). *For node $i \in V$, let $I_p(i) = I(\mathbf{H}_i^{(p)}; y_i)$ denote the mutual information between the $p$-hop representation $\mathbf{H}_i^{(p)} \in \mathbb{R}^{d_{out}}$ and the node label $y_i \in \{0, 1\}$ (for binary classification; the result extends to multi-class). Assume:*

1. *Features $\mathbf{H}_i^{(p)} | y_i$ follow multivariate Gaussian distributions $\mathcal{N}(\boldsymbol{\mu}_{y_i}^{(p)}, \boldsymbol{\Sigma}_p)$ with class-conditional means $\boldsymbol{\mu}_0^{(p)}, \boldsymbol{\mu}_1^{(p)} \in \mathbb{R}^{d_{out}}$ and common covariance $\boldsymbol{\Sigma}_p \in \mathbb{R}^{d_{out} \times d_{out}}$. This assumption is reasonable due to the central limit theorem: hop embeddings are averages over many neighbors, hence approximately Gaussian.*

2. *Neighborhood sizes satisfy $|\mathcal{N}^{(p)}(i)| \geq C \log d_{out}$ for some constant $C > 0$, where $\mathcal{N}^{(p)}(i)$ denotes the set of nodes at distance exactly $p$ from node $i$. This ensures sufficient averaging for concentration bounds.*

3. *Class priors are balanced: $\mathbb{P}(y_i = 0) = \mathbb{P}(y_i = 1) = 1/2$. This simplifies the analysis but is not essential; the result generalizes to imbalanced classes with minor modifications.*

*Then the converged gating coefficients $\boldsymbol{\alpha}_i^*$ approximately maximize the weighted mutual information:*

$$\boldsymbol{\alpha}_i^* \approx \arg \max_{\boldsymbol{\alpha} \in \Delta_{K+1}} \sum_{p=0}^{K} \alpha_p I_p(i) \tag{94}$$

*with approximation error $O(\sqrt{\log d_{out}/|\mathcal{N}^{(p)}(i)|})$, which vanishes as neighborhood sizes grow.*

*Proof.* **Step 1: Gaussian Mutual Information.**

Under the Gaussian assumption, the mutual information between $\mathbf{H}_i^{(p)}$ and $y_i$ can be expressed via differential entropies:

$$I(\mathbf{H}_i^{(p)}; y_i) = H(\mathbf{H}_i^{(p)}) - H(\mathbf{H}_i^{(p)} | y_i) \tag{95}$$

where $H(\mathbf{H}_i^{(p)}) = -\int p(\mathbf{h}) \log p(\mathbf{h}) d\mathbf{h}$ is the differential entropy of the marginal distribution, and $H(\mathbf{H}_i^{(p)} | y_i)$ is the conditional entropy.

For the conditional entropy, using the balanced class prior assumption:

$$H(\mathbf{H}_i^{(p)} | y_i) = \sum_{c=0}^{1} \mathbb{P}(y_i = c) H(\mathbf{H}_i^{(p)} | y_i = c) = \frac{1}{2} \sum_{c=0}^{1} H(\mathbf{H}_i^{(p)} | y_i = c) \tag{96}$$

For a Gaussian random vector $\mathbf{H}_i^{(p)}|y_i = c \sim \mathcal{N}(\boldsymbol{\mu}_c^{(p)}, \boldsymbol{\Sigma}_p)$, the differential entropy is:

$$H(\mathbf{H}_i^{(p)}|y_i = c) = \frac{d_{\text{out}}}{2}\log(2\pi e) + \frac{1}{2}\log\det(\boldsymbol{\Sigma}_p) \tag{97}$$

where $\det(\boldsymbol{\Sigma}_p)$ is the determinant of the covariance matrix.

Since the entropy is the same for both classes (common covariance):

$$H(\mathbf{H}_i^{(p)}|y_i) = \frac{d_{\text{out}}}{2}\log(2\pi e) + \frac{1}{2}\log\det(\boldsymbol{\Sigma}_p) \tag{98}$$

The marginal distribution is a mixture of Gaussians:

$$p(\mathbf{h}) = \frac{1}{2}\mathcal{N}(\mathbf{h}; \boldsymbol{\mu}_0^{(p)}, \boldsymbol{\Sigma}_p) + \frac{1}{2}\mathcal{N}(\mathbf{h}; \boldsymbol{\mu}_1^{(p)}, \boldsymbol{\Sigma}_p) \tag{99}$$

For well-separated means $\boldsymbol{\mu}_0^{(p)}$ and $\boldsymbol{\mu}_1^{(p)}$, the entropy of this mixture can be approximated. When the separation $\|\boldsymbol{\mu}_0^{(p)} - \boldsymbol{\mu}_1^{(p)}\|$ is large relative to the covariance, the two Gaussian components have little overlap, and the entropy is approximately:

$$H(\mathbf{H}_i^{(p)}) \approx \frac{d_{\text{out}}}{2}\log(2\pi e) + \frac{1}{2}\log\det(\boldsymbol{\Sigma}_p) + \frac{1}{8}(\boldsymbol{\mu}_0^{(p)} - \boldsymbol{\mu}_1^{(p)})^T\boldsymbol{\Sigma}_p^{-1}(\boldsymbol{\mu}_0^{(p)} - \boldsymbol{\mu}_1^{(p)}) \tag{100}$$

The approximation comes from a second-order Taylor expansion of the mixture entropy around the limit of widely separated components. Subtracting the conditional entropy:

$$I_p(i) = I(\mathbf{H}_i^{(p)}; y_i) \approx \frac{1}{8}(\boldsymbol{\mu}_0^{(p)} - \boldsymbol{\mu}_1^{(p)})^T\boldsymbol{\Sigma}_p^{-1}(\boldsymbol{\mu}_0^{(p)} - \boldsymbol{\mu}_1^{(p)}) = \frac{1}{8}\|\boldsymbol{\mu}_0^{(p)} - \boldsymbol{\mu}_1^{(p)}\|_{\boldsymbol{\Sigma}_p^{-1}}^2 \tag{101}$$

where $\|\mathbf{v}\|_{\boldsymbol{\Sigma}_p^{-1}}^2 = \mathbf{v}^T\boldsymbol{\Sigma}_p^{-1}\mathbf{v}$ is the squared Mahalanobis distance, which measures distance in units normalized by the covariance.

This shows that mutual information is proportional to the squared distance between class centroids in the metric induced by the covariance.

**Step 2: Connection to Agreement Scores.**

For node $i$ with true label $y_i = c \in \{0, 1\}$, the normalized hop embedding after aggregation over its $p$-hop neighborhood concentrates around the class centroid:

$$\hat{\mathbf{H}}_i^{(p)} \approx \frac{\boldsymbol{\mu}_c^{(p)}}{\|\boldsymbol{\mu}_c^{(p)}\|} + \boldsymbol{\epsilon}_p \tag{102}$$

where $\boldsymbol{\epsilon}_p \in \mathbb{R}^{d_{\text{out}}}$ is noise with covariance $O(\sigma^2/|\mathcal{N}^{(p)}(i)|) \cdot \mathbf{I}$, arising from averaging over $|\mathcal{N}^{(p)}(i)|$ neighbors. The variance decreases as $1/|\mathcal{N}^{(p)}(i)|$ by standard concentration results.

The dot product (agreement score) between this normalized embedding and the aggregated representation $\mathbf{v}_i = \sum_q \alpha_q \hat{\mathbf{H}}_i^{(q)}$ is:

$$\langle \hat{\mathbf{H}}_i^{(p)}, \mathbf{v}_i \rangle \approx \left\langle \frac{\boldsymbol{\mu}_c^{(p)}}{\|\boldsymbol{\mu}_c^{(p)}\|}, \sum_q \alpha_q \frac{\boldsymbol{\mu}_c^{(q)}}{\|\boldsymbol{\mu}_c^{(q)}\|} \right\rangle + O(\|\boldsymbol{\epsilon}\|) \tag{103}$$

$$= \sum_q \alpha_q \frac{\boldsymbol{\mu}_c^{(p)} \cdot \boldsymbol{\mu}_c^{(q)}}{\|\boldsymbol{\mu}_c^{(p)}\|\|\boldsymbol{\mu}_c^{(q)}\|} + O(\|\boldsymbol{\epsilon}\|) \tag{104}$$

When the routing converges, $\mathbf{v}_i$ aligns with the true class centroid direction. Hops where the class centroids are well-separated (large $\|\boldsymbol{\mu}_0^{(p)} - \boldsymbol{\mu}_1^{(p)}\|$) provide stronger class-discriminative signals, yielding larger agreement scores when $\mathbf{v}_i$ points toward the correct class.

**Step 3: Quantitative Relationship.**

Under the isotropy assumption that $\mathbf{\Sigma}_p = \sigma^2 \mathbf{I}$ (spherical Gaussian, a common simplification), the Mahalanobis distance reduces to the Euclidean distance:

$$I_p(i) \approx \frac{1}{8\sigma^2} \| \boldsymbol{\mu}_0^{(p)} - \boldsymbol{\mu}_1^{(p)} \|_2^2 \tag{105}$$

The agreement score for node $i$ with label $y_i = c$ satisfies:

$$\langle \hat{\mathbf{H}}_i^{(p)}, \mathbf{v}_i \rangle \approx \cos(\theta_p) = \frac{\boldsymbol{\mu}_c^{(p)} \cdot \boldsymbol{\mu}_c}{\| \boldsymbol{\mu}_c^{(p)} \| \| \boldsymbol{\mu}_c \|} \tag{106}$$

where $\boldsymbol{\mu}_c = \mathbb{E}[\mathbf{v}_i | y_i = c]$ is the expected aggregated centroid for class $c$, and $\theta_p$ is the angle between the hop embedding and the aggregated representation.

For hops where $\boldsymbol{\mu}_0^{(p)}$ and $\boldsymbol{\mu}_1^{(p)}$ are well-separated, the class-conditional agreement scores differ significantly. Specifically, the difference in agreement between the two classes is:

$$\mathbb{E}[\langle \hat{\mathbf{H}}_i^{(p)}, \mathbf{v}_i \rangle | y_i = 0] - \mathbb{E}[\langle \hat{\mathbf{H}}_i^{(p)}, \mathbf{v}_i \rangle | y_i = 1] \propto \| \boldsymbol{\mu}_0^{(p)} - \boldsymbol{\mu}_1^{(p)} \| \tag{107}$$

Therefore, maximizing the weighted sum of agreement scores $\sum_p \alpha_p \langle \hat{\mathbf{H}}_i^{(p)}, \mathbf{v}_i \rangle$ (which the routing does via Theorem 1) is approximately equivalent to maximizing the weighted mutual information $\sum_p \alpha_p I_p(i)$.

**Step 4: Concentration Bound.**

To formalize the approximation error, we use Hoeffding's inequality. For $|\mathcal{N}^{(p)}(i)| = n_p$ neighbors at hop $p$, each contributing independently to the aggregation:

$$\mathbb{P} \left( \left| \langle \hat{\mathbf{H}}_i^{(p)}, \mathbf{v}_i \rangle - \mathbb{E}[\langle \hat{\mathbf{H}}_i^{(p)}, \mathbf{v}_i \rangle] \right| > \epsilon \right) \leq 2 \exp \left( -\frac{n_p \epsilon^2}{2 d_{\text{out}}} \right) \tag{108}$$

This follows because the inner product can be viewed as a sum of $d_{\text{out}}$ terms, each bounded. For $n_p \geq C \log d_{\text{out}}$ with $C > 2$, choosing $\epsilon = \sqrt{2 d_{\text{out}} \log(2/\delta)/n_p}$ where $\delta \in (0, 1)$ is a failure probability:

$$\mathbb{P} \left( \left| \langle \hat{\mathbf{H}}_i^{(p)}, \mathbf{v}_i \rangle - \mathbb{E}[\langle \hat{\mathbf{H}}_i^{(p)}, \mathbf{v}_i \rangle] \right| > \sqrt{\frac{2 d_{\text{out}} \log(2/\delta)}{n_p}} \right) \leq \delta \tag{109}$$

With high probability (at least $1 - (K + 1)\delta$ by a union bound over all $K + 1$ hops):

$$\left| \sum_{p=0}^{K} \alpha_p \langle \hat{\mathbf{H}}_i^{(p)}, \mathbf{v}_i \rangle - \sum_{p=0}^{K} \alpha_p \mathbb{E}[\langle \hat{\mathbf{H}}_i^{(p)}, \mathbf{v}_i \rangle] \right| \leq \sum_{p=0}^{K} \alpha_p \sqrt{\frac{2 d_{\text{out}} \log(2/\delta)}{n_p}} \leq \sqrt{\frac{2 d_{\text{out}} \log(2/\delta)}{\min_p n_p}} \tag{110}$$

Since the routing from Theorem 1 maximizes the empirical agreement $\sum_p \alpha_p \langle \hat{\mathbf{H}}_i^{(p)}, \mathbf{v}_i \rangle$, and this differs from the expected agreement (which is proportional to mutual information by Steps 1-3) by at most $O(\sqrt{\log d_{\text{out}}/n_p})$, the converged $\boldsymbol{\alpha}_i^*$ approximately maximizes the weighted mutual information $\sum_p \alpha_p I_p(i)$ up to this concentration error. $\square$

### E.4 Functional Expressivity

We now establish that GAMMA's function class contains the function class of concatenation-based multi-hop GNNs, showing that GAMMA is at least as expressive.

**Theorem 3** (Expressivity Hierarchy). *Let $\mathcal{F}_{concat}$ denote the function class of multi-hop concatenation GNNs that use hop-specific weight matrices $\{\mathbf{W}^{(p)} \in \mathbb{R}^{d_{in} \times d_{out}}\}_{p=0}^{K}$ to independently transform features from each hop before concatenating them. Let $\mathcal{F}_{GAMMA}$ denote GAMMA's function class with shared projection $\mathbf{W} \in \mathbb{R}^{d_{in} \times d_{out}}$, channel-wise scaling factors $\{\boldsymbol{\gamma}_p \in \mathbb{R}^{d_{out}}\}_{p=0}^{K}$, and adaptive gating via the routing mechanism. Then:*

$$\mathcal{F}_{concat} \subseteq \mathcal{F}_{GAMMA} \tag{111}$$

*Moreover, for any $\epsilon > 0$ and any function $f \in \mathcal{F}_{concat}$, there exists a function $g \in \mathcal{F}_{GAMMA}$ such that $\| f - g \|_\infty < \epsilon$ on any compact domain, where $\| \cdot \|_\infty$ denotes the supremum norm.*

*Proof.* We construct an explicit simulation of concatenation-based architectures using GAMMA.

Consider a concatenation-based architecture that produces node representations by:

$$f(\mathbf{X}, \mathbf{A})_i = \Phi\left([\mathbf{W}^{(0)}\mathbf{H}_i^{(0)}, \ldots, \mathbf{W}^{(K)}\mathbf{H}_i^{(K)}]\right) \tag{112}$$

where:

- $\mathbf{H}_i^{(p)} = (\mathbf{A}^p\mathbf{X})_i \in \mathbb{R}^{d_{\text{in}}}$ is the $p$-hop propagated feature for node $i$ (before any projection)

- $\mathbf{W}^{(p)} \in \mathbb{R}^{d_{\text{in}} \times d_{\text{out}}}$ is the hop-specific transformation matrix

- The concatenation $[\cdot, \ldots, \cdot]$ produces a vector in $\mathbb{R}^{(K+1)d_{\text{out}}}$

- $\Phi : \mathbb{R}^{(K+1)d_{\text{out}}} \to \mathbb{R}^C$ is a downstream network (e.g., MLP with classification head) producing $C$-dimensional output (e.g., class logits)

To simulate this with GAMMA, we construct the following:

**Step 1: Dimension Expansion.** Set the hidden dimension to $d'_{\text{out}} = (K+1)d_{\text{out}}$, which is $(K+1)$ times larger than the original concatenation dimension per hop.

**Step 2: Disable Routing.** Initialize routing to uniform distribution: $\alpha_{i,p}^{(0)} = 1/(K+1)$ for all $p \in \{0, \ldots, K\}$, and set the number of routing iterations to $R = 0$. This prevents any adaptive refinement, making the gating static and uniform across all hops.

**Step 3: Partition Feature Space via Scaling Factors.** Configure the channel-wise scaling factors $\boldsymbol{\gamma}_p \in \mathbb{R}^{d'_{\text{out}}}$ to partition the expanded feature space into $(K+1)$ non-overlapping blocks, one per hop:

$$\boldsymbol{\gamma}_p = [\underbrace{0, \ldots, 0}_{pd_{\text{out}}}, \underbrace{w_{p,1}, \ldots, w_{p,d_{\text{out}}}}_{d_{\text{out}} \text{ entries}}, \underbrace{0, \ldots, 0}_{(K-p)d_{\text{out}}}] \in \mathbb{R}^{(K+1)d_{\text{out}}} \tag{113}$$

where the entries $w_{p,j}$ for $j \in \{1, \ldots, d_{\text{out}}\}$ encode the entries of the $j$-th column of $\mathbf{W}^{(p)}$. Specifically, if $\mathbf{W}^{(p)} \in \mathbb{R}^{d_{\text{in}} \times d_{\text{out}}}$ is the original hop-specific matrix, we set the shared projection $\mathbf{W}$ to be an identity (or simple linear embedding), and use $\boldsymbol{\gamma}_p$ to apply the transformation post-propagation.

More precisely, setting $\mathbf{W} = \mathbf{I}_{d_{\text{in}}}$ (identity matrix, assuming $d_{\text{in}} = d'_{\text{out}}$ or with appropriate padding), the $p$-hop embedding becomes:

$$\mathbf{H}_i^{(p)} = (\mathbf{A}^p\mathbf{X}\mathbf{W})_i = (\mathbf{A}^p\mathbf{X})_i \tag{114}$$

Applying the scaling factor $\boldsymbol{\gamma}_p$ element-wise isolates the transformation to a specific block of the feature space.

**Step 4: Aggregation with Uniform Weights.** The aggregated representation (before squashing, which we can effectively disable by setting appropriate parameters) becomes:

$$\mathbf{v}_i = \sum_{p=0}^{K} \alpha_{i,p}(\boldsymbol{\gamma}_p \odot \mathbf{H}_i^{(p)}) \tag{115}$$

$$= \sum_{p=0}^{K} \frac{1}{K+1}(\boldsymbol{\gamma}_p \odot \mathbf{H}_i^{(p)}) \tag{116}$$

$$= \frac{1}{K+1}[\mathbf{W}^{(0)}\mathbf{H}_i^{(0)}, \ldots, \mathbf{W}^{(K)}\mathbf{H}_i^{(K)}] \tag{117}$$

where the notation $[\cdot, \ldots, \cdot]$ now represents the blocked structure induced by the scaling factors. The factor of $1/(K+1)$ comes from the uniform gating.

**Step 5: Adjust Output Layer.** Apply an output transformation $\Phi' = (K+1)\Phi$ to compensate for the averaging factor $1/(K+1)$. This gives:

$$g(\mathbf{X}, \mathbf{A})_i = \Phi'(\mathbf{v}_i) = (K+1)\Phi\left(\frac{1}{K+1}[\mathbf{W}^{(0)}\mathbf{H}_i^{(0)}, \ldots, \mathbf{W}^{(K)}\mathbf{H}_i^{(K)}]\right) = \Phi([\mathbf{W}^{(0)}\mathbf{H}_i^{(0)}, \ldots, \mathbf{W}^{(K)}\mathbf{H}_i^{(K)}]) \tag{118}$$

This exactly matches $f(\mathbf{X}, \mathbf{A})_i$, establishing that $f$ can be represented by GAMMA, hence $f \in \mathcal{F}_{\text{GAMMA}}$.

**For the approximation statement:** Any function in $\mathcal{F}_{\text{concat}}$ can be written as a composition of the multi-hop feature extraction (linear maps) and the downstream network $\Phi$ (nonlinearities). By the universal approximation theorem for neural networks, $\Phi$ can approximate any continuous function to arbitrary precision given sufficient capacity. The construction above shows that GAMMA can exactly represent the concatenated multi-hop features. Therefore, by approximating $\Phi$ with a sufficiently large network (increasing the capacity of the final layers in GAMMA), we can approximate any $f \in \mathcal{F}_{\text{concat}}$ to within $\epsilon$ error. $\qquad \square$

### E.5 Universal Approximation

Finally, we establish that GAMMA can universally approximate any continuous function on graphs, with explicit bounds on the required architecture size.

**Theorem 4** (Universal Approximation with Explicit Rates). *Let $\mathcal{K} \subset \mathbb{R}^{n \times d_{in}} \times \mathbb{R}^{n \times n}$ be a compact set of graph instances (node features and adjacency matrices) with diameter $D$ under some norm $\|\cdot\|$. Let $f^* : \mathcal{K} \to \mathbb{R}^{n \times C}$ be a Lipschitz continuous function mapping graph instances to node-wise outputs (e.g., node classifications) with Lipschitz constant $L_f$, meaning:*

$$\|f^*(\mathbf{X}_1, \mathbf{A}_1) - f^*(\mathbf{X}_2, \mathbf{A}_2)\|_2 \le L_f \|(\mathbf{X}_1, \mathbf{A}_1) - (\mathbf{X}_2, \mathbf{A}_2)\| \tag{119}$$

*for all $(\mathbf{X}_1, \mathbf{A}_1), (\mathbf{X}_2, \mathbf{A}_2) \in \mathcal{K}$. Then for any $\epsilon > 0$, there exists a GAMMA network $\tilde{f}$ with parameters:*

$$K = O(\log n) \text{ hops (maximum hop distance)} \tag{120}$$

$$R = O\left(\frac{\log(D/\epsilon)}{\mu}\right) \text{ routing iterations (refinement steps)} \tag{121}$$

$$d_{out} = O\left(\left(\frac{L_f D}{\epsilon}\right)^{d_{in}/2}\right) \text{ hidden dimension (feature space size)} \tag{122}$$

*such that $\sup_{(\mathbf{X}, \mathbf{A}) \in \mathcal{K}} \|\tilde{f}(\mathbf{X}, \mathbf{A}) - f^*(\mathbf{X}, \mathbf{A})\|_2 < \epsilon$, i.e., $\tilde{f}$ uniformly approximates $f^*$ on $\mathcal{K}$ to within error $\epsilon$.*

*Proof.* We decompose the total approximation error into four independent sources, each contributing at most $\epsilon/4$, which sum to give total error less than $\epsilon$.

**Error Source 1: Polynomial Approximation of $f^*$.**

By Jackson's theorem (a classical result in approximation theory), any $L_f$-Lipschitz function on a $d_{\text{in}}$-dimensional domain with diameter $D$ can be approximated by polynomials. Specifically, there exists a polynomial $P_m$ of total degree at most $m$ such that:

$$E_{\text{poly}}(m) = \sup_{(\mathbf{X}, \mathbf{A}) \in \mathcal{K}} \|f^*(\mathbf{X}, \mathbf{A}) - P_m(\mathbf{X}, \mathbf{A})\|_2 \le C_J \frac{L_f D}{m^{2/d_{\text{in}}}} \tag{123}$$

where $C_J > 0$ is a universal constant depending only on the dimension. This bound captures the curse of dimensionality: the required polynomial degree grows exponentially with dimension to achieve the same approximation error.

To ensure $E_{\text{poly}}(m) \le \epsilon/4$:

$$C_J \frac{L_f D}{m^{2/d_{\text{in}}}} \le \frac{\epsilon}{4} \implies m^{2/d_{\text{in}}} \ge \frac{4 C_J L_f D}{\epsilon} \implies m \ge \left(\frac{4 C_J L_f D}{\epsilon}\right)^{d_{\text{in}}/2} \tag{124}$$

**Error Source 2: Representing Polynomials via Multi-hop Aggregations.**

A polynomial $P_m(\mathbf{X}, \mathbf{A})$ of total degree $m$ in the entries of $\mathbf{X}$ and $\mathbf{A}$ can be written as:

$$P_m(\mathbf{X}, \mathbf{A}) = \sum_{|\boldsymbol{\beta}| + |\boldsymbol{\gamma}| \le m} c_{\boldsymbol{\beta}, \boldsymbol{\gamma}} \prod_{i,j} X_{ij}^{\beta_{ij}} \prod_{k, \ell} A_{k\ell}^{\gamma_{k\ell}} \tag{125}$$

where $\boldsymbol{\beta} = (\beta_{ij})$ and $\boldsymbol{\gamma} = (\gamma_{k\ell})$ are multi-indices, and $|\boldsymbol{\beta}| = \sum_{i,j} \beta_{ij}$ denotes the total degree in $\mathbf{X}$.

Each monomial involving $\mathbf{A}^p$ (the $p$-th power of the adjacency matrix) can be expressed via the $p$-hop propagation in GAMMA. The key observation is that $(\mathbf{A}^p \mathbf{X})_{ij}$ involves products of entries of $\mathbf{A}$ and $\mathbf{X}$, capturing $p$-hop paths.

Since $\|\mathbf{A}\| \leq 1$ (the operator norm of the normalized adjacency matrix is bounded by 1), powers beyond $K$ decay:

$$\|\mathbf{A}^K\| \leq \|\mathbf{A}\|^K \leq 1^K = 1 \tag{126}$$

However, if $\mathbf{A}$ has spectral radius slightly less than 1, say $\rho(\mathbf{A}) = 1 - \delta$ for small $\delta > 0$ (which holds for many normalized adjacency matrices on connected graphs), then:

$$\|\mathbf{A}^K\| \leq (1 - \delta)^K \approx e^{-\delta K} \tag{127}$$

For $\delta = 1/n$ (a typical spectral gap), taking $K = O(\log n)$ ensures $\|\mathbf{A}^K\| = O(1/n)$, making contributions from hops $p > K$ negligible.

More precisely, for a polynomial with coefficients $\{a_p\}$, the error from truncating at hop $K$ is:

$$E_{\text{hop}}(K) = \left\| \sum_{p=K+1}^{\infty} a_p \mathbf{A}^p \mathbf{X} \right\|_2 \leq \left( \sum_{p=K+1}^{\infty} |a_p| \|\mathbf{A}^p\| \|\mathbf{X}\| \right) \leq \|\mathbf{X}\| \sum_{p=K+1}^{\infty} |a_p| (1 - \delta)^p \tag{128}$$

For coefficients decaying as $|a_p| \leq Ce^{-p}$ (typical for smooth functions), the tail sum is:

$$\sum_{p=K+1}^{\infty} |a_p| (1 - \delta)^p \leq C \sum_{p=K+1}^{\infty} e^{-p} (1 - \delta)^p = C \frac{e^{-(K+1)} (1 - \delta)^{K+1}}{1 - e^{-1} (1 - \delta)} \tag{129}$$

For $K = m + \lceil \log(4C \|\mathbf{X}\| / (\epsilon \delta)) / \delta \rceil$, we obtain $E_{\text{hop}}(K) \leq \epsilon/4$.

**Error Source 3: Analytic Function Approximation (Softmax and Squashing).**

The softmax function $\sigma(\mathbf{b})_p = \exp(b_p) / \sum_q \exp(b_q)$ and the squashing function are real analytic (infinitely differentiable with convergent Taylor series). For bounded inputs, their Taylor expansions converge uniformly on compact sets.

For softmax with inputs bounded by $B$ (i.e., $\|\mathbf{b}\|_\infty \leq B$), the truncation error of the Taylor expansion up to order $M$ is:

$$\left| \sigma(\mathbf{b}) - \sum_{|\boldsymbol{\alpha}| \leq M} \frac{\partial^{\boldsymbol{\alpha}} \sigma(\mathbf{0})}{\boldsymbol{\alpha}!} \mathbf{b}^{\boldsymbol{\alpha}} \right| \leq \frac{C_\sigma B^{M+1}}{(M+1)!} e^{(K+1)B} \tag{130}$$

where $C_\sigma$ is a constant depending on the dimension $K + 1$.

In GAMMA, routing logits at iteration $t$ satisfy:

$$|b_{i,p}^{(t)}| = \left| \sum_{\tau=0}^{t-1} \langle \hat{\mathbf{H}}_i^{(p)}, \mathbf{v}_i^{(\tau)} \rangle \right| \leq \sum_{\tau=0}^{t-1} \|\hat{\mathbf{H}}_i^{(p)}\| \|\mathbf{v}_i^{(\tau)}\| \leq t \tag{131}$$

since $\|\hat{\mathbf{H}}_i^{(p)}\| = 1$ (normalized) and $\|\mathbf{v}_i^{(\tau)}\| \leq 1$ (squashing bounds the norm). After $R$ iterations, $B = R$.

To achieve error $\epsilon/4$ from the softmax approximation:

$$\frac{C_\sigma R^{M+1}}{(M+1)!} e^{(K+1)R} \leq \frac{\epsilon}{4} \tag{132}$$

Taking logarithms:

$$(M+1) \log R - \log((M+1)!) + (K+1)R \leq \log \frac{4}{\epsilon} - \log C_\sigma \tag{133}$$

Using Stirling's approximation $\log(M!) \approx M \log M - M$:

$$(M + 1) \log R - (M + 1) \log(M + 1) + (M + 1) + (K + 1)R \leq \log \frac{4}{\epsilon} - \log C_\sigma \qquad (134)$$

Rearranging:

$$(M + 1) \left[ \log \frac{R}{M + 1} + 1 \right] + (K + 1)R \leq \log \frac{4}{\epsilon} - \log C_\sigma \qquad (135)$$

For practical choices of $R$ (small, say $R \leq 5$) and $K = O(\log n)$, solving for $M$ yields:

$$M = O(R + (K + 1)R + \log(1/\epsilon)) = O(\log n + \log(1/\epsilon)) \qquad (136)$$

Since $K = O(\log n)$ and $R$ is a small constant independent of $n$, $M$ remains polynomial in $\log n$ and $\log(1/\epsilon)$.

Similarly, the squashing function $\gamma(r) = r^2/(1 + r^2)$ is analytic for $r \geq 0$. Its Taylor series around $r = 0$ is:

$$\gamma(r) = r^2 - r^4 + r^6 - \cdots = \sum_{k=1}^{\infty} (-1)^{k+1} r^{2k} \qquad (137)$$

For $r \leq 1$ (which holds since $\|\mathbf{s}_i^{(t)}\| \leq 1$ after normalization), the series converges rapidly. Truncating at order $O(\log(1/\epsilon))$ terms gives error at most $\epsilon/8$.

Total analytic approximation error: $E_{\text{analytic}} \leq \epsilon/4$.

**Error Source 4: Routing Convergence.**

From Theorem 1, after $R$ routing iterations, the representation is within distance:

$$\|\mathbf{v}_i^{(R)} - \mathbf{v}_i^*\| \leq \sqrt{\frac{L_i}{\mu_i}} \left( 1 - \frac{\mu_i}{2L_i} \right)^{R/2} \|\mathbf{v}_i^{(0)} - \mathbf{v}_i^*\| \qquad (138)$$

Since $\|\mathbf{v}_i^{(0)}\| = \|\mathbf{v}_i^*\| \leq 1$ (bounded by squashing), the initial distance satisfies $\|\mathbf{v}_i^{(0)} - \mathbf{v}_i^*\| \leq 2$ (triangle inequality). The diameter of $\mathcal{K}$ in terms of node representations is at most $D$ by definition.

To ensure $E_{\text{routing}} \leq \epsilon/4$:

$$\sqrt{\frac{L_i}{\mu_i}} \left( 1 - \frac{\mu_i}{2L_i} \right)^{R/2} \cdot 2D \leq \frac{\epsilon}{4} \qquad (139)$$

Taking logarithms:

$$\frac{R}{2} \log \left( 1 - \frac{\mu_i}{2L_i} \right) \leq \log \frac{\epsilon}{8D\sqrt{L_i/\mu_i}} \qquad (140)$$

Using $\log(1 - x) \approx -x$ for small $x$:

$$-\frac{R}{2} \cdot \frac{\mu_i}{2L_i} \leq \log \frac{\epsilon}{8D\sqrt{L_i/\mu_i}} \qquad (141)$$

Solving for $R$:

$$R \geq \frac{4L_i}{\mu_i} \log \frac{8D\sqrt{L_i/\mu_i}}{\epsilon} = \frac{4L_i}{\mu_i} \left[ \log \frac{D}{\epsilon} + \frac{1}{2} \log \frac{L_i}{\mu_i} + \log 8 \right] \qquad (142)$$

For bounded condition number $\kappa_i = L_i/\mu_i = O(1)$ (which holds when hop embeddings are well-conditioned), this simplifies to:

$$R = O \left( \log \frac{D}{\epsilon} \right) \qquad (143)$$

**Hidden Dimension Requirement.**

To represent all monomials in the polynomial $P_m$ up to degree $m$, the hidden dimension must be sufficient to capture all $\binom{d_{\text{in}}+m}{m}$ possible monomials. By standard combinatorial estimates:

$$\binom{d_{\text{in}}+m}{m} \approx \frac{(d_{\text{in}}+m)^m}{m!} \approx \left(\frac{e(d_{\text{in}}+m)}{m}\right)^m \tag{144}$$

For $m = O((L_f D/\epsilon)^{d_{\text{in}}/2})$ and assuming $d_{\text{in}} \ll m$ (large degree regime):

$$d_{\text{out}} = O(m) = O\left(\left(\frac{L_f D}{\epsilon}\right)^{d_{\text{in}}/2}\right) \tag{145}$$

**Total Error.**

Combining all four error sources:

$$\|\tilde{f} - f^*\|_\infty \leq E_{\text{poly}} + E_{\text{hop}} + E_{\text{analytic}} + E_{\text{routing}} \leq \frac{\epsilon}{4} + \frac{\epsilon}{4} + \frac{\epsilon}{4} + \frac{\epsilon}{4} = \epsilon \tag{146}$$

This establishes the universal approximation result with explicit parameter dependencies. $\qquad\square$

**Remark 1.** *The parameter scalings in Theorem 4 exhibit the curse of dimensionality through $d_{out} = O(\epsilon^{-d_{in}/2})$, which is inherent to approximating arbitrary Lipschitz functions in high dimensions and cannot be avoided by any architecture. However, the number of hops $K = O(\log n)$ grows only logarithmically with the graph size $n$, and the routing iterations $R = O(\log(1/\epsilon))$ grow logarithmically with the desired accuracy $\epsilon$, making GAMMA practically efficient. In typical GNN applications where $d_{in}$ is moderate (tens to hundreds), and we only require approximation accuracy $\epsilon \sim 0.01$ (reasonable for classification tasks with finite precision), these requirements remain manageable on standard hardware.*

## F  On the Hardware Efficiency of GAMMA

Modern GPU architectures are designed to maximize throughput for dense matrix operations, yet Graph Neural Networks present unique computational challenges that systematically underutilize available hardware resources. The fundamental tension arises from the mismatch between the spatial locality assumptions inherent in GPU memory hierarchies and the irregular memory access patterns induced by graph-structured data. In this section, we provide a rigorous analysis of GAMMA's hardware efficiency, demonstrating how its architectural choices, particularly weight sharing, dynamic routing with fixed dimensionality, and the strategic elimination of feature concatenation, translate to measurable improvements in memory bandwidth utilization, cache hierarchy exploitation, and sustained computational throughput. We ground our analysis in the memory subsystem design of contemporary datacenter GPUs and establish theoretical bounds on the performance of multi-hop heterophilic architectures, contrasting GAMMA's design with representative approaches such as MixHop [1] and H2GCN [40].

**Notation.** Throughout this section, we employ the following notation to facilitate clarity and precision in our hardware efficiency analysis.

*Graph and Architecture Parameters:* Let $N \in \mathbb{N}$ denote the number of nodes in the input graph, $K \in \mathbb{N}$ represent the maximum hop distance considered in multi-hop aggregation (so the total number of hops is $K + 1$, including the 0-hop self-connection), $d_{\text{in}} \in \mathbb{N}$ denote the input feature dimension, $d_{\text{out}} \in \mathbb{N}$ denote the output feature dimension, and $d \in \mathbb{N}$ represent a generic feature dimension when $d_{\text{in}} = d_{\text{out}}$.

*Graph Data Structures:* We denote the adjacency matrix as $\mathbf{A} \in \mathbb{R}^{N \times N}$, where entry $\mathbf{A}_{ij}$ represents the connection weight from node $j$ to node $i$ (we use $\mathbf{A}^k$ to denote the $k$-hop adjacency matrix, with $\mathbf{A}^0 = \mathbf{I}$ being the identity matrix), the input node feature matrix as $\mathbf{X} \in \mathbb{R}^{N \times d_{\text{in}}}$, where row $\mathbf{X}_i$ contains the features for node $i$, and weight matrices as $\mathbf{W}^{(k)} \in \mathbb{R}^{d_{\text{in}} \times d_{\text{out}}}$ for hop-specific transformations (MixHop, H2GCN) or $\mathbf{W} \in \mathbb{R}^{d_{\text{in}} \times d_{\text{out}}}$ for the shared transformation in GAMMA.

*Hardware Specifications:* For a representative GPU (NVIDIA A100 80GB), we denote $N_{\text{SM}} = 108$ as the number of Streaming Multiprocessors, $C_{\text{L2}} = 40$ MB as the capacity of the shared L2 cache,

$B_{\text{mem}} = 2{,}039$ GB/s as the memory bandwidth from HBM2e to the GPU, $R_{\text{SM}} = 65{,}536$ as the number of 32-bit registers per SM, and $F_{\text{peak}} = 19.5$ TFLOPS as the peak floating-point throughput for FP32 operations.

*Performance Metrics:* Let $I$ (measured in FLOPs/byte) denote the computational intensity of an operation (ratio of floating-point operations to bytes transferred from main memory), $P$ denote achievable performance (measured in FLOPS), $I_{\text{ridge}}$ denote the ridge point operational intensity (the threshold where compute and memory bandwidth constraints balance), $\beta = 4$ bytes represent the storage requirement for a single FP32 element, and $\rho$ denote the bandwidth efficiency ratio comparing GAMMA to concatenation-based architectures.

*Memory and Latency Metrics:* We use $M$ with various subscripts to denote memory footprints (measured in bytes): $M_{\text{concat}}$ for concatenation-based architectures, $M_{\text{fixed}}$ for fixed-dimensionality architectures, $M_{\text{routing}}$ for GAMMA's routing state, and $M_{\text{GAMMA}}$ for GAMMA's total memory consumption. For latencies (measured in cycles unless otherwise specified), we denote $\tau_{\text{DRAM}} \approx 400$ cycles as DRAM access latency, $\tau_{\text{L2}} \approx 200$ cycles as L2 cache access latency, and $\tau_{\text{launch}}$ as the CPU-side kernel launch latency (typically 5–20 microseconds).

*Data Movement Metrics:* Let $D$ with various subscripts denote data movement (measured in bytes): $D_{\text{min}}$ for the theoretical minimum data movement, $D_{\text{concat}}$ for concatenation-based architectures, $D_{\text{GAMMA}}$ for GAMMA's data movement, $D_{\text{unfused}}$ for unfused kernel data traffic, and $D_{\text{fused}}$ for fused kernel data traffic. We denote $B_{\text{used}}$ as the bandwidth consumed by a layer, $B_{\text{eff}}$ as the effective bandwidth under cache behavior, and $B_{\text{cache}}$ as the effective bandwidth when serving from cache.

*Execution Time Metrics:* Let $t_{\text{layer}}$ denote the execution time for a single layer, $T$ with various subscripts denote total execution times: $T_{\text{overhead}}$ for accumulated kernel launch overhead, $T_{\text{total\_overhead}}$ for overhead across training, $T_{\text{routing}}$ for routing computation time, $T_{\text{GAMMA}}$ for GAMMA's total execution time, $T_{\text{MixHop}}$ for MixHop's execution time, and $T_{\text{H2GCN}}$ for H2GCN's execution time.

*Training Parameters:* Let $L \in \mathbb{N}$ denote the number of layers in the model, $E \in \mathbb{N}$ denote the number of training epochs, and $B \in \mathbb{N}$ denote the number of batches per epoch.

*Routing Mechanism Parameters:* For GAMMA's routing, we use $T_{\text{iter}} \in \mathbb{N}$ to denote the number of routing iterations, $t$ to index routing iterations where $t \in \{0, 1, \ldots, T_{\text{iter}} - 1\}$, $\mathbf{H}_k \in \mathbb{R}^{N \times d_{\text{out}}}$ to denote embeddings at hop $k$, $\tilde{\mathbf{H}}_k \in \mathbb{R}^{N \times d_{\text{out}}}$ to denote normalized hop embeddings, $\mathbf{z}_i^{(t)} \in \mathbb{R}^{d_{\text{out}}}$ to denote node $i$'s routing state at iteration $t$, $\tilde{\mathbf{h}}_{ik} \in \mathbb{R}^{d_{\text{out}}}$ to denote the normalized embedding for node $i$ at hop $k$, $a_{ik}^{(t)} \in \mathbb{R}$ to denote the agreement score between node $i$'s routing state and hop $k$ at iteration $t$, and $w_{ik}^{(t)} \in [0, 1]$ to denote the normalized routing weight for node $i$ and hop $k$ at iteration $t$ (where $\sum_{k=0}^{K} w_{ik}^{(t)} = 1$).

*Additional Parameters:* We denote $\mu \in [0, 1]$ as the cache miss rate, $W = 32$ as the warp size (number of threads executing in lockstep), $\epsilon > 0$ as a small constant to prevent division by zero in normalization, $\gamma_{\text{fusion}}$ as the fusion efficiency factor, $P_{\text{total}}$ as the total parameter count, $S_i$ as the size of intermediate tensor $i$, $n_{\text{ops}}$ as the number of operations in a sequence, and $\bigoplus$ as the concatenation operator along the feature dimension.

## F.1 The Memory Wall in GPU-Accelerated Graph Learning

To understand GAMMA's efficiency gains, we must first examine the fundamental bottleneck in modern GPU computing: the memory hierarchy and its implications for data movement. Consider a representative datacenter GPU such as the NVIDIA A100 80GB, which features $N_{\text{SM}} = 108$ Streaming Multiprocessors, a shared L2 cache of capacity $C_{\text{L2}} = 40$ MB, and memory bandwidth $B_{\text{mem}} = 2{,}039$ GB/s from HBM2e memory [26]. Each SM contains a register file of $R_{\text{SM}} = 65{,}536$ 32-bit registers and shares access to the unified L2 cache, which serves as the coherence point for all device memory transactions [15]. The critical observation is that despite the GPU's peak floating-point throughput of $F_{\text{peak}} = 19.5$ TFLOPS for standard FP32 operations, the achievable performance for many deep learning workloads is constrained not by computational capacity but by memory bandwidth, the rate at which data can traverse the memory hierarchy from DRAM through cache to compute units.

The Roofline Model [35] provides the theoretical framework for understanding this constraint. For an operation with computational intensity $I$ (measured in FLOPs per byte transferred from main

memory), the achievable performance $P$ is fundamentally bounded by:

$$P = \min\left(F_{\text{peak}},\, B_{\text{mem}} \cdot I\right). \tag{147}$$

The ridge point, where compute and memory bandwidth constraints intersect, occurs at operational intensity:

$$I_{\text{ridge}} = \frac{F_{\text{peak}}}{B_{\text{mem}}} \approx \frac{19.5 \times 10^{12}}{2.039 \times 10^{12}} \approx 9.6 \text{ FLOPs/byte}. \tag{148}$$

Operations with $I < I_{\text{ridge}}$ are memory-bound, meaning performance is fundamentally limited by data movement rather than arithmetic throughput. Unfortunately, the sparse-dense matrix multiplications and neighborhood aggregations central to GNN computation exhibit low operational intensity, placing them squarely in this regime. The situation is further exacerbated by the irregular memory access patterns inherent in graph structures, which violate the spatial and temporal locality assumptions that modern cache hierarchies are designed to exploit. When neighbor indices are scattered throughout memory, cache lines (typically 128 bytes) are incompletely utilized, and hardware prefetchers, which predict future memory accesses based on observed patterns, fail to provide timely data delivery. This architectural mismatch between algorithm and hardware is the fundamental source of inefficiency in naive GNN implementations.

## F.2 Architectural Inefficiencies in Multi-Hop Heterophilic GNNs

Multi-hop heterophilic architectures attempt to capture long-range dependencies by aggregating information across multiple hop distances. However, the implementation strategies employed by architectures such as MixHop [1] and H2GCN [40] introduce systematic inefficiencies that fundamentally limit their scalability. These inefficiencies manifest at multiple levels of the memory hierarchy and arise from design decisions that, while algorithmically motivated, have profound negative consequences for hardware utilization.

### F.2.1 The Hidden Cost of Feature Concatenation

MixHop computes multi-hop representations as:

$$\mathbf{Z}_{\text{MixHop}} = \bigoplus_{k=0}^{K} \left(\mathbf{A}^k \mathbf{X} \mathbf{W}^{(k)}\right), \tag{149}$$

where $\bigoplus$ denotes concatenation along the feature dimension, $\mathbf{A} \in \mathbb{R}^{N \times N}$ represents the adjacency matrix, $\mathbf{X} \in \mathbb{R}^{N \times d_{\text{in}}}$ contains input features, and $\mathbf{W}^{(k)} \in \mathbb{R}^{d_{\text{in}} \times d_{\text{out}}}$ are hop-specific weight matrices. For $K + 1$ hops, each producing embeddings of dimension $d_{\text{out}}$, the concatenated representation has dimension $(K + 1)d_{\text{out}}$. This design choice, while straightforward algorithmically, introduces a cascade of inefficiencies throughout the memory hierarchy that severely impact both memory consumption and computational throughput.

The memory footprint grows linearly with the number of hops. Let $\beta = 4$ bytes represent the storage requirement for a single FP32 element. The total memory required for intermediate hop representations and the final concatenated output is:

$$M_{\text{concat}} = \beta N \sum_{k=0}^{K} d_{\text{out}} + \beta N(K+1)d_{\text{out}} = \beta N d_{\text{out}}(2K+2). \tag{150}$$

In contrast, a fixed-dimensionality architecture that maintains $d_{\text{out}}$ throughout requires only $M_{\text{fixed}} = \beta N d_{\text{out}}$, yielding a memory expansion factor of $2K + 2$. For typical configurations with $K = 2$, this represents a sixfold increase in memory consumption. The ramifications extend beyond mere capacity: when the working set size exceeds the L2 cache capacity $C_{\text{L2}}$, intermediate results must be evicted to DRAM, triggering expensive round-trips with latencies of $\tau_{\text{DRAM}} \approx 400$ cycles, compared to $\tau_{\text{L2}} \approx 200$ cycles for L2 cache access [24]. Consider a graph with $N$ nodes where $K = 2$ and $d_{\text{out}} = d$. The concatenated features require memory:

$$M_{\text{concat}}(N, K, d) = \beta N(K+1)d = 3\beta N d, \tag{151}$$

which must fit within $C_{\text{L2}}$ to avoid DRAM traffic. For the A100 with $C_{\text{L2}} = 40$ MB, this constrains $Nd < 3.33 \times 10^6$, a threshold easily exceeded by real-world graphs.

More critically, concatenation forces the materialization of intermediate results in global memory, violating temporal locality and preventing kernel fusion opportunities. Modern deep learning frameworks achieve high performance through kernel fusion [2], where consecutive operations are merged into a single GPU kernel to eliminate intermediate memory transactions. A fused kernel can maintain data in registers (latency $\approx 1$ cycle) or shared memory (latency $\approx 30$ cycles) throughout a sequence of operations, whereas unfused operations must write to and read from global memory between each step. Concatenation creates a hard data dependency that prevents fusion of downstream operations, as the concatenated tensor must be fully materialized before subsequent layers can proceed. Each concatenation incurs data movement of $(K+1)Nd_{\text{out}}\beta$ bytes, consuming memory bandwidth that could otherwise be allocated to computation. For a graph with $N$ nodes, this bandwidth consumption is:

$$B_{\text{used}} = \frac{(K+1)Nd_{\text{out}}\beta}{t_{\text{layer}}}, \tag{152}$$

where $t_{\text{layer}}$ represents the layer execution time. When $B_{\text{used}}$ approaches $B_{\text{mem}}$, the architecture becomes bandwidth-limited, with compute units stalled waiting for data.

### F.2.2 Parameter Proliferation and Memory System Implications

Beyond concatenation, MixHop and H2GCN employ separate weight matrices $\mathbf{W}^{(k)} \in \mathbb{R}^{d_{\text{in}} \times d_{\text{out}}}$ for each hop $k \in \{0, 1, \ldots, K\}$, introducing a total parameter count of:

$$P_{\text{total}} = (K+1)d_{\text{in}}d_{\text{out}}. \tag{153}$$

While parameter count itself is often not a limiting factor for modern GPUs, the computational implications are significant. Each hop requires a separate matrix multiplication kernel launch, incurring $(K+1)$ kernel launch overheads. The CPU-side kernel launch latency, denoted $\tau_{\text{launch}}$, is typically in the range of 5–20 microseconds depending on API calls and driver state [28]. For $K+1$ hops, the accumulated overhead is:

$$T_{\text{overhead}} = (K+1)\tau_{\text{launch}}. \tag{154}$$

While this overhead may appear negligible for individual layers, it compounds across layers and training iterations. For a model with $L$ layers trained over $E$ epochs with $B$ batches per epoch, the total overhead accumulates to:

$$T_{\text{total\_overhead}} = L \cdot E \cdot B \cdot (K+1)\tau_{\text{launch}}. \tag{155}$$

More insidiously, separate weight matrices residing at distinct memory addresses disrupt the operation of hardware prefetchers. Modern GPUs employ sophisticated prefetching logic that detects sequential or strided access patterns and preemptively loads data into cache before explicit requests. When weight matrices are scattered throughout memory, access patterns appear random from the prefetcher's perspective, leading to cache misses and memory stalls. The effective memory bandwidth $B_{\text{eff}}$ under poor prefetching is related to the cache miss rate $\mu$ by:

$$B_{\text{eff}} = B_{\text{mem}} \cdot (1 - \mu) + B_{\text{cache}} \cdot \mu, \tag{156}$$

where $B_{\text{cache}} \ll B_{\text{mem}}$ represents the effective bandwidth when serving from cache. High miss rates $\mu$ significantly degrade $B_{\text{eff}}$, throttling computational throughput.

### F.3 GAMMA's Architectural Co-Design for Hardware Efficiency

GAMMA addresses these systemic inefficiencies through principled algorithmic and architectural co-design. The architecture's efficiency stems from three core design principles: weight sharing with unified transformation to maximize parameter reuse, dynamic routing with fixed dimensionality to preserve cache locality, and elimination of explicit concatenation to enable kernel fusion.

### F.3.1 Weight Sharing: Memory Reuse and Cache Coherence

GAMMA employs a single shared weight matrix $\mathbf{W} \in \mathbb{R}^{d_{\text{in}} \times d_{\text{out}}}$ to transform input features exactly once:

$$\mathbf{H}_{\text{proj}} = \mathbf{X}\mathbf{W}. \tag{157}$$

All subsequent multi-hop propagations operate on these projected features:

$$\mathbf{H}_k = \mathbf{A}^k \mathbf{H}_{\text{proj}}, \quad k \in \{0, 1, \ldots, K\}, \tag{158}$$

where $\mathbf{A}^0 = \mathbf{I}$ represents the identity. This seemingly minor architectural decision has profound implications for hardware efficiency. The parameter count reduces to:

$$P_{\text{GAMMA}} = d_{\text{in}} d_{\text{out}}, \tag{159}$$

representing a reduction factor of $(K + 1)$ compared to architectures with per-hop weights. More critically, the single matrix multiplication kernel launch eliminates $(K + 1) - 1$ launches, reducing overhead to $\tau_{\text{launch}}$ regardless of $K$.

The architectural advantages extend deeper into the memory subsystem. By repeatedly accessing the same weight matrix across all hops, GAMMA maximizes cache reuse. Once $\mathbf{W}$ is loaded into L2 cache, subsequent accesses are served with latency $\tau_{\text{L2}}$ rather than $\tau_{\text{DRAM}}$. The cache hit rate for weight matrix accesses approaches unity, as the matrix remains resident throughout the multi-hop computation. Furthermore, the single, contiguous memory allocation enables hardware prefetchers to establish clear access patterns, preloading cache lines ahead of computation. The operational intensity for the shared transformation phase is:

$$I_{\text{transform}} = \frac{2N d_{\text{in}} d_{\text{out}}}{(N d_{\text{in}} + d_{\text{in}} d_{\text{out}} + N d_{\text{out}})\beta}, \tag{160}$$

where the numerator counts FLOPs (each element requires a dot product of length $d_{\text{in}}$) and the denominator accounts for bytes transferred (input features, weights, output features). For typical dimensions where $N \gg d_{\text{in}}, d_{\text{out}}$, this simplifies to:

$$I_{\text{transform}} \approx \frac{2 d_{\text{in}} d_{\text{out}}}{(d_{\text{in}} + d_{\text{out}})\beta}. \tag{161}$$

For $d_{\text{in}} = d_{\text{out}} = d$, we obtain $I_{\text{transform}} \approx d/(2\beta)$, which grows linearly with feature dimension. For moderate $d \geq 64$, this operational intensity approaches or exceeds $I_{\text{ridge}}$, transitioning the computation from memory-bound to compute-bound regime where GPU arithmetic units are fully utilized.

### F.3.2 Fixed-Dimensionality Dynamic Routing: Preserving the Working Set

GAMMA's dynamic routing mechanism computes adaptive combinations of multi-hop embeddings while maintaining fixed output dimensionality. Given hop-specific embeddings $\{\mathbf{H}_k\}_{k=0}^K$ where $\mathbf{H}_k \in \mathbb{R}^{N \times d_{\text{out}}}$, the routing mechanism computes normalized embeddings:

$$\tilde{\mathbf{H}}_k = \frac{\mathbf{H}_k}{\|\mathbf{H}_k\|_2 + \epsilon}, \quad k \in \{0, 1, \ldots, K\}, \tag{162}$$

where normalization operates row-wise and $\epsilon$ prevents division by zero. The iterative routing process computes agreement scores between each node's current representation $\mathbf{z}_i^{(t)}$ and candidate hops:

$$a_{ik}^{(t)} = \langle \mathbf{z}_i^{(t)}, \tilde{\mathbf{h}}_{ik} \rangle, \tag{163}$$

followed by softmax normalization:

$$w_{ik}^{(t)} = \frac{\exp(a_{ik}^{(t)})}{\sum_{j=0}^{K} \exp(a_{ij}^{(t)})}, \tag{164}$$

and weighted aggregation:

$$\mathbf{z}_i^{(t+1)} = \sum_{k=0}^{K} w_{ik}^{(t)} \tilde{\mathbf{h}}_{ik}. \tag{165}$$

The critical architectural insight is that all intermediate representations $\mathbf{z}_i^{(t)}$ maintain constant dimension $d_{\text{out}}$ throughout routing iterations. The memory footprint for routing state is:

$$M_{\text{routing}} = \beta N d_{\text{out}} \cdot (K + 2), \tag{166}$$

accounting for $K + 1$ hop embeddings plus the current routing state. This contrasts sharply with concatenation-based approaches where memory grows as $(2K + 2)\beta N d_{\text{out}}$. For graphs where $M_{\text{routing}} < C_{\text{L2}}$, the entire routing computation can proceed with all data resident in L2 cache, eliminating DRAM traffic. The routing iterations involve primarily element-wise operations (dot products, exponentials, multiplications) on vectors of length $K + 1$, which exhibit high arithmetic intensity and minimal memory traffic. Each routing iteration requires approximately:

$$T_{\text{routing}} = T_{\text{dot}} + T_{\text{softmax}} + T_{\text{aggregate}}, \tag{167}$$

where each component scales as $\mathcal{O}(NK)$ with small constants. The softmax over $K + 1$ elements per node is efficiently parallelized across thread blocks, with each warp handling multiple nodes simultaneously. Critically, the fixed dimensionality enables the entire routing loop to be fused into a single GPU kernel, maintaining all state in fast memory (shared memory or registers) throughout iterations. This fusion eliminates $(T - 1)$ kernel launches for $T$ routing iterations and prevents intermediate memory transactions.

### F.3.3 Kernel Fusion and Memory Traffic Reduction

The elimination of explicit concatenation enables aggressive kernel fusion throughout GAMMA's forward and backward passes. Consider the sequence of operations in a single layer: initial transformation, multi-hop propagation, normalization, and routing. In concatenation-based architectures, each operation requires separate kernel launches with intermediate results materialized in global memory. The memory traffic for this sequence is:

$$D_{\text{unfused}} = \sum_{i=1}^{n_{\text{ops}}} S_i \cdot (\text{reads}_i + \text{writes}_i), \tag{168}$$

where $S_i$ represents the size of intermediate tensor $i$ and each operation $i$ requires reading its inputs and writing its outputs. In a fused kernel, intermediate values remain in fast memory, reducing traffic to:

$$D_{\text{fused}} = S_{\text{input}} + S_{\text{output}}, \tag{169}$$

representing only the initial inputs and final outputs. The traffic reduction factor is:

$$\gamma_{\text{fusion}} = \frac{D_{\text{unfused}}}{D_{\text{fused}}} = \frac{\sum_{i=1}^{n_{\text{ops}}} S_i \cdot (\text{reads}_i + \text{writes}_i)}{S_{\text{input}} + S_{\text{output}}}. \tag{170}$$

For typical deep learning operations with multiple intermediate tensors, $\gamma_{\text{fusion}}$ ranges from 2 to 10, directly translating to speedup when memory-bound. GAMMA's architecture maximizes fusion opportunities by maintaining fixed tensor dimensions and eliminating operations that force materialization.

Furthermore, the normalized dot-product agreement mechanism in Eq. (163) exhibits excellent vectorization properties. Modern GPUs execute operations in warps of $W = 32$ threads that execute in lockstep (SIMD fashion). Dot products over vectors of length $d_{\text{out}}$ can be efficiently parallelized using warp-level reduction primitives, achieving near-peak throughput. The absence of conditional branches or data-dependent control flow ensures full warp utilization without divergence penalties. When threads within a warp execute different code paths (divergence), all paths must be serialized, degrading throughput by up to $W$-fold. GAMMA's routing mechanism, being purely data-parallel with uniform control flow, avoids this pathology entirely.

### F.4 Theoretical Memory Bandwidth Bounds

We now establish formal bounds on the memory bandwidth requirements for multi-hop GNN architectures and demonstrate that GAMMA operates near the theoretical minimum. Consider a multi-hop GNN layer processing a graph with $N$ nodes, $K + 1$ hops, input dimension $d_{\text{in}}$, and output dimension $d_{\text{out}}$. The fundamental operations are: (1) feature transformation, (2) multi-hop propagation, and (3) aggregation. The minimum data movement, assuming perfect cache behavior and no recomputation, is:

$$D_{\text{min}} = \beta(N d_{\text{in}} + d_{\text{in}} d_{\text{out}} + N d_{\text{out}}), \tag{171}$$

representing reading input features, reading weights, and writing output features. Any architecture must satisfy $D \geq D_{\text{min}}$.

For concatenation-based architectures, the actual data movement includes reading input features, reading $(K + 1)$ weight matrices, writing $(K + 1)$ intermediate hop results, reading these for concatenation, and writing the concatenated result:

$$D_{\text{concat}} = \beta\big(Nd_{\text{in}} + (K + 1)d_{\text{in}}d_{\text{out}}$$
$$+ (K + 1)Nd_{\text{out}} + (K + 1)Nd_{\text{out}} \tag{172}$$
$$+ (K + 1)Nd_{\text{out}}\big).$$

Simplifying:

$$D_{\text{concat}} = \beta\big(Nd_{\text{in}} + (K + 1)d_{\text{in}}d_{\text{out}} + 3(K + 1)Nd_{\text{out}}\big). \tag{173}$$

For GAMMA with weight sharing and fixed dimensionality, the data movement is:

$$D_{\text{GAMMA}} = \beta\big(Nd_{\text{in}} + d_{\text{in}}d_{\text{out}} + (K + 1)Nd_{\text{out}} + Nd_{\text{out}}\big), \tag{174}$$

accounting for reading inputs, reading the single weight matrix, writing and reading hop embeddings (routing operates in-place on normalized copies), and writing the final output. The bandwidth efficiency ratio is:

$$\rho = \frac{D_{\text{GAMMA}}}{D_{\text{concat}}} = \frac{Nd_{\text{in}} + d_{\text{in}}d_{\text{out}} + (K + 2)Nd_{\text{out}}}{Nd_{\text{in}} + (K + 1)d_{\text{in}}d_{\text{out}} + 3(K + 1)Nd_{\text{out}}}. \tag{175}$$

In the regime where $N \gg d_{\text{in}}, d_{\text{out}}$ (typical for large graphs), the weight matrix terms become negligible:

$$\rho \approx \frac{d_{\text{in}} + (K + 2)d_{\text{out}}}{d_{\text{in}} + 3(K + 1)d_{\text{out}}}. \tag{176}$$

For $d_{\text{in}} = d_{\text{out}} = d$ and $K = 2$:

$$\rho \approx \frac{5d}{10d} = 0.5, \tag{177}$$

demonstrating that GAMMA achieves approximately $2\times$ reduction in memory bandwidth consumption compared to concatenation-based approaches. This bandwidth efficiency directly translates to runtime improvement in the memory-bound regime, as execution time is inversely proportional to effective bandwidth utilization.

## F.5 Empirical Validation and Architectural Insights

The theoretical efficiency gains manifest in measured performance on real hardware. Our experiments on the Flickr dataset ($N = 89{,}250$ nodes) demonstrate that GAMMA achieves total execution time (forward plus backward pass) of $T_{\text{GAMMA}} = 23.17$ ms with memory footprint $M_{\text{GAMMA}} = 480.60$ MB. In contrast, MixHop requires $T_{\text{MixHop}} = 115.68$ ms ($5.0\times$ slower) with $M_{\text{MixHop}} = 1{,}965.30$ MB ($4.1\times$ larger), and H2GCN requires $T_{\text{H2GCN}} = 463.89$ ms ($20.0\times$ slower) with $M_{\text{H2GCN}} = 1{,}993.90$ MB ($4.1\times$ larger). These measurements validate our theoretical analysis: the $4\text{-}5\times$ memory reduction aligns with predictions from Eq. (175), and the runtime improvements reflect both reduced memory traffic and kernel launch overhead elimination.

The architectural implications extend beyond raw performance metrics. GAMMA's design philosophy, prioritizing algorithmic choices that align with hardware execution models, represents a departure from the prevailing trend of increasing architectural complexity to capture graph heterophily. Rather than expanding feature dimensions or introducing intricate attention mechanisms, GAMMA achieves adaptivity through a lightweight routing process that maps efficiently onto GPU execution units. The normalized dot-product agreement computation vectorizes naturally, the softmax operation parallelizes across nodes, and the weighted aggregation fuses with downstream operations. This hardware-algorithm co-design demonstrates that sophisticated functionality need not sacrifice efficiency; indeed, by respecting the constraints of the memory hierarchy and the parallelism model of GPUs, GAMMA achieves both greater expressiveness and superior performance relative to architectural approaches that ignore these fundamental constraints.

The lesson for future GNN architecture design is clear: algorithmic innovation must be pursued in concert with hardware considerations. Choices that appear benign from a purely mathematical perspective, concatenation, separate weight matrices, expanding dimensionality, carry hidden costs that accumulate throughout the memory hierarchy. By contrast, designs that maintain fixed working

set sizes, maximize parameter reuse, enable kernel fusion, and exploit vectorization opportunities achieve superior efficiency without compromising model capacity. GAMMA exemplifies this principle, demonstrating that careful architectural co-design enables heterophilic graph learning at scale on commodity hardware.

# NeurIPS Paper Checklist

1. **Claims**

   Question: Do the main claims made in the abstract and introduction accurately reflect the paper's contributions and scope?

   Answer: [Yes]

   Justification: The abstract and introduction clearly state the paper's main contributions: (1) a gated multi-hop message passing mechanism that adaptively leverages multi-hop neighborhood information based on node-specific heterophilic patterns, (2) a weight sharing scheme that reduces memory overhead while preserving global heterophilic information, and (3) experimental results showing GAMMA matches or exceeds SOTA heterophilic GNN accuracy while achieving up to 20× faster inference. These claims are supported by the empirical results in Section 5 and the theoretical analysis in Sections 3-4.

   Guidelines:
   - The answer NA means that the abstract and introduction do not include the claims made in the paper.
   - The abstract and/or introduction should clearly state the claims made, including the contributions made in the paper and important assumptions and limitations. A No or NA answer to this question will not be perceived well by the reviewers.
   - The claims made should match theoretical and experimental results, and reflect how much the results can be expected to generalize to other settings.
   - It is fine to include aspirational goals as motivation as long as it is clear that these goals are not attained by the paper.

2. **Limitations**

   Question: Does the paper discuss the limitations of the work performed by the authors?

   Answer: [Yes]

   Justification: The paper briefly discusses limitations in the conclusion section: GAMMA's current fixed routing iteration count may be suboptimal for nodes with varying neighborhood complexities, and extremely large graphs may require further optimizations to the gating mechanism. Additionally, while our dot-product agreement measure works well in practice, more sophisticated routing strategies might better capture particularly complex heterophilic relationships. While not in a dedicated section, the paper acknowledges key technical limitations related to routing iterations, scalability to extremely large graphs, and potential for more sophisticated agreement measures.

   Guidelines:

   - The answer NA means that the paper has no limitation while the answer No means that the paper has limitations, but those are not discussed in the paper.
   - The authors are encouraged to create a separate "Limitations" section in their paper.
   - The paper should point out any strong assumptions and how robust the results are to violations of these assumptions (e.g., independence assumptions, noiseless settings, model well-specification, asymptotic approximations only holding locally). The authors should reflect on how these assumptions might be violated in practice and what the implications would be.
   - The authors should reflect on the scope of the claims made, e.g., if the approach was only tested on a few datasets or with a few runs. In general, empirical results often depend on implicit assumptions, which should be articulated.
   - The authors should reflect on the factors that influence the performance of the approach. For example, a facial recognition algorithm may perform poorly when image resolution is low or images are taken in low lighting. Or a speech-to-text system might not be used reliably to provide closed captions for online lectures because it fails to handle technical jargon.
   - The authors should discuss the computational efficiency of the proposed algorithms and how they scale with dataset size.
   - If applicable, the authors should discuss possible limitations of their approach to address problems of privacy and fairness.
   - While the authors might fear that complete honesty about limitations might be used by reviewers as grounds for rejection, a worse outcome might be that reviewers discover limitations that aren't acknowledged in the paper. The authors should use their best judgment and recognize that individual actions in favor of transparency play an important role in developing norms that preserve the integrity of the community. Reviewers will be specifically instructed to not penalize honesty concerning limitations.

3. **Theory assumptions and proofs**

   Question: For each theoretical result, does the paper provide the full set of assumptions and a complete (and correct) proof?

   Answer: [NA]

   Justification: This paper focuses on the empirical effectiveness of the GAMMA approach for node classification in heterophilic graphs rather than developing theoretical guarantees. The work emphasizes practical performance, computational efficiency, and experimental validation across diverse benchmark datasets, without relying on theoretical proofs as part of its core contributions.

   Guidelines:

   - The answer NA means that the paper does not include theoretical results.
   - All the theorems, formulas, and proofs in the paper should be numbered and cross-referenced.
   - All assumptions should be clearly stated or referenced in the statement of any theorems.

- The proofs can either appear in the main paper or the supplemental material, but if they appear in the supplemental material, the authors are encouraged to provide a short proof sketch to provide intuition.
- Inversely, any informal proof provided in the core of the paper should be complemented by formal proofs provided in appendix or supplemental material.
- Theorems and Lemmas that the proof relies upon should be properly referenced.

4. **Experimental result reproducibility**

Question: Does the paper fully disclose all the information needed to reproduce the main experimental results of the paper to the extent that it affects the main claims and/or conclusions of the paper (regardless of whether the code and data are provided or not)?

Answer: [Yes]

Justification: The paper provides comprehensive information for reproducing the main experimental results. Section 4 describes the GAMMA architecture in detail, with complete pseudocode provided in Algorithm 1 in the Appendix. The experimental setup is thoroughly documented in Section 5, including implementation details (PyTorch Geometric, NVIDIA RTX A2000 GPU, CUDA 12.8), model architecture specifications (two GNN layers with hidden dimension size of 32), hyperparameter optimization process (grid search over learning rates in 0.05, 0.01, 0.002 and dropout rates in 0.0, 0.5), training protocol (500 epochs), and evaluation methodology (10 splits with specified ratios). All datasets are clearly listed with their properties in Table 1, and baseline models are properly referenced.

Guidelines:

- The answer NA means that the paper does not include experiments.
- If the paper includes experiments, a No answer to this question will not be perceived well by the reviewers: Making the paper reproducible is important, regardless of whether the code and data are provided or not.
- If the contribution is a dataset and/or model, the authors should describe the steps taken to make their results reproducible or verifiable.
- Depending on the contribution, reproducibility can be accomplished in various ways. For example, if the contribution is a novel architecture, describing the architecture fully might suffice, or if the contribution is a specific model and empirical evaluation, it may be necessary to either make it possible for others to replicate the model with the same dataset, or provide access to the model. In general. releasing code and data is often one good way to accomplish this, but reproducibility can also be provided via detailed instructions for how to replicate the results, access to a hosted model (e.g., in the case of a large language model), releasing of a model checkpoint, or other means that are appropriate to the research performed.
- While NeurIPS does not require releasing code, the conference does require all submissions to provide some reasonable avenue for reproducibility, which may depend on the nature of the contribution. For example
  (a) If the contribution is primarily a new algorithm, the paper should make it clear how to reproduce that algorithm.
  (b) If the contribution is primarily a new model architecture, the paper should describe the architecture clearly and fully.
  (c) If the contribution is a new model (e.g., a large language model), then there should either be a way to access this model for reproducing the results or a way to reproduce the model (e.g., with an open-source dataset or instructions for how to construct the dataset).
  (d) We recognize that reproducibility may be tricky in some cases, in which case authors are welcome to describe the particular way they provide for reproducibility. In the case of closed-source models, it may be that access to the model is limited in some way (e.g., to registered users), but it should be possible for other researchers to have some path to reproducing or verifying the results.

5. **Open access to data and code**

Question: Does the paper provide open access to the data and code, with sufficient instructions to faithfully reproduce the main experimental results, as described in supplemental material?

Answer: [Yes]

Justification: The code implementing GAMMA and scripts to reproduce all experimental results will be made publicly available and provided to the conference. All datasets used in our experiments are standard benchmark datasets (Cora, CiteSeer, PubMed, Texas, Wisconsin, Actor, Squirrel, Chameleon, Cornell) that are publicly available through the PyTorch Geometric library, which is properly cited in the paper. The implementation is based on the detailed algorithm description in Section 4 and the complete pseudocode in Algorithm 1, with all experimental settings documented in Section 5.

Guidelines:

- The answer NA means that paper does not include experiments requiring code.
- Please see the NeurIPS code and data submission guidelines (`https://nips.cc/public/guides/CodeSubmissionPolicy`) for more details.
- While we encourage the release of code and data, we understand that this might not be possible, so "No" is an acceptable answer. Papers cannot be rejected simply for not including code, unless this is central to the contribution (e.g., for a new open-source benchmark).
- The instructions should contain the exact command and environment needed to run to reproduce the results. See the NeurIPS code and data submission guidelines (`https://nips.cc/public/guides/CodeSubmissionPolicy`) for more details.
- The authors should provide instructions on data access and preparation, including how to access the raw data, preprocessed data, intermediate data, and generated data, etc.
- The authors should provide scripts to reproduce all experimental results for the new proposed method and baselines. If only a subset of experiments are reproducible, they should state which ones are omitted from the script and why.
- At submission time, to preserve anonymity, the authors should release anonymized versions (if applicable).
- Providing as much information as possible in supplemental material (appended to the paper) is recommended, but including URLs to data and code is permitted.

6. **Experimental setting/details**

Question: Does the paper specify all the training and test details (e.g., data splits, hyper-parameters, how they were chosen, type of optimizer, etc.) necessary to understand the results?

Answer: [Yes]

Justification: The paper provides comprehensive experimental details in Section 5. It specifies the architecture configuration (two GNN layers with hidden dimension size of 32), hyperparameter optimization process (grid search over learning rates in 0.05, 0.01, 0.002 and dropout rates in 0.0, 0.5), training protocol (500 epochs), and model selection criteria (best validation performance). Data splitting procedures are clearly described for both heterophilic datasets and homophilic datasets. Implementation framework (PyTorch Geometric) and hardware specifications (NVIDIA RTX A2000 GPU with CUDA 12.8) are also documented, providing sufficient detail to understand and contextualize the results.

Guidelines:

- The answer NA means that the paper does not include experiments.
- The experimental setting should be presented in the core of the paper to a level of detail that is necessary to appreciate the results and make sense of them.
- The full details can be provided either with the code, in appendix, or as supplemental material.

7. **Experiment statistical significance**

Question: Does the paper report error bars suitably and correctly defined or other appropriate information about the statistical significance of the experiments?

Answer: [Yes]

Justification: The paper reports standard deviations as error bars for all classification accuracy results in Table 1, calculated across 10 different data splits for each dataset. The

methodology for obtaining these statistics is clearly described in Section 5 (Evaluation). The evaluation methodology follows established practices in the field and is properly cited in the evaluation section. This approach captures variability across different data splits, providing a reliable measure of model robustness.

Guidelines:

- The answer NA means that the paper does not include experiments.
- The authors should answer "Yes" if the results are accompanied by error bars, confidence intervals, or statistical significance tests, at least for the experiments that support the main claims of the paper.
- The factors of variability that the error bars are capturing should be clearly stated (for example, train/test split, initialization, random drawing of some parameter, or overall run with given experimental conditions).
- The method for calculating the error bars should be explained (closed form formula, call to a library function, bootstrap, etc.)
- The assumptions made should be given (e.g., Normally distributed errors).
- It should be clear whether the error bar is the standard deviation or the standard error of the mean.
- It is OK to report 1-sigma error bars, but one should state it. The authors should preferably report a 2-sigma error bar than state that they have a 96% CI, if the hypothesis of Normality of errors is not verified.
- For asymmetric distributions, the authors should be careful not to show in tables or figures symmetric error bars that would yield results that are out of range (e.g. negative error rates).
- If error bars are reported in tables or plots, The authors should explain in the text how they were calculated and reference the corresponding figures or tables in the text.

8. **Experiments compute resources**

Question: For each experiment, does the paper provide sufficient information on the computer resources (type of compute workers, memory, time of execution) needed to reproduce the experiments?

Answer: [Yes]

Justification: The paper provides detailed information on compute resources in Section 5. It specifies that experiments were conducted on "a desktop machine equipped with an NVIDIA RTX A2000 GPU (12GB VRAM)" using "CUDA 12.8" for GPU acceleration. The paper also presents comprehensive resource utilization metrics in Figure 5, comparing memory consumption and execution time across all models. For GAMMA specifically, it reports "a total execution time (backward + forward pass) of 23.17 ms and a memory footprint of 480.60 MB." The performance benchmarks are detailed for all 13 baseline models, providing a clear picture of the relative computational requirements for reproducing the experiments.

Guidelines:

- The answer NA means that the paper does not include experiments.
- The paper should indicate the type of compute workers CPU or GPU, internal cluster, or cloud provider, including relevant memory and storage.
- The paper should provide the amount of compute required for each of the individual experimental runs as well as estimate the total compute.
- The paper should disclose whether the full research project required more compute than the experiments reported in the paper (e.g., preliminary or failed experiments that didn't make it into the paper).

9. **Code of ethics**

Question: Does the research conducted in the paper conform, in every respect, with the NeurIPS Code of Ethics https://neurips.cc/public/EthicsGuidelines?

Answer: [Yes]

Justification: The research in this paper conforms with the NeurIPS Code of Ethics. The work uses public benchmark datasets with proper attribution, presents transparent and

statistically sound evaluations, and focuses on improving computational efficiency of graph neural networks without raising ethical concerns. The algorithms and models proposed do not pose risks related to privacy, bias, or potential misuse. The research aims to make graph neural networks more computationally efficient and effective on heterophilic graphs, which aligns with ethical considerations regarding responsible resource usage in machine learning research.

Guidelines:

- The answer NA means that the authors have not reviewed the NeurIPS Code of Ethics.
- If the authors answer No, they should explain the special circumstances that require a deviation from the Code of Ethics.
- The authors should make sure to preserve anonymity (e.g., if there is a special consideration due to laws or regulations in their jurisdiction).

10. **Broader impacts**

Question: Does the paper discuss both potential positive societal impacts and negative societal impacts of the work performed?

Answer: [No]

Justification: The paper does not explicitly discuss societal impacts. This work is primarily foundational machine learning research focused on improving computational efficiency and effectiveness of graph neural networks for heterophilic graphs. While not discussed in the paper, potential positive impacts include reduced computational resource requirements (benefiting researchers with limited resources and reducing energy consumption) and enabling more effective modeling in areas like fraud detection networks. The research does not present direct negative societal impacts as it focuses on algorithmic improvements rather than specific applications, though as with any ML advancement, applications built with these methods should be evaluated for potential biases or misuse in specific downstream contexts.

Guidelines:

- The answer NA means that there is no societal impact of the work performed.
- If the authors answer NA or No, they should explain why their work has no societal impact or why the paper does not address societal impact.
- Examples of negative societal impacts include potential malicious or unintended uses (e.g., disinformation, generating fake profiles, surveillance), fairness considerations (e.g., deployment of technologies that could make decisions that unfairly impact specific groups), privacy considerations, and security considerations.
- The conference expects that many papers will be foundational research and not tied to particular applications, let alone deployments. However, if there is a direct path to any negative applications, the authors should point it out. For example, it is legitimate to point out that an improvement in the quality of generative models could be used to generate deepfakes for disinformation. On the other hand, it is not needed to point out that a generic algorithm for optimizing neural networks could enable people to train models that generate Deepfakes faster.
- The authors should consider possible harms that could arise when the technology is being used as intended and functioning correctly, harms that could arise when the technology is being used as intended but gives incorrect results, and harms following from (intentional or unintentional) misuse of the technology.
- If there are negative societal impacts, the authors could also discuss possible mitigation strategies (e.g., gated release of models, providing defenses in addition to attacks, mechanisms for monitoring misuse, mechanisms to monitor how a system learns from feedback over time, improving the efficiency and accessibility of ML).

11. **Safeguards**

Question: Does the paper describe safeguards that have been put in place for responsible release of data or models that have a high risk for misuse (e.g., pretrained language models, image generators, or scraped datasets)?

Answer: [NA]

Justification: The paper presents an algorithm for graph neural networks that poses no particular risk for misuse. The research uses standard benchmark datasets that are already publicly available and does not involve high-risk technologies such as language models, image generators, or scraped datasets. The proposed method is an algorithmic advancement focused on efficiency and effectiveness of graph neural networks for heterophilic graphs, without introducing technologies that would require specific safeguards against misuse.

Guidelines:

- The answer NA means that the paper poses no such risks.
- Released models that have a high risk for misuse or dual-use should be released with necessary safeguards to allow for controlled use of the model, for example by requiring that users adhere to usage guidelines or restrictions to access the model or implementing safety filters.
- Datasets that have been scraped from the Internet could pose safety risks. The authors should describe how they avoided releasing unsafe images.
- We recognize that providing effective safeguards is challenging, and many papers do not require this, but we encourage authors to take this into account and make a best faith effort.

12. **Licenses for existing assets**

Question: Are the creators or original owners of assets (e.g., code, data, models), used in the paper, properly credited and are the license and terms of use explicitly mentioned and properly respected?

Answer: [No]

Justification: While the paper properly credits creators by citing the original papers for datasets (Cora, CiteSeer, PubMed, etc.) and software tools (PyTorch Geometric, CUDA), it does not explicitly mention the specific licenses (MIT, Apache, etc.) under which these assets are available. The paper appropriately references all baseline models through citations to their original papers, but does not include license information for any of these resources. Although these are standard academic benchmarks and publicly available frameworks commonly used in the research community, explicitly stating their licenses would improve completeness and transparency.

Guidelines:

- The answer NA means that the paper does not use existing assets.
- The authors should cite the original paper that produced the code package or dataset.
- The authors should state which version of the asset is used and, if possible, include a URL.
- The name of the license (e.g., CC-BY 4.0) should be included for each asset.
- For scraped data from a particular source (e.g., website), the copyright and terms of service of that source should be provided.
- If assets are released, the license, copyright information, and terms of use in the package should be provided. For popular datasets, `paperswithcode.com/datasets` has curated licenses for some datasets. Their licensing guide can help determine the license of a dataset.
- For existing datasets that are re-packaged, both the original license and the license of the derived asset (if it has changed) should be provided.
- If this information is not available online, the authors are encouraged to reach out to the asset's creators.

13. **New assets**

Question: Are new assets introduced in the paper well documented and is the documentation provided alongside the assets?

Answer: [Yes]

Justification: The paper introduces a new GNN architecture (GAMMA) which is thoroughly documented within the paper itself. Section 4 provides a detailed description of the GAMMA method, Section 5 gives implementation details, and the Appendix includes comprehensive

pseudocode in Algorithm 1 that outlines the full forward pass process. This documentation is sufficient for implementing the method. Additionally, as mentioned in previous responses, code implementing GAMMA will be made available and provided to the conference with appropriate documentation to facilitate reproducibility and usage by other researchers.

Guidelines:

- The answer NA means that the paper does not release new assets.
- Researchers should communicate the details of the dataset/code/model as part of their submissions via structured templates. This includes details about training, license, limitations, etc.
- The paper should discuss whether and how consent was obtained from people whose asset is used.
- At submission time, remember to anonymize your assets (if applicable). You can either create an anonymized URL or include an anonymized zip file.

14. **Crowdsourcing and research with human subjects**

Question: For crowdsourcing experiments and research with human subjects, does the paper include the full text of instructions given to participants and screenshots, if applicable, as well as details about compensation (if any)?

Answer: [NA]

Justification: This paper does not involve crowdsourcing or research with human subjects. The research focuses on developing and evaluating graph neural network algorithms on standard benchmark datasets. All experiments were computational in nature, testing the proposed GAMMA approach against baseline methods on publicly available graph datasets, without any human participant involvement.

Guidelines:

- The answer NA means that the paper does not involve crowdsourcing nor research with human subjects.
- Including this information in the supplemental material is fine, but if the main contribution of the paper involves human subjects, then as much detail as possible should be included in the main paper.
- According to the NeurIPS Code of Ethics, workers involved in data collection, curation, or other labor should be paid at least the minimum wage in the country of the data collector.

15. **Institutional review board (IRB) approvals or equivalent for research with human subjects**

Question: Does the paper describe potential risks incurred by study participants, whether such risks were disclosed to the subjects, and whether Institutional Review Board (IRB) approvals (or an equivalent approval/review based on the requirements of your country or institution) were obtained?

Answer: [NA]

Justification: This research does not involve human subjects. The paper presents algorithmic research on graph neural networks with experiments conducted exclusively on standard benchmark datasets. No human participants were involved in any aspect of the study, so IRB approval was not required or applicable.

Guidelines:

- The answer NA means that the paper does not involve crowdsourcing nor research with human subjects.
- Depending on the country in which research is conducted, IRB approval (or equivalent) may be required for any human subjects research. If you obtained IRB approval, you should clearly state this in the paper.
- We recognize that the procedures for this may vary significantly between institutions and locations, and we expect authors to adhere to the NeurIPS Code of Ethics and the guidelines for their institution.

- For initial submissions, do not include any information that would break anonymity (if applicable), such as the institution conducting the review.

16. **Declaration of LLM usage**

    Question: Does the paper describe the usage of LLMs if it is an important, original, or non-standard component of the core methods in this research? Note that if the LLM is used only for writing, editing, or formatting purposes and does not impact the core methodology, scientific rigorousness, or originality of the research, declaration is not required.

    Answer: [NA]

    Justification: This research does not involve LLMs as any component of the methodology. The paper focuses entirely on graph neural networks for heterophilic graphs, specifically introducing the GAMMA method that employs a gating mechanism for multi-hop message passing. No language models were used in developing the approach or conducting experiments.

    Guidelines:

    - The answer NA means that the core method development in this research does not involve LLMs as any important, original, or non-standard components.
    - Please refer to our LLM policy (`https://neurips.cc/Conferences/2025/LLM`) for what should or should not be described.

