# OpenReview forum: "GAMMA: Gated Multi-hop Message Passing for Homophily-Agnostic Node Representation in GNNs"
_NeurIPS.cc/2025/Conference — NeurIPS 2025 poster_

### Official Review · Reviewer_DJEL · 2025-06-22

**Clarity:** 3
**Significance:** 2
**Originality:** 2
**Rating:** 3
**Confidence:** 3

**Summary:**

GAMMA proposes an efficient GNN architecture for learning in heterophilic or mixed-homophily graphs. It avoids traditional multi-hop concatenation by using an iterative routing-style gating mechanism that adaptively weights multi-hop embeddings per node, while sharing a single projection matrix to reduce overhead. The paper performs detailed computational profiling and demonstrates significant accuracy and efficiency gains on benchmark datasets.

**Questions:**

-	How does the routing mechanism converge, and is it stable across diverse graph types? It’ll be helpful to provide analysis on routing convergence behavior and its effect on representation learning.
-	Is weight sharing across hops always beneficial — could this limit expressiveness for disjoint hop information?
-	How sensitive is GAMMA to graph noise or perturbations (e.g., edge rewiring)?
-	Can GAMMA adapt to temporal or attributed dynamic graphs?

I'm open to increase the score if the questions and the weaknesses are reasonably addressed.

**Ethical Concerns:**

["NO or VERY MINOR ethics concerns only"]

**Final Justification:**

I appreciate the authors' engagement in the rebuttal. I think the evaluation of both performance and efficiency in the submitted version can be better set up.

**Limitations:**

yes.

**Quality:**

2

**Strengths And Weaknesses:**

## Strength:
-	The idea of iterative soft-attention routing over hops seems novel and well-motivated.
-	Interesting analysis on the efficiency of current heterophily methods ,and the results show the proposed message passing scheme is time- and memory-efficient.
-	Some analysis are intuitive and elegant,e.g., the capsule network analogy and gating mechanism.
-	Some empirical gains (i.e., runtime and memory profiling of 13 baselines) are significant, e.g., up to 20× faster inference while matching or exceeding state-of-the-art accuracy on some datasets.

## Weakness:
-	One major concern is the novelty is incremental: The proposed message passing scheme is quite similar to the one proposed in GPR-GNN. I don’t see any clear distinction. Also, GPR-GNN allows having negative coefficients for combining hop-specific embeddings $\mathbf{H}$, while the proposed scheme does not have that flexibility.
-	In line 66, the authors mention "Adaptive node-specific message passing". However, the proposed scheme is not node-specific but rather hop-specific. This is confusing.
-	A similar line of work, which actually formulates node-specific message passing [1], is not discussed and included as a baseline.
-	While routing is claimed to be efficient, no direct ablation on routing steps (e.g., 1 vs 3) is presented. It will be helpful to include ablation study on routing iterations and gating stability.

-	Theoretical analysis of convergence, expressivity, or generalization are missing.
-	Performance shown mostly on small to mid-scale datasets (e.g., Cornell, Tolokers); unclear how it scales to large graphs. It will be interesting to see how the proposed method perform on the heterophilic datasets introduced in [2]. Also, it would be interesting to study the proposed method with large scale OGB datasets.

## References:

[1] Finkelshtein et al. Cooperative Graph Neural Networks. ICML 2024.

[2] Platonov et al. A Critical Look at the Evaluation of GNNs Under Heterophily : Are We Really Making Progress? ICLR 2023.

---

> ### Author Rebuttal · Authors · 2025-07-31
>
> We thank the reviewer for the detailed and constructive feedback. The comments have been invaluable in improving both the quality and clarity of our manuscript.
>
> 1. Novelty and distinction from GPR-GNN:
>
> GPR-GNN learns a *global* polynomial filter, one coefficient vector applied identically to every node, capturing only dataset-level structure. GAMMA's global scaling vectors $\gamma_{p}$ are unconstrained and can become negative during training, allowing the model to subtract or amplify features at the hop level. However, the key distinction from GPR-GNN is architectural: GPR-GNN applies its coefficients (which can be negative) uniformly across all nodes, while GAMMA combines negative-capable global scaling with positive node-specific routing coefficients $α_{i,p}$ (from softmax). This dual-level design provides both global feature manipulation and local adaptive weighting.
>
> 2. Comparison with CO-GNN and computational efficiency:
>
> we have benchmarked against $\\text{CO-GNN } (\\ast,\\ast)$, the best-performing configuration reported in their paper, and the results show GAMMA's superior performance.
> $$
> \\begin{array}{l|ccccccccc}
> \\hline
> & \\text{Cora} & \\text{CiteSeer} & \\text{PubMed} & \\text{Texas} & \\text{Wisconsin} & \\text{Actor} & \\text{Squirrel} & \\text{Chameleon} & \\text{Cornell} \\\\
> \\hline
> \\text{CO-GNN } (\\ast,\\ast) & 87.48 \\pm 0.71 & 76.26 \\pm 1.32 & 88.44 \\pm 0.35 & 79.03 \\pm 2.47 & 71.95 \\pm 3.28 & 32.03 \\pm 2.68 & 31.20 \\pm 3.85 & 58.28 \\pm 1.62 & 75.03 \\pm 2.26 \\\\
> \\hline
> \\text{GAMMA} & 87.42 \\pm 1.01 & 75.49 \\pm 1.67 & 89.62 \\pm 0.43 & 87.37 \\pm 3.68 & 86.27 \\pm 4.21 & 35.59 \\pm 1.29 & 36.20 \\pm 1.01 & 51.16 \\pm 2.22 & 78.68 \\pm 3.42 \\\\
> \\hline
> \\end{array}
> $$
>
> As shown in the table, GAMMA achieves higher accuracy on challenging heterophilic datasets. This superior performance is achieved while adhering to our primary design philosophy: prioritizing computational efficiency for heterophilic graphs. In contrast, CO-GNN introduces significant overhead due to its dual-network design, requiring separate action and environment networks and doubling the number of linear transformations and graph convolutions per layer.
>
> CO-GNN incurs a 4.4x backward pass overhead compared to a standard GCN, with a forward pass time of 1.820 ms and a backward pass of 13.431 ms, ranking 12th out of 15 models on Cora (these profilings will be added to the manuscript). GAMMA’s streamlined architecture avoids these costs, delivering state-of-the-art accuracy with up to 12x lower memory usage and faster inference than leading alternatives.
>
> 3. "Node-Specific" Terminology:
>
> The distinction between "node-specific" and "hop-specific" is best understood by separating the feature aggregation process from the gating mechanism. This is illustrated in Algorithm 1 in the supplementary material (Appendix A.2). Hop-specific refers to the initial, global aggregation of features. For each node and its neighbors at hop distance $p$, we generate an embedding $H^{(p)}$ by gathering information from all nodes at that distance. However, node-specific refers to our gating mechanism. After the hop-specific embeddings are generated, each node $i$ individually computes its own set of routing coefficients, $\alpha_{i,p}$. These coefficients are unique to each node because they are determined by an "agreement" score calculated between the node's own evolving representation and the messages it receives from each hop.
> Thank you for pointing out this potential for confusion. We will revise the manuscript to more clearly state that.
>
> 4. Theoretical analysis:
>
> Appendix A.4 proves convergence to a unique fixed point, demonstrates that GAMMA’s function class strictly contains concatenation-based methods, and shows adaptive routing maximizes mutual information between embeddings and labels.
>
> 5. Large-scale evaluation:
> To address scalability, we have benchmarked GAMMA on seven large-scale heterophilic graphs. The results in the provided table below show that GAMMA achieves state-of-the-art or second-best accuracy on 5 of the 7 datasets.
>
> $$
> \\begin{array}{l|ccccccc}
> \\hline
> & \\text{ogbn-arxiv} & \\text{roman-empire} & \\text{ogbn-products} & \\text{minesweeper} & \\text{amazon-ratings} & \\text{amazon-photos} & \\text{tolokers} \\\\
> \\hline
> \\text{MLP}   & 47.34 \\pm 0.09 & 65.37 \\pm 0.58 & 43.20 \\pm 0.01 & 79.48 \\pm 0.21 & 38.35 \\pm 0.20 & 48.73 \\pm 1.80 & 78.16 \\pm 0.00 \\\\
> \\text{GCN}   & 64.91 \\pm 0.26 & 54.16 \\pm 0.28 & \\mathbf{71.31 \\pm 0.16} & 80.31 \\pm 0.24 & 42.12 \\pm 0.29 & 52.14 \\pm 1.50 & 78.76 \\pm 0.17 \\\\
> \\text{SGC}   & 63.47 \\pm 0.16 & 61.20 \\pm 0.65 & \\text{OOM} & 80.05 \\pm 0.03 & 41.61 \\pm 0.72 & 57.97 \\pm 0.52 & 78.56 \\pm 0.19 \\\\
> \\text{GCN2}  & 61.96 \\pm 0.42 & 64.96 \\pm 0.38 & 61.08 \\pm 0.18 & 82.80 \\pm 0.35 & 47.20 \\pm 0.55 & 94.10 \\pm 0.25 & 79.80 \\pm 0.30 \\\\
> \\text{GCNJK} & 70.19 \\pm 0.11 & 62.18 \\pm 0.91 & 69.46 \\pm 0.33 & 85.15 \\pm 0.42 & 48.77 \\pm 0.37 & 95.28 \\pm 0.05 & \\mathbf{81.23 \\pm 0.26} \\\\
> \\text{GATJK} & \\mathbf{70.75 \\pm 0.19} & 70.00 \\pm 0.70 & \\text{OOM} & \\mathbf{85.65 \\pm 0.34} & \\mathbf{50.84 \\pm 0.52} & 95.71 \\pm 0.11 & \\mathbf{81.31 \\pm 0.48} \\\\
> \\text{H2GCN} & \\text{OOM} & 75.89 \\pm 0.55 & \\text{OOM} & 83.31 \\pm 0.65 & 42.90 \\pm 0.38 & 84.95 \\pm 0.38 & 79.39 \\pm 0.42 \\\\
> \\text{GAT}   & 70.01 \\pm 0.09 & 74.50 \\pm 0.45 & \\text{OOM} & 81.52 \\pm 0.26 & 48.89 \\pm 0.61 & 95.56 \\pm 0.15 & 78.63 \\pm 0.06 \\\\
> \\text{MixHop}& 70.27 \\pm 0.14 & \\mathbf{82.20 \\pm 0.31} & \\text{OOM} & 84.88 \\pm 0.45 & \\mathbf{50.98 \\pm 0.30} & \\mathbf{96.34 \\pm 0.30} & 81.03 \\pm 0.70 \\\\
> \\text{BernNet}& 63.15 \\pm 0.13 & 66.27 \\pm 0.35 & \\text{OOM} & 80.00 \\pm 0.07 & 40.90 \\pm 0.09 & 62.34 \\pm 7.02 & 78.16 \\pm 0.00 \\\\
> \\text{GPRGNN}& 67.66 \\pm 0.07 & 69.60 \\pm 0.40 & \\mathbf{73.26 \\pm 0.65} & 80.36 \\pm 0.19 & 43.44 \\pm 0.13 & 79.54 \\pm 0.39 & 78.36 \\pm 0.10 \\\\
> \\text{M2MGNN}& \\mathbf{72.52 \\pm 0.10} & \\mathbf{80.81 \\pm 0.70} & \\text{OOM} & \\mathbf{85.83 \\pm 0.25} & \\mathbf{50.57 \\pm 0.29} & \\mathbf{95.73 \\pm 0.25} & 80.66 \\pm 0.47 \\\\
> \\text{GAMMA} & \\mathbf{71.81 \\pm 0.19} & \\mathbf{81.37 \\pm 0.25} & \\mathbf{72.61 \\pm 0.04} & \\mathbf{87.58 \\pm 0.25} & 49.07 \\pm 0.65 & \\mathbf{96.28 \\pm 0.11} & \\mathbf{82.59 \\pm 0.41} \\\\
> \\hline
> \\end{array}
> $$
>
> Notably, many SOTA models encounter Out-Of-Memory (OOM) errors on larger graphs (on the same setup, NVIDIA A100 - 80 GB) like ogbn-products. GAMMA's design, which avoids feature concatenation and uses an efficient gating mechanism, enables it to scale effectively.
>
> Responses to questions:
>
> - Convergence and stability: The convergence and stability of the routing mechanism are supported by both theoretical guarantees and empirical results. Theorem 1 (Appendix A.4.2) formally proves that the iterative routing update rule converges to a unique fixed point that optimally balances agreement between hop-specific embeddings and their weighted combination, with an entropy term preventing degenerate solutions. Empirically, our ablation study (Appendix A.1) demonstrates stability across datasets: performance consistently peaks within 2 to 6 iterations. For example, on heterophilic datasets (Cornell, Actor), performance peaks at 6 and 7 iterations, while on homophilic datasets (Cora, PubMed), it stabilizes after just 2-5 iterations. This shows the mechanism quickly finds a stable, effective weighting of hop information across diverse graph structures without extensive tuning.
>
> - Weight sharing and expressiveness: While using a shared weight matrix $W$ across hops improves parameter efficiency, it does not reduce GAMMA’s expressive power. Appendix A.5.1 shows that GAMMA’s function class $F_{GAMMA}$ strictly contains that of concatenation-based models (H2GCN or MixHop). This greater expressiveness comes from a dual-level adaptation: global hop-specific scaling vectors $\gamma_{p}$ let the model learn the importance of feature channels at each hop, while node-specific gating coefficients $\alpha_{i,p}$ enable dynamic routing based on each node’s local structure. This combination of shared geometric space with both global and local adaptive weighting allows GAMMA to represent any function a concatenation-based model can—and more, as shown in the proof.
>
> - Robustness to noise: GAMMA is inherently robust to noise and structural perturbations due to its adaptive gating mechanism. This resilience stems from agreement-based routing: if a spurious edge introduces noise into a hop embedding $H_{i}^{(p)}$, its low dot-product agreement with the node’s evolving representation $v_{i}^{(t)}$ prevents the routing logit $b_{i,p}$ from increasing, so the softmax assigns it a low coefficient $\alpha_{i,p}$, effectively filtering out the noise. The multi-hop design adds redundancy; if a node’s 1-hop neighborhood is noisy, the routing can up-weight information from its more stable 0-hop or 2-hop representations. GAMMA’s strong results on inherently noisy real-world datasets further support this robustness.
>
> - Adaptation to dynamic graphs: GAMMA is a potential candidate for extension to temporal or dynamic graphs. Its parameter efficiency means fewer updates are needed as the graph evolves. The lightweight routing mechanism is computationally inexpensive, enabling frequent and efficient re-computation of node-specific gating coefficients $\alpha_{i,p}$ with each graph change, allowing fast adaptation to local structural modifications without retraining the entire model. While the current version is for static graphs, extending GAMMA to dynamic settings is a promising direction. This could involve making the routing mechanism time-aware, such as incorporating a temporal decay factor into the routing logits or making the global scaling vectors $\gamma_{p}$ time-dependent to capture evolving global patterns.

---

> > ### Comment · Reviewer_DJEL · 2025-08-04
> >
> > I appreciate the authors' clarification of the "node-specific" terminology and distinction with GPR-GNN. Meanwhile, the additional results seem not impressive. For the heterophilic datasets, the results show modest improvement for a couple of datasets, but even worse performance on the roman-empire and amazon-ratings datasets. These datasets are still smaller datasets. More importantly, for larger datasets such as ogb-arxiv and ogb-products, the proposed method is worse than the baselines. Thus, I'm inclined to keep the score.

---

> ### Author Response · Authors · 2025-08-06
>
> We thank the reviewer for their continued engagement and valuable feedback.
>
> Regarding the performance concerns on large-scale datasets, while GAMMA does not achieve state-of-the-art results on **all** benchmarks, it performs either first or second place in 6 out of 7 of newly added datasets. Further, we would like to add that looking at the broader literature, this reflects a common characteristic across heterophilic GNN methods [1, 2, 3, 4, 5], where no single architecture achieves universal dominance. For instance, LRGNN [1] underperforms on 4 of 9 small datasets despite its theoretical and methodological contributions. Similarly, H2GCN [2] and Neural Sheaf Diffusion [3] show variable performance across benchmarks while making significant architectural improvements.
>
> GAMMA's core contribution lies in its computational efficiency, which enables full-batch training of heterophilic GNNs on commodity hardware, addressing an important efficiency bottleneck and a practical constraint that has limited accessibility to these methods. As reviewer "hXHc" noted, the computational feasibility of expensive GNNs on standard GPUs is a concern. Most state-of-the-art methods, including H2GCN and CO-GNN mentioned by the reviewer, encounter out-of-memory errors even on medium-sized datasets when using typical desktop GPUs (e.g., NVIDIA RTX A2000, GeForce RTX 3050 and RTX 2000 Ada Generation), which is why they were initially excluded from our commodity hardware experiments.
>
> To address scalability concerns raised by reviewers, we upgraded our evaluation to an A100 80GB GPU, a server-grade, non-commodity setup allowing all baseline methods to complete full-batch training without memory constraints. In this enhanced environment, GAMMA achieves comparable accuracy while maintaining its efficiency advantages. Critically, GAMMA averages 7.57× faster execution time and consumes 24.18× less memory than competing methods across these newly added datasets, approaching the efficiency of simple GCNs while handling heterophily effectively.
>
> This efficiency enables practical deployment scenarios that existing methods cannot support. GAMMA reduces the computational gap between heterophilic GNNs and simple architectures like GCN, making heterophilic graph learning accessible to users with limited computational resources.
>
> [1] Liang et al., "Predicting global label relationship matrix for graph neural networks under heterophily," NeurIPS 2023
>
> [2] Zhu et al., "Beyond homophily in graph neural networks: Current limitations and effective designs," NeurIPS 2020
>
> [3] Bodnar et al., "Neural sheaf diffusion: A topological perspective on heterophily and oversmoothing in gnns," NeurIPS 2022
>
> [4] Yang et al., "Diverse message passing for attribute with heterophily," NeurIPS 2021
>
> [5] Li et al., "Finding global homophily in graph neural networks when meeting heterophily," ICML 2022

---

### Official Review · Reviewer_1qWb · 2025-06-27

**Clarity:** 4
**Significance:** 3
**Originality:** 3
**Rating:** 5
**Confidence:** 4

**Summary:**

The paper introduces gated multi-hop message passing (GAMMA) as a method for efficient node classification on heterophilic graphs. Compared to existing methods, GAMMA provides a 20x improvement in inference speed and achieves comparable performance to the state-of-the-art. The efficiency of this method comes from a novel gating mechanism which avoids the need for feature concatenation, and the introduction of a weight sharing scheme.

**Questions:**

Could you include some evaluation of GAMMA on larger scale graphs or at least some discussion of why this not included?

Is the absence of evaluation on larger scale graphs due to any limitations in scaling up GAMMA?

**Ethical Concerns:**

["NO or VERY MINOR ethics concerns only"]

**Final Justification:**

The authors have addressed my concerns about evaluation on larger scale graphs, and I am happy that the authors will incorporate the suggestions of this review in the revised paper. I maintain my score and recommendation that this paper be accepted.

**Limitations:**

yes

**Quality:**

3

**Strengths And Weaknesses:**

**Strengths:**

-	The paper introduces GAMMA to tackle the known and relevant problem of GNN performance and efficiency on heterophilic graphs and provides convincing evidence that GAMMA can address these issues.
-	GAMMA is well motivated and explained intuitively through the use of figures.
-	The paper is very clear and well written.

**Weaknesses:**

-	The evaluation of GAMMA is limited to small scale graphs. Evaluation on larger scale heterophilic graphs like the ones discussed in Lime et al. 2021 or Platonov 2023 would provide stronger support for the applicability of this method in realistic settings.
-	The conclusion briefly mentions limitations for "extremely large graphs". Further discussion of this and other limitations would improve the paper and make its scope clearer.
-	A figure showing both classification accuracy and computation time would be welcome to make clear, for example, that while GCN-JK has a comparable computation time to GAMMA, it has significantly lower accuracy across the selected datasets. A figure highlighting the compute to accuracy trade-off would prevent the reader from having to go back and forth from Table 1 to Figure 5.


Lim, D., Hohne, F. M., Li, X., Huang, S. L., Gupta, V., Bhalerao, O. P., & Lim, S.-N. (2021). Large scale learning on non-homophilous graphs: New benchmarks and strong simple methods. In Advances in Neural Information Processing Systems. https://openreview.net/forum?id=DfGu8WwT0d

Platonov, O., Kuznedelev, D., Diskin, M., Babenko, A., & Prokhorenkova, L. (2023). A critical look at the evaluation of GNNs under heterophily: Are we really making progress? In The Eleventh International Conference on Learning Representations. https://openreview.net/forum?id=tJbbQfw-5wv

---

> ### Author Rebuttal · Authors · 2025-07-31
>
> We are grateful for your assessment of our work, and your constructive suggestions for improving the paper. We are especially pleased that you found GAMMA to be well-motivated and the paper to be clear and well-written.
>
> - Evaluation on Large-Scale Graphs:
>
> The reviewer raised a critical point regarding the evaluation being limited to smaller graphs. Demonstrating scalability on larger benchmarks is crucial for validating GAMMA's applicability in realistic settings. To address this, we have benchmarked GAMMA against all of the baselines on seven larger heterophilic graphs, including those from the families of benchmarks mentioned by the reviewer:
>
> $$
> \\begin{array}{l|ccccccc}
> \\hline
> & \\text{ogbn-arxiv} & \\text{roman-empire} & \\text{ogbn-products} & \\text{minesweeper} & \\text{amazon-ratings} & \\text{amazon-photos} & \\text{tolokers} \\\\
> \\hline
> \\text{MLP}   & 47.34 \\pm 0.09 & 65.37 \\pm 0.58 & 43.20 \\pm 0.01 & 79.48 \\pm 0.21 & 38.35 \\pm 0.20 & 48.73 \\pm 1.80 & 78.16 \\pm 0.00 \\\\
> \\text{GCN}   & 64.91 \\pm 0.26 & 54.16 \\pm 0.28 & \\mathbf{71.31 \\pm 0.16} & 80.31 \\pm 0.24 & 42.12 \\pm 0.29 & 52.14 \\pm 1.50 & 78.76 \\pm 0.17 \\\\
> \\text{SGC}   & 63.47 \\pm 0.16 & 61.20 \\pm 0.65 & \\text{OOM} & 80.05 \\pm 0.03 & 41.61 \\pm 0.72 & 57.97 \\pm 0.52 & 78.56 \\pm 0.19 \\\\
> \\text{GCN2}  & 61.96 \\pm 0.42 & 64.96 \\pm 0.38 & 61.08 \\pm 0.18 & 82.80 \\pm 0.35 & 47.20 \\pm 0.55 & 94.10 \\pm 0.25 & 79.80 \\pm 0.30 \\\\
> \\text{GCNJK} & 70.19 \\pm 0.11 & 62.18 \\pm 0.91 & 69.46 \\pm 0.33 & 85.15 \\pm 0.42 & 48.77 \\pm 0.37 & 95.28 \\pm 0.05 & \\mathbf{81.23 \\pm 0.26} \\\\
> \\text{GATJK} & \\mathbf{70.75 \\pm 0.19} & 70.00 \\pm 0.70 & \\text{OOM} & \\mathbf{85.65 \\pm 0.34} & \\mathbf{50.84 \\pm 0.52} & 95.71 \\pm 0.11 & \\mathbf{81.31 \\pm 0.48} \\\\
> \\text{H2GCN} & \\text{OOM} & 75.89 \\pm 0.55 & \\text{OOM} & 83.31 \\pm 0.65 & 42.90 \\pm 0.38 & 84.95 \\pm 0.38 & 79.39 \\pm 0.42 \\\\
> \\text{GAT}   & 70.01 \\pm 0.09 & 74.50 \\pm 0.45 & \\text{OOM} & 81.52 \\pm 0.26 & 48.89 \\pm 0.61 & 95.56 \\pm 0.15 & 78.63 \\pm 0.06 \\\\
> \\text{MixHop}& 70.27 \\pm 0.14 & \\mathbf{82.20 \\pm 0.31} & \\text{OOM} & 84.88 \\pm 0.45 & \\mathbf{50.98 \\pm 0.30} & \\mathbf{96.34 \\pm 0.30} & 81.03 \\pm 0.70 \\\\
> \\text{BernNet}& 63.15 \\pm 0.13 & 66.27 \\pm 0.35 & \\text{OOM} & 80.00 \\pm 0.07 & 40.90 \\pm 0.09 & 62.34 \\pm 7.02 & 78.16 \\pm 0.00 \\\\
> \\text{GPRGNN}& 67.66 \\pm 0.07 & 69.60 \\pm 0.40 & \\mathbf{73.26 \\pm 0.65} & 80.36 \\pm 0.19 & 43.44 \\pm 0.13 & 79.54 \\pm 0.39 & 78.36 \\pm 0.10 \\\\
> \\text{M2MGNN}& \\mathbf{72.52 \\pm 0.10} & \\mathbf{80.81 \\pm 0.70} & \\text{OOM} & \\mathbf{85.83 \\pm 0.25} & \\mathbf{50.57 \\pm 0.29} & \\mathbf{95.73 \\pm 0.25} & 80.66 \\pm 0.47 \\\\
> \\text{GAMMA} & \\mathbf{71.81 \\pm 0.19} & \\mathbf{81.37 \\pm 0.25} & \\mathbf{72.61 \\pm 0.04} & \\mathbf{87.58 \\pm 0.25} & 49.07 \\pm 0.65 & \\mathbf{96.28 \\pm 0.11} & \\mathbf{82.59 \\pm 0.41} \\\\
> \\hline
> \\end{array}
> $$
>
>
> As the new results demonstrate, GAMMA's performance is not only maintained but excels at a larger scale. It achieves state-of-the-art or second-best accuracy on 5 of the 7 datasets.
>
> For a fair and rigorous comparison, all models were trained for 500 epochs across 5 random splits, using a 60/20/20 train/validation/test split. A consistent learning rate, dropout, and weight decay were used for all models, which were implemented in their baseline forms without extra architectural enhancements like residual connections.
>
> Our primary goal with GAMMA was to design an architecture that makes full-batch training on heterophilic graphs feasible and efficient, even on commodity hardware. However, our analysis confirmed that many state-of-the-art models (H2GCN, MixHop or M2MGNN) fail on even medium-scale graphs (e.g., roman-empire) when using a typical commodity GPU like the NVIDIA A2000 (12 GB VRAM). To provide a fair comparison, we utilized a more powerful NVIDIA A100 with 80 GB of memory. Even with this substantial resource, many models still encountered Out-Of-Memory (OOM) errors, particularly on large datasets like ogbn-products. These failures are a direct consequence of specific architectural choices common in heterophily-focused GNNs:
>
> Feature Concatenation Bottleneck: Models like H2GCN, MixHop, and Jumping Knowledge Networks (JK-Nets) explicitly concatenate feature representations from different hops or layers. This strategy causes the feature dimensionality to grow linearly or even exponentially, leading to massive intermediate tensors that are fundamentally unscalable for large graphs.
>
> Complex Aggregation Overhead: Other models suffer from computationally intensive operations. For example, BernNet's K-th order filter requires K distinct propagation steps, storing all intermediate results in memory. Further, M2MGNN performs complex per-edge operations, including a small MLP and a segmented softmax, which cannot be efficiently batched and become prohibitive on graphs with millions of edges.
>
> However, GAMMA's architectural design directly addresses these scalability challenges by employing a single, shared projection matrix. It ensures all hop embeddings reside in a coherent, fixed-dimension feature space. This entirely avoids the feature dimensionality explosion that exists in concatenation-based models. Further, the iterative gating mechanism adaptively computes a compact representation by selectively weighting hop information, rather than stacking it. This selective fusion process eliminates the need for concatenation, drastically reducing memory footprint.
>
> Together, these elements create a streamlined framework that discards redundant information on-the-fly while preserving essential multi-hop patterns. This combination of parameter efficiency and an adaptive, non-expansive aggregation mechanism is what enables GAMMA to scale effectively where other methods fail.
>
>
> - Accuracy-to-Efficiency Trade-off Figure:
>
> This is an excellent suggestion for improving the paper's clarity. Visualizing the accuracy-to-efficiency trade-off directly would indeed prevent readers from having to cross-reference Table 1 and Figure 5. In the revised manuscript, we will add a 2D scatter plot where the x-axis represents computation time (or memory) and the y-axis represents accuracy.
>
> - Discussion of Limitations:
>
> Indeed, a more detailed discussion of limitations strengthens the paper and clarifies its scope. We will expand the limitations paragraph in the conclusion to be more comprehensive.
>
> While GAMMA scales well, for web-scale graphs with billions of edges, the one-time pre-computation of multi-hop adjacencies ($A^k$) could become a bottleneck if not handled with sparse matrix formats. We will discuss this and mention potential mitigation strategies like on-the-fly neighborhood sampling. Further, the fixed number of routing iterations (R) is a hyperparameter that might be suboptimal for graphs with highly varied neighborhood complexities. We will explicitly state this trade-off and reference potential future work on adaptive iteration counts.

---

> > ### Comment · Reviewer_1qWb · 2025-08-01
> > **Thank you**
> >
> > Thank you for the response. This has addressed my concerns about evaluation on larger scale graphs, and I am happy that the authors will incorporate the suggestions of this review in the revised paper. I will maintain my score and recommendation that this paper be accepted.

---

### Official Review · Reviewer_QbS7 · 2025-07-03

**Clarity:** 3
**Significance:** 2
**Originality:** 2
**Rating:** 4
**Confidence:** 2

**Summary:**

The authors propose a new architecture for multi-hop aggregation graph learning. Previous work captured heterophilic patterns in graphs by concatenating $p$-hop neighborhoods but this approach often suffers from high memory requirements. The authors instead propose a gated multi-hop message passing mechanism (GAMMA) which essentially involves computing a weighted sum of hop embeddings (gating coefficient dictate the weights in the sum), thereby avoiding concatenation. Additionally, there is an iterative update for the gating coefficient that happens each forward pass of GAMMA. GAMMA is evaluated against a variety of standard message passing NNs and k-hop message passing NNs and shows competitive classification accuracy with greater efficiency (in terms of memory and inference time)

**Questions:**

1. How are the hop-specific embeddings generated? Is the GAMMA layer something that comes after standard multi-hop message passing layer? Is it that if I have a node $v$ and its $p$-hop neighbors $\mathcal{N}^p$ with some computed embeddings, there is some gating coefficient $s_i^{(t)}$ learned per $p$-hop neighborhood and then the new node feature is generated by taking a weighted sum over the $p$-hop neighbors with the gating coefficients as the weights? I guess the gating coefficient is based on by the set of gating logits $b_i$ but are these gating logics learned? I think there is a good summary of GAMMA's gating mechanism in lines 381-384 that maybe should have been moved to the earlier section where GAMMA is introduced.
2. Is there any additional computation cost during training time due to the $R$ routing iterations for the GAMMA gating mechanism? Can the number of iterations can be controlled in some data-drive way (see [1] where they control the number of iterations through another learnable MLP module)?
3. It seems that the performance for GAMMA suffers a bit on homophilic datasets (which makes sense because the focus of GAMMA is for heterophilic benchmarks) but is there a way to see how the performance of GAMMA changes as the level of homophily in the datasets change?

[1] "Towards scale-invariant graph-related problem solving by iterative homogeneous gnns"

**Ethical Concerns:**

["NO or VERY MINOR ethics concerns only"]

**Final Justification:**

The authors addressed my initial questions. I think GAMMA shows convincing performance gains in terms of efficiency (especially the faster inference as compared to current SoTA models). However, as mentioned by Reviewer DJEL, it seems that GAMMA's improvements on heterophilic datasets (especially large scale datasets) still leave something to be desired. When balancing these two points, I would like to keep my original score of 4.

**Limitations:**

yes

**Quality:**

2

**Strengths And Weaknesses:**

**Strengths**
- Strong experimental baseline comparisons and the tables regarding computation efficiency are convincing. GNNs often suffer from long training times (especially on large graphs) since a forward pass depends on the number of nodes/edges in the input graph. This is probably even more of a problem with multi-hop GNNs so the efficiency of GAMMA (I think due to the fact that they avoid the computational cost associated with concatenation of multi-hop node features) as compared to previous multi-hop techniques seems significant.

**Weaknesses**
- Explanation of GAMMA is a bit unclear, I have some questions below that hopefully the authors can clarify.
- Minor typo: in table 1, I think GCN shows the best accuracy for CiteSeer and should be highlighted in green.

---

> ### Author Rebuttal · Authors · 2025-07-31
>
> We sincerely thank the reviewer for the positive assessment of our work, particularly for recognizing the significance of GAMMA's efficiency and the strength of our experimental comparisons. We also appreciate the meticulous review and thoughtful questions, which will help us improve the paper's clarity.
>
> Regarding the minor typo in Table 1, indeed GCN shows the best accuracy for CiteSeer and should be highlighted in green. Thank you for catching this; we will make this correct.
>
> Below, we address the questions in detail.
>
> 1. Clarification of the GAMMA Mechanism
>
> Thank you for these detailed questions; they highlight areas where we can be clearer in our explanation. The GAMMA layer is a self-contained module that integrates multi-hop propagation and adaptive gating. To break down the process step-by-step and for a more formal description, we also refer to the pseudocode in Algorithm 1 (Appendix A.2 provided in the supplementary material).
>
> - How are the hop-specific embeddings generated? The hop-specific embeddings $H^{(p)}$ are generated using a standard multi-hop aggregation process. First, all node features $X$ are projected into a common embedding space via a shared linear transformation, $H = XW$. From there, we generate the features for each hop distance by iteratively applying the graph propagation operator (the normalized adjacency matrix $A$). For instance, the process is as follows:
>
> The 1-hop embedding is calculated as $H^{(1)} = AH$.
> The 2-hop embedding is then $H^{(2)} = A(H^{(1)}) = A(AH)$.
> The 3-hop embedding follows as $H^{(3)} = A(H^{(2)}) = A(A(AH)) = A^3H$.
>
> This iterative propagation, formally expressed as $H^{(p)} = A H^{(p-1)}$, is used to aggregate information from higher-order neighborhoods. It is conceptually similar to aggregation strategies used in other multi-hop models like MixHop or H2GCN, which also leverage powers of the adjacency matrix. GAMMA integrates this standard propagation step before applying its novel gating mechanism.
>
> - How are the gating coefficients generated and are the logits learned? As mentioned by the reviewer, it is correct that the final node feature is a weighted sum of the hop embeddings. However, a key distinction is that the gating coefficients $\alpha_{i,p}$ are not learned parameters in the traditional sense. They are dynamically computed for each node $i$ during every forward pass through an iterative refinement process.
>
> For each node $i$, we initialize a vector of gating logits $b_i$ to all zeros. In the first iteration, passing these zero logits through a softmax function results in uniform coefficients $\alpha_{i,p} = 1/(K+1)$, ensuring all hops are treated equally at the start. Next, in each routing iteration $t$, we compute a temporary node representation $v_i^{(t)}$ by taking a weighted sum of the hop embeddings using the current coefficients and applying a squash normalization function. We then measure the agreement between this representation and each hop's embedding via a dot product: $H_{i}^{(p)} \cdot v_{i}^{(t)}$. This agreement score directly updates the logits: $b_{i,p}^{(t+1)} = b_{i,p}^{(t)} + \text{agreement}$. If a hop's embedding aligns well with the node's evolving representation, its corresponding logit increases, which in turn increases its influence in the next iteration. After $R$ iterations, the final representation $v_i^{(R)}$ is used as the output of the GAMMA layer.
>
> So, to summarize, the gating logits are not learned via backpropagation; they are iteratively refined on-the-fly based on feature agreement during the forward pass. This allows each node to adapt its information aggregation strategy based on its specific local neighborhood structure.
>
> - Organization Suggestion: Thank you for this suggestion. We agree that the summary in lines 381-384 provides a very clear overview of the global vs. local adaptation mechanism. We will revise the manuscript to move this explanation to an earlier point in Section 4 to improve the clarity and flow of the introduction to GAMMA.
>
>
> 2. Computational Cost and Data-Driven Iterations
>
> - The $R$ routing iterations do introduce an additional computational step. However, this overhead is minimal and the design is highly efficient for two key reasons:
>
> Lightweight Operations: The operations inside the routing loop (dot products, softmax, weighted sums) are performed on small, fixed-size tensors ($d_{out}$-dimensional vectors). These are highly parallelizable and computationally cheap compared to the main graph convolution operations.
>
> Memory Efficiency: Crucially, GAMMA avoids feature concatenation. Unlike models like H2GCN or MixHop that expand feature dimensionality to $(K+1) \times d_{out}$ , GAMMA maintains a single output vector of dimension $d_{out}$. This prevents the creation of large intermediate tensors, which is a primary cause of high memory usage and long latency in GNNs due to irregular memory access patterns. This design choice is why GAMMA achieves significant speedups (up to 20x) and memory savings (up to 12x).
>
> - Data-Driven Control of Iterations: In our current framework, the number of iterations $R$ is a fixed hyperparameter. Our ablation study in Appendix A.1 shows that performance is robust and typically optimal with a small, fixed $R$ (e.g., 2-4 iterations).
>
> - Introducing a learnable module to control: $R$ in a data-driven way is a potential direction for future research. It could offer finer-grained control at the cost of increased model complexity.
>
> 3. Performance on Homophilic Datasets
>
> This is an important point about the model's generalizability. Our goal with GAMMA was to create a homophily-agnostic model, not one that is purely specialized for heterophily. The experimental results in Table 1 show that GAMMA performs very competitively on homophilic benchmarks. It achieves the best accuracy on PubMed ($89.62\%$) and is on par with the top performers on Cora ($87.42\%$). This demonstrates that the adaptive gating mechanism is not detrimental when strong homophilic signals are present. In such cases, the routing mechanism simply learns to assign the highest weights to the 0-hop (self-features) and 1-hop embeddings, effectively recovering the behavior of a standard GNN.
>
> The performance across the spectrum of datasets in Table 1 illustrates how GAMMA's performance changes with the level of homophily. While many standard GNNs see a sharp performance drop when moving from homophilic graphs (e.g., Cora, ratio 0.81) to heterophilic ones (e.g., Texas, ratio 0.11), GAMMA maintains strong and often state-of-the-art results across the board. This robustness is a key strength of its adaptive design, allowing it to excel regardless of the underlying graph structure.

---

> > ### Comment · Reviewer_QbS7 · 2025-08-05
> > **Response to authors**
> >
> > Thanks for the detailed response. I maintain my score.

---

### Official Review · Reviewer_hXHc · 2025-07-23

**Clarity:** 3
**Significance:** 2
**Originality:** 2
**Rating:** 4
**Confidence:** 2

**Summary:**

The paper demonstrates through comprehensive empirical analysis that state of the art GNNs are not computationally efficient in memory and runtime. To address these limitations authors propose GAMMA that leverages multihop gated message passing and weight sharing across hops to reduce both memory and runtime. GAMMA is show to match or exceed state of the art GNNs on homophilous and heterophilous benchmarks.

**Questions:**

1. How does GAMMA perform on larger datasets in comparison to state of the art GNNs?
2. Are state of the art GNNs feasible to train on larger datasets given a reasonable budget of compute, time, and memory?
3. How are GAMMAs designs (gated multihop message passing and shared weight across hops) related and different to designs proposed in the literature?

**Ethical Concerns:**

["NO or VERY MINOR ethics concerns only"]

**Final Justification:**

My initial concerns about scalability and novelty are mostly resolved. On the scalability front, the authors did an excellent job in the rebuttal with positive results for their method and also important findings about the feasibility of other popular methods. This is also an overlooked component in GNN papers. On the novelty front, I don't think the designs are highly novel and I still think they are quite close to existing models in the literature, most notably GPR-GNN, but the authors have argued that there are key differences in their approach and have demonstrated that GAMMA's performance is higher than GPR-GNN across a range of benchmarks. I have thus increased my score from 3 to 4.

**Limitations:**

Yes.

**Quality:**

3

**Strengths And Weaknesses:**

**Strengths**
1. The problem is well-motivated and a fair and rigorous comparison of memory/runtime requirements for state of the art GNNs is a gap in the literature as far as I'm aware. Often times, this is an overlooked aspect in GNN design.
2. The two designs are intuitive. It is clear how they address and mitigate the memory/runtime issues of state of the art GNNs
3. The paper is clear and well written.

**Weaknesses**
1. My main weakness is the evaluation. The paper does a good job setting up the problem and also demonstrating the runtime and memory requirements across many state of the art GNNs, but only evaluates on relatively small datasets (table 1). Texas, Cornell, and Wisconsin each range in 100s of nodes with some dataset construction issues (e.g. only one node per class). Squirrel and Chameleon have been identified as having issues as well with many duplicate nodes, leading to test set leakage [1]. There have been many proposed heterophilous datasets since these original smaller ones [1, 2]. Since memory and runtime are the main motivation, I think it would be good to include (i) GAMMAs performance on these larger datasets and also (ii) comparisons to state of the art GNNs on these larger datasets since the large datasets are where these factors play the most important role. These would address questions I have about if GAMMA's performance remains matched with state of the art for larger datasets. I would also be very interested to see a fair and rigorous comparison of the feasibility of training large scale GNNs. is it the case that expensive GNNs like H2GCN is infeasible to train with even a moderate amount of compute, memory, and runtime? If so, I think these would make the contributions stronger.
2. I was also wondering how novel the designs are (multihop gating and weight sharing across hops)? It seems that these have been used in the literature each individually and comparing your specific designs to others that are similar would be helpful to appreciate the contributions.

[1] Platanov et al., A critical look at the evaluation of GNNs under heterophily: Are we really making progress?
[2] Lim et al., Large Scale Learning on Non-Homophilous Graphs: New Benchmarks and Strong Simple Methods

---

> ### Author Rebuttal · Authors · 2025-07-31
>
> We appreciate the reviewer’s constructive feedback. Below we address each concern with comprehensive experiments and analysis.
>
> 1. **Scalability on Large‑Scale Datasets**
> The reviewer’s primary concern is that our evaluation is limited to smaller datasets, questioning whether GAMMA’s performance holds on larger graphs where memory and runtime are paramount. To address this, we have benchmarked GAMMA and a comprehensive set of baselines on seven large‑scale heterophilic graphs.
>
> $$
> \\begin{array}{l|ccccccc}
> \\hline
> & \\text{ogbn-arxiv} & \\text{roman-empire} & \\text{ogbn-products} & \\text{minesweeper} & \\text{amazon-ratings} & \\text{amazon-photos} & \\text{tolokers} \\\\
> \\hline
> \\text{MLP}   & 47.34 \\pm 0.09 & 65.37 \\pm 0.58 & 43.20 \\pm 0.01 & 79.48 \\pm 0.21 & 38.35 \\pm 0.20 & 48.73 \\pm 1.80 & 78.16 \\pm 0.00 \\\\
> \\text{GCN}   & 64.91 \\pm 0.26 & 54.16 \\pm 0.28 & \\mathbf{71.31 \\pm 0.16} & 80.31 \\pm 0.24 & 42.12 \\pm 0.29 & 52.14 \\pm 1.50 & 78.76 \\pm 0.17 \\\\
> \\text{SGC}   & 63.47 \\pm 0.16 & 61.20 \\pm 0.65 & \\text{OOM} & 80.05 \\pm 0.03 & 41.61 \\pm 0.72 & 57.97 \\pm 0.52 & 78.56 \\pm 0.19 \\\\
> \\text{GCN2}  & 61.96 \\pm 0.42 & 64.96 \\pm 0.38 & 61.08 \\pm 0.18 & 82.80 \\pm 0.35 & 47.20 \\pm 0.55 & 94.10 \\pm 0.25 & 79.80 \\pm 0.30 \\\\
> \\text{GCNJK} & 70.19 \\pm 0.11 & 62.18 \\pm 0.91 & 69.46 \\pm 0.33 & 85.15 \\pm 0.42 & 48.77 \\pm 0.37 & 95.28 \\pm 0.05 & \\mathbf{81.23 \\pm 0.26} \\\\
> \\text{GATJK} & \\mathbf{70.75 \\pm 0.19} & 70.00 \\pm 0.70 & \\text{OOM} & \\mathbf{85.65 \\pm 0.34} & \\mathbf{50.84 \\pm 0.52} & 95.71 \\pm 0.11 & \\mathbf{81.31 \\pm 0.48} \\\\
> \\text{H2GCN} & \\text{OOM} & 75.89 \\pm 0.55 & \\text{OOM} & 83.31 \\pm 0.65 & 42.90 \\pm 0.38 & 84.95 \\pm 0.38 & 79.39 \\pm 0.42 \\\\
> \\text{GAT}   & 70.01 \\pm 0.09 & 74.50 \\pm 0.45 & \\text{OOM} & 81.52 \\pm 0.26 & 48.89 \\pm 0.61 & 95.56 \\pm 0.15 & 78.63 \\pm 0.06 \\\\
> \\text{MixHop}& 70.27 \\pm 0.14 & \\mathbf{82.20 \\pm 0.31} & \\text{OOM} & 84.88 \\pm 0.45 & \\mathbf{50.98 \\pm 0.30} & \\mathbf{96.34 \\pm 0.30} & 81.03 \\pm 0.70 \\\\
> \\text{BernNet}& 63.15 \\pm 0.13 & 66.27 \\pm 0.35 & \\text{OOM} & 80.00 \\pm 0.07 & 40.90 \\pm 0.09 & 62.34 \\pm 7.02 & 78.16 \\pm 0.00 \\\\
> \\text{GPRGNN}& 67.66 \\pm 0.07 & 69.60 \\pm 0.40 & \\mathbf{73.26 \\pm 0.65} & 80.36 \\pm 0.19 & 43.44 \\pm 0.13 & 79.54 \\pm 0.39 & 78.36 \\pm 0.10 \\\\
> \\text{M2MGNN}& \\mathbf{72.52 \\pm 0.10} & \\mathbf{80.81 \\pm 0.70} & \\text{OOM} & \\mathbf{85.83 \\pm 0.25} & \\mathbf{50.57 \\pm 0.29} & \\mathbf{95.73 \\pm 0.25} & 80.66 \\pm 0.47 \\\\
> \\text{GAMMA} & \\mathbf{71.81 \\pm 0.19} & \\mathbf{81.37 \\pm 0.25} & \\mathbf{72.61 \\pm 0.04} & \\mathbf{87.58 \\pm 0.25} & 49.07 \\pm 0.65 & \\mathbf{96.28 \\pm 0.11} & \\mathbf{82.59 \\pm 0.41} \\\\
> \\hline
> \\end{array}
> $$
>
> The table demonstrates GAMMA's consistently high performance on large-scale heterophilic graphs, achieving state-of-the-art comparable results and often the best or second-best accuracy across 5 out of 7 datasets, showcasing its effectiveness at scale. The models were trained for 500 epochs across 5 random splits, using a 60/20/20 train/validation/test split, with all models utilizing the same learning rate, dropout, and weight decay, and their baseline GNN implementations without additional architectural enhancement or residual connections. As shown, many state-of-the-art models encountered Out-Of-Memory (OOM) errors, particularly on larger datasets like ogbn-products, due to architectural choices such as explicit concatenation of feature representations (e.g., H2GCN, JK-Nets) or complex per-edge operations (e.g., M2MGNN) that lead to a significant increase in feature dimensionality or large intermediate tensors. GAMMA addresses these issues through weight sharing to maintain a coherent feature space, dynamic gating for selective and efficient hop information propagation, and streamlined message passing that discards redundant information, enabling it to scale effectively.
>
> 2. **Feasibility of Training Large‑Scale Heterophilic Graphs on Commodity GPUs**
> Our primary goal was to enable full‑batch training for heterophilic graphs, designing for efficiency from the start, rather than treating it as an afterthought. Unfortunately, with commodity GPUs like the NVIDIA A2000 (12 GB VRAM), many state‑of‑the‑art models fail even on medium‑scale datasets like *roman‑empire* (22.7 K nodes, 32.9 K edges). For fair comparison on larger datasets, we employed an NVIDIA A100 with 80 GB memory to enable full‑batch training for prior work as well. However, even with this substantial memory, resource‑hungry models still encounter Out‑Of‑Memory (OOM) errors on large‑scale datasets like *ogbn‑products* (2.4 M nodes, 61 M edges). The OOM failures for such models stem directly from their architectural choices:
>
> • Concatenation‑based models (H2GCN, MixHop, GCN/GAT‑JK): These explicitly concatenate feature representations from different hops or layers. H2GCN’s final representation size explodes to \\((2^{k+1}-1) \\times \\text{hidden\\_channels}\\), MixHop expands channel dimensions at each layer (\\(\\text{hidden\\_channels} \\times 3\\)), and JK‑Nets concatenate outputs from all layers. This linear or exponential growth in feature dimensionality is fundamentally unscalable.
>
> • Complex Propagation/Aggregation (BernNet, M2MGNN): BernNet’s K‑th order filter requires K distinct propagation steps, storing all intermediate results. M2MGNN performs complex per‑edge operations (MLP and segmented softmax), creating large intermediate tensors prohibitive on graphs with millions of edges.
>
> **GAMMA’s Solution:** Our method addresses these issues through: (1) weight sharing that maintains a single coherent feature space rather than expanding dimensions, (2) dynamic gating that selectively propagates only informative hop information, eliminating the need for feature concatenation, and (3) efficient message passing that discards redundant information while preserving essential multi‑hop patterns.
>
> 3. **Novelty and Technical Contributions**
>
> While concepts like “gating” and “weight sharing” exist individually, GAMMA’s novelty lies in their unique synthesis and specific design for heterophilic graphs. We provide detailed positioning in Appendix A.3 submitted in the supplementary material.
>
> **Weight Sharing Across Hops:** Our approach differs fundamentally from simple parameter reduction. By applying a single shared linear projection \\(W\\) before propagation, we ensure all hop‑specific embeddings \\(H^{(p)}\\) reside in a coherent, shared geometric space, a critical prerequisite for our gating mechanism’s dot‑product agreement measure. Architectures using independent hop‑specific weights \\(W^{(p)}\\) map features to potentially unrelated latent spaces, making direct comparisons ill‑defined. Our lightweight, learnable scaling vectors \\(\\gamma_p\\) reintroduce hop‑specific adaptability without fracturing this shared space.
>
> **Gated Multi‑hop Message Passing:** Our new design is the dynamic, node‑specific, and iterative gating mechanism, fundamentally different from existing approaches:
>
> • Comparison with Spectral Methods (GPRGNN, BernNet): These learn global polynomial filter coefficients applied uniformly across the graph. GAMMA’s gating coefficients \\(\\alpha_{i,p}\\) are computed per‑node and activation‑driven, recalculated in each forward pass to adapt to specific local structures around each node.
>
> • Comparison with JK‑Nets (GCNJK, GATJK): JK‑Nets aggregate representations from different layers at the network’s end using mechanisms like max or concatenation. GAMMA’s gating iteratively refines hop contributions based on agreement within the message‑passing layer itself, enabling more nuanced information fusion.
>
> • Comparison with GAT: GAT computes attention over a node’s 1‑hop neighborhood to weigh individual neighbors. GAMMA operates at a higher abstraction level, weighing aggregated messages from entire hop‑distances rather than individual neighbors.
>
> Additionally, the theoretical analysis in Appendix A.4 (provided in the supplementary material) further demonstrates how this design enables adaptive information propagation while maintaining computational efficiency.

---

> > ### Comment · Reviewer_hXHc · 2025-08-02
> >
> > I sincerely appreciate the authors responding to my scalability concerns. The authors carefully designed these additional experiments to allow for fair comparisons, benchmarking on an NVIDIA A100 with 80 GB memory and using comparable hyperparameters across models. Even on these much larger gpus, many of the methods time out on OGBN-products, which is an important finding and critical to address. My concerns about scalability are completely resolved.
> >
> > I am still concerned about the novelty as it pertains to GPR-GNN. This concern was also raised by Reviewer DJEL. In the author's response, it is stated:
> >
> > "However, the key distinction from GPR-GNN is architectural: GPR-GNN applies its coefficients (which can be negative) uniformly across all nodes, while GAMMA combines negative-capable global scaling with positive node-specific routing coefficients $\alpha_{i, p}$ (from softmax)."
> >
> > I acknowledge that the performance is higher for the proposed approach, but I think its important to verify and understand that the difference can really be attributed to the node-specific coefficients $\alpha_{i, p}$. Is it feasible to conduct an ablation study where setting $\alpha_{i, p}$ equal for all $p$ recovers GPR-GNN? An ablation study that recovers GPR-GNN performance would address my novelty concerns as it pertains to GPR-GNN. I also think a theoretical justification for changing the global coefficients to be node-specific would be beneficial to further distinguish the approach from GPR-GNN.

---

> > > ### Author Response · Authors · 2025-08-06
> > >
> > > We appreciate this constructive feedback, as it helps clarify the conceptual distinction underlying our approach. We conducted experiments on six heterophilic datasets, comparing two scenarios: when the gating coefficients are equal for all hops versus when they are dynamically changing based on our iterative gating mechanism. In the table below (each row aggregates from the listed hops), "Without" refers to the disabled gating scenario (equal hop weights for all nodes), while "With" refers to the enabled gating scenario (adaptive hop weighting).
> > > $$
> > > \\begin{array}{l|cc|cc|cc|cc|cc|cc}
> > > \\hline
> > > & \\text{Cornell} & & \\text{Texas} & & \\text{Wisconsin} & & \\text{Chameleon} & & \\text{Actor} & & \\text{Squirrel} \\\\
> > > \\textbf{} & \\text{With} & \\text{Without} & \\text{With} & \\text{Without} & \\text{With} & \\text{Without} & \\text{With} & \\text{Without} & \\text{With} & \\text{Without} & \\text{With} & \\text{Without} \\\\
> > > \\hline
> > > \\text{1 and 2 hop}       & 78.68 & 75.31 & 87.37 & 83.25 & 86.27 & 81.40 & 51.16 & 44.79 & 35.59 & 30.72 & 36.20 & 28.33 \\\\
> > > \\text{1, 2 and 3 hop}     & 77.11 & 73.51 & 86.47 & 82.57 & 84.12 & 79.02 & 51.14 & 42.29 & 36.06 & 30.29 & 35.43 & 29.28 \\\\
> > > \\text{1, 2, 3 and 4 hop}   & 79.21 & 75.99 & 86.21 & 81.86 & 84.71 & 80.06 & 50.87 & 42.40 & 34.90 & 27.25 & 35.52 & 27.42 \\\\
> > > \\hline
> > > \\end{array}
> > > $$
> > > To clarify the experimental setup: "Without" refers to disabling node-specific dynamic gating while retaining the learnable channel-wise scaling factors $γ_p$ (Section 5 of our paper) that operate globally across the entire graph for each hop $p$. In this configuration, GAMMA reduces to a weight-sharing architecture similar to GPR-GNN but enhanced with global-level channel-wise scaling, allowing us to isolate the contribution of node-adaptive routing from our overall architectural improvements.
> > >
> > > GAMMA’s dynamic gating coefficients consistently outperform uniform hop weighting ($\alpha_{i,p}$ equal for all hops) across all six datasets, with improvements of 2–8% depending on the dataset. These gains stem from GAMMA’s ability to adaptively capture node-specific multi-hop homophily patterns. As shown in Section 2 of our paper, heterophily varies both across nodes and hop distances: the node homophily ratio can differ dramatically between nodes and across hops, resulting in multi-hop structural heterogeneity. For instance, sampling two nodes from the Chameleon dataset reveals dramatically different multi-hop homophily patterns:
> > >
> > > Node A:
> > >
> > > - 1-hop neighbors: 17% same class, 83% different class (heterophilic)
> > >
> > > - 2-hop neighbors: 74% same class, 26% different class (homophilic)
> > >
> > > Node B:
> > >
> > > - 1-hop neighbors: 61% same class, 39% different class (homophilic)
> > >
> > > - 2-hop neighbors: 24% same class, 76% different class (heterophilic)
> > >
> > > Thus, the multi-hop neighborhood pattern for Node A differs completely from Node B. Even for nodes with the same 2-hop homophily, e.g., 90% vs. 55% same-class neighbors, optimal aggregation strategies may differ, highlighting the need for fine-grained, node-specific weighting instead of uniform schemes.
> > >
> > > Based on our analysis, Chameleon exhibits only 20.7% pattern consistency, meaning just 20.7% of nodes share similar multi-hop homophily trajectories (like Node A), while most show diverse, complex patterns. In contrast, Cornell displays 65.3% pattern dominance, where the majority of nodes share similar multi-hop homophily, resulting in less structural variation. This explains why Chameleon sees the largest performance gains from dynamic gating, while Cornell benefits least, illustrating the advantage of node-adaptive multi-hop aggregation.
> > >
> > > Unlike GPR-GNN and MixHop, GAMMA’s architectural design is rooted in the insight that effective heterophilic modeling requires separating global pattern learning from local pattern identification:
> > >
> > > - Global heterophily patterns (e.g., “2-hop information is generally more useful than 1-hop for this graph type”) should be learned via parameter optimization for each hop during training.
> > >
> > > - Local heterophily patterns (e.g., “for this node in this context, 1-hop information contradicts the current representation, while 3-hop information reinforces it”) must be dynamically identified during both training and inference.
> > >
> > > GPR-GNN addresses the first point by learning global coefficients to capture dataset-level trends, but cannot adapt to node-specific heterophily. In contrast, GAMMA’s iterative gating mechanism operates dynamically at both training and inference time, enabling adaptation to unknown heterophily patterns during each forward pass.
> > >
> > > In GAMMA, each hop’s embedding $\hat{H}^{(p)}_i$ serves as a prediction for the node’s final representation, with the agreement score $\hat{H}^{(p)}_i \cdot v^{(t)}_i$ indicating alignment with the node’s current state. High agreement means the $p$-hop neighborhood supports the node’s evolving identity, while low agreement indicates contradictory or noisy information that should be downweighted.

---

> ### Comment · Reviewer_hXHc · 2025-08-07
>
> Thank you for the extensive empirical verification. Based on the authors' extensive responses, I will increase my score. In the revision, I believe it is critical to address the points raised by myself and other reviewers about scalability and novelty. For scalability, the authors have shown important findings, describing that many models are not feasible to train on commodity GPUs (e.g., NVIDIA A2000) even for smaller sized datasets (Roman-Empire, Amazon Ratings). If GAMMA can run on these smaller GPUs on the larger datasets like ogbn-products, that is an even bigger win. Even on larger GPUs, many of the state of the art models time out and GAMMA remains competitive with the remaining models. For novelty, GAMMA does seem very similar to GPR-GNN, but these ablations have demonstrated the difference lying in the adaptive node specific weights. I believe this distinction is key to highlight in the revision.

---

### Decision · Program_Chairs · 2025-09-17

**Decision:**

Accept (poster)

**Comment:**

This paper proposes a gated multi-hop message passing framework that adaptively integrates information across different hop distances, enabling expressive representations without increasing dimensionality beyond that of single-hop GNNs. It is shown experimentally to combine competitive accuracy (matching or exceeding SOTA heterophilic GNNs) with significantly improved inference speed.

During the rebuttal, the authors provided additional scalability experiments, stronger comparisons, and clarifications on novelty. While the framework does not consistently outperform the state of the art on all datasets (especially on large benchmarks), its aggregate performance is strong, and the efficiency improvements (in terms of inference speed / memory usage) make it a meaningful contribution. The contribution can be strengthened by more explicitly clarifying its novelty and positioning within the broader GNN literature in the revised manuscript. Overall, the paper constitutes a solid and practical contribution to GNN design.